# CoDaPO: Confidence and Difficulty-Adaptive Policy Optimization for Language Models

## Abstract

Reinforcement learning (RL) post-training strengthens reasoning in large language models (LLMs), yet the prevailing GRPO algorithm exhibits persistent issues. Using a PRAG lens (Probability, Reward, Advantage, Gradient), we diagnose three mechanisms: (i) *probability inflation*—clipping induces one-way confidence drift with weak KL correction, collapsing entropy; (ii) *advantage contraction*—group normalization dulls update signals as accuracy rises; and (iii) *hierarchical convergence*—easy questions improve quickly while hard ones advance slowly via rare discoveries. We then introduce *CoDaPO*, a confidence- and difficulty–adaptive policy optimization framework that rescales per-trajectory advantages by confidence and difficulty to penalize overconfident errors and align confidence with correctness and difficulty. Across multiple benchmarks, CoDaPO demonstrates consistent improvements on reasoning tasks for small and middle-scale models.

## 1 Introduction

Large language models (LLMs) increasingly support decision making through explicit multi-step reasoning [30, 17, 44]. Reinforcement-learning (RL) post-training algorithms, exemplified by Group Relative Policy Optimization (GRPO) and its variants, enhance such reasoning without gold-standard step-by-step traces by rewarding high-quality trajectories with rule- or model-based estimators [33, 34, 24]. This approach yields gains in mathematics, code generation, and formal logic [34, 26, 40].

Nevertheless, GRPO-style training exhibits three persistent pathologies: (i) *entropy collapse*, in which probability mass concentrates on a few tokens, and thus limits exploration [38, 5]; (ii) *low-contrast advantages*, encountering weaker or even indistinguishable update signals as accuracy rises [45]; and (iii) *limited learning on hard questions*, where rapid convergence on easy questions shrinks overall gradients, limiting further learning on harder questions. These issues remain insufficiently understood and solved, potentially resulting in suboptimal products in the RL post-training of language models.

To pinpoint the root causes of these shortcomings, we carry out a systematic analysis of GRPO's training dynamics through the lens of PRAG: probability, reward, advantage, and gradient. Empirical results and mathematical understanding of post-training Qwen2.5-Math-1.5B [43] with GRPO on the MATH [11] dataset reveal that: (i) *probability increases* because the clipped policy update creates a one-way upward drift in token confidence while the reference KL term provides only a weak pull back to the reference model; trajectory-level credit further inflates non-essential tokens, leading to confidence saturation, collapsed entropy, and occasional overconfidence; (ii) *advantage contracts* as accuracy rises: group normalization compresses the signal so most trajectories cluster near zero advantage, with a thinner but larger-magnitude negative tail, yielding weaker learning signals over time; (iii) *Training converges hierarchically*: easy questions improve quickly via many small positive pushes, whereas hard questions advance through rare correct discoveries followed by amplification, producing slower gains and a lingering error mass until discoveries become common.

Building on these insights, we propose CoDaPO, a confidence- and difficulty–adaptive RL training framework for LLM reasoning. CoDaPO dynamically rescales per-trajectory advantages by *confidence* and *difficulty*: confidence scaling counteracts probability drift and overconfidence, while difficulty scaling combats advantage contraction and accelerates learning on hard problems. At the macro level, CoDaPO (i) normalizes advantages globally at the sequence level, avoiding token-wise scaling that can over-penalize longer responses, and (ii) removes KL regularization, which often restricts exploration by tethering the model to a frozen reference. At the micro level, CoDaPO accommodates diverse reweighting tactics; in our implementation, correct, high-confidence trajectories

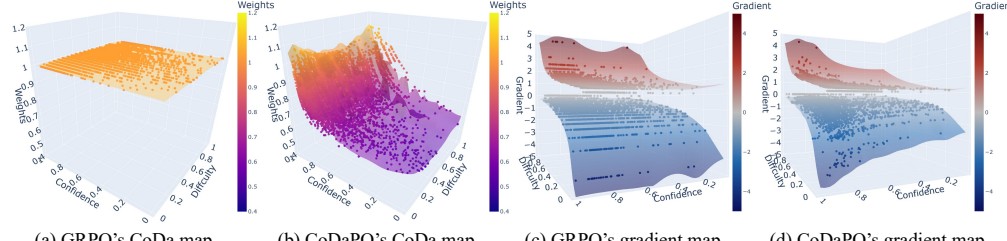

(a) GRPO's CoDa map     (b) CoDaPO's CoDa map     (c) GRPO's gradient map     (d) CoDaPO's gradient map

Figure 1: CoDa maps and gradient maps of GRPO and CoDaPO. Each plot shows trajectories positioned by the model-estimated difficulty (x-axis) and confidence (y-axis). The *CoDa maps* show the joint weighting effect by confidence and difficulty: GRPO uses a constant weight of 1, while CoDaPO emphasizes more confident trajectories for more difficult questions. *Gradient maps* plot the resulting gradients through advantages: raw gradients in GRPO, and weighted gradients in CoDaPO.

on complex questions receive proportionally stronger reinforcement, while overconfident errors are down-weighted. Together, these choices focus updates on the most informative trajectories and mitigate the shortcomings identified in our analysis. In Fig. 1, we compare GRPO with CoDaPO via i) the confidence-difficulty weighting effects from the CoDa map and ii) the resulting gradient map.

Finally, we empirically validate CoDaPO on seven widely used reasoning-task benchmarks. The experimental results indicate that CoDaPO achieves higher reasoning accuracy and better generalization across various benchmarks compared to existing algorithms. This improvement is particularly evident for base models that have not undergone instruction-tuned training. For instance, CoDaPO boosts the accuracy of the Qwen2.5-Math-1.5B model from $31.20\%$ to $71.68\%$ on the in-domain MATH500 benchmark, and from $18.25\%$ to $32.80\%$ on the out-of-domain Olympiad Bench. These findings underscore CoDaPO's significant potential to enhance the reasoning capabilities of LLMs, resulting in substantial gains in accuracy across both in-domain and out-of-domain evaluations.

The main contributions of this work are summarized as follows:

- We provide an empirical and mathematical understanding of GRPO's training dynamics; explain *probability inflation*, *advantage contraction*, and *hierarchical convergence* in post-training (Sec. 3).
- We introduce CoDaPO, a framework that rescales per-trajectory advantages by *confidence* and *difficulty*, prioritizing the most informative trajectories and alleviating GRPO's limitations (Sec. 4).
- We validate CoDaPO on seven reasoning benchmarks and show that it consistently surpasses GRPO and other algorithms, with significant gains for non-instruction-tuned base models (Sec. 5).

## 2 PRELIMINARIES

**Notations.** A post-training dataset $\mathcal{D}$ consists of question–answer pairs $(q, a)$. Given a question $q$, a language model $f_{\theta}$ generates a trajectory $o \sim f_{\theta}(\cdot \mid q)$ that contains both the step-by-step reasoning thoughts and the final answer. We maintain three policies, ordered by update frequency: the *current* policy $f_{\theta}$ (updated every step), a frozen *behavior* policy $f_{\text{old}}$ (a recent snapshot used for sampling), and a *reference* policy $f_{\text{ref}}$ (an older snapshot used for divergence control).

**Group Relative Policy Optimization (GRPO).** GRPO [34] removes the learned value function in PPO [33] and estimates advantages by standardizing rewards across a group of sampled outputs. For each $q$, sample $G$ rollouts from the behavior policy: $\{o_i\}_{i=1}^{G} \sim f_{\text{old}}(\cdot \mid q)$. Compute a deterministic reward $r_i$ for each $o_i$ and form the group $\{r_i\}_{i=1}^{G}$. The (normalized) advantage for rollout $i$ is $\hat{A}_i \triangleq \tilde{r}_i = \frac{r_i - \text{mean}(\{r_i\}_{i=1}^{G})}{\text{std}(\{r_i\}_{i=1}^{G})}$. Let $\rho_{i,t} = \frac{f_{\theta}(o_{i,t}|q,o_{i,<t})}{f_{\text{old}}(o_{i,t}|q,o_{i,<t})}$ be the per-token importance ratio, GRPO maximizes the clipped policy improvement while constraining the policy close to the reference:

$$\mathcal{J}_{\text{GRPO}}(f_{\theta}) \triangleq \mathbb{E}_{(q,a)\sim\mathcal{D}, \{o_i\}\sim f_{\text{old}}(\cdot|q)} \left[ \frac{1}{G}\sum_{i=1}^{G}\frac{1}{|o_i|}\sum_{t=1}^{|o_i|}\min\left(\rho_{i,t}\hat{A}_i, \text{clip}\left(\rho_{i,t}, 1-\epsilon, 1+\epsilon\right)\hat{A}_i\right) - \beta\mathbb{D}_{\text{KL}}[f_{\theta}\|f_{\text{ref}}] \right],$$

where $\text{clip}$ bounds the ratio to stabilize and $\mathbb{D}_{\text{KL}}$ penalizes deviation from the reference policy.

**Related work.** Subsequent efforts refine GRPO's objective, *e.g.*, DAPO [45], Dr. GRPO [23], REINFORCE++ [13], CPPO [22], and GPG [3]; other works extend the setting to multimodal [48, 14, 3, 25, 47] and logical reasoning [40]. Appendix F provides a detailed review of these works.

# 3 ANALYSIS OF THE TRAINING DYNAMICS IN RL POST-TRAINING

## 3.1 THE PRAG FRAMEWORK: PROBABILITY, REWARD, ADVANTAGE, AND GRADIENT

This section analyzes the GRPO using the PRAG framework. For each $(q, a) \in \mathcal{D}$, we sample a *group* of $G$ trajectories $\{o_i\}_{i=1}^G \sim f_{\text{old}}(\cdot \mid q)$ from the behavior policy, where $o_i = (o_{i,1}, \ldots, o_{i,|o_i|})$.

**Probability (token-level).** We operate in log-probability space for stability. For trajectory $o_i$ and token position $t$, define $\ell_{\boldsymbol{\theta}}(o_{i,t}) := \log f_{\boldsymbol{\theta}}(o_{i,t} \mid q, o_{i,<t})$ and $\ell_{\text{old}}(o_{i,t}) := \log f_{\text{old}}(o_{i,t} \mid q, o_{i,<t})$, yielding the importance ratio $\rho_{i,t} := \frac{f_{\boldsymbol{\theta}}(o_{i,t}|q,o_{i,<t})}{f_{\text{old}}(o_{i,t}|q,o_{i,<t})} = \exp(\ell_{\boldsymbol{\theta}}(o_{i,t}) - \ell_{\text{old}}(o_{i,t}))$ w.r.t. $f_{\text{old}}$; if reporting probabilities, $p_{\boldsymbol{\theta}}(o_{i,t}) = \exp(\ell_{\boldsymbol{\theta}}(o_{i,t}))$. We use post-softmax log-probabilities (not logits).

**Reward (trajectory-level).** For simplicity, we only adopt the binary accuracy reward $r_i$ by matching the ground-truth answer $a$ and the predicted answer in trajectory $o_i$. Namely, $r_i :=$ CorrectAnswer$(o_i, a) \in \{0, 1\}$, where $r_i = 1$ if the predicted answer is correct, and $r_i = 0$ otherwise.

**Advantage (trajectory-level).** GRPO replaces model-based value estimation by group-normalizing rewards: $\hat{A}_i := \frac{r_i - \text{mean}(\{r_j\}_{j=1}^G)}{\text{std}(\{r_j\}_{j=1}^G) + \varepsilon}$ with a small $\varepsilon > 0$ (*e.g.*, $10^{-8}$) for stability; treat $\hat{A}_i$ as constant w.r.t. $\boldsymbol{\theta}$ (stop-gradient), and do not backpropagate through the group mean or standard deviation.

**Gradient (token-level).** GRPO maximizes the clipped surrogate with a reference-KL penalty:

$$\mathcal{J}_{\text{GRPO}}(f_{\boldsymbol{\theta}}) = \frac{1}{G} \sum_{i=1}^G \frac{1}{|o_i|} \sum_{t=1}^{|o_i|} \Big( \underbrace{\min(\rho_{i,t}\hat{A}_i, \, \text{clip}(\rho_{i,t}, 1-\epsilon, 1+\epsilon)\hat{A}_i)}_{\text{clipped surrogate}} - \underbrace{\beta \, \mathbb{D}_{\text{KL}}[f_{\boldsymbol{\theta}} \, \| \, f_{\text{ref}}]_{i,t}}_{\text{KL penalty}} \Big), \quad (1)$$

where $\epsilon > 0$, $\text{clip}(x, 1-\epsilon, 1+\epsilon)$ bounds $x$ to $[1-\epsilon, 1+\epsilon]$, and the per-token KL is approximated by $\mathbb{D}_{\text{KL}}[f_{\boldsymbol{\theta}} \, \| \, f_{\text{ref}}]_{i,t} = \frac{f_{\text{ref}}(o_{i,t}|q,o_{i,<t})}{f_{\boldsymbol{\theta}}(o_{i,t}|q,o_{i,<t})} - \log \frac{f_{\text{ref}}(o_{i,t}|q,o_{i,<t})}{f_{\boldsymbol{\theta}}(o_{i,t}|q,o_{i,<t})} - 1$. Let $\ell_{\boldsymbol{\theta}}(o_{i,t}) := \log f_{\boldsymbol{\theta}}(o_{i,t} \mid q, o_{i,<t})$ and $\ell_{\text{ref}}(o_{i,t}) := \log f_{\text{ref}}(o_{i,t} \mid q, o_{i,<t})$, and define $u_{i,t} := \frac{f_{\text{ref}}(o_{i,t}|q,o_{i,<t})}{f_{\boldsymbol{\theta}}(o_{i,t}|q,o_{i,<t})} = \exp(\ell_{\text{ref}}(o_{i,t}) - \ell_{\boldsymbol{\theta}}(o_{i,t}))$. Then the per-token derivative of the KL term is

$$\nabla_{\boldsymbol{\theta}} \, \mathbb{D}_{\text{KL}}[f_{\boldsymbol{\theta}} \, \| \, f_{\text{ref}}]_{i,t} = \Big(1 - \frac{f_{\text{ref}}(o_{i,t} \mid q, o_{i,<t})}{f_{\boldsymbol{\theta}}(o_{i,t} \mid q, o_{i,<t})}\Big) \nabla_{\boldsymbol{\theta}} \ell_{\boldsymbol{\theta}}(o_{i,t}) = \big(1 - u_{i,t}\big) \nabla_{\boldsymbol{\theta}} \ell_{\boldsymbol{\theta}}(o_{i,t}). \quad (2)$$

The clipped surrogate contributes only when the *unclipped* branch is active:

$$\nabla_{\boldsymbol{\theta}} \, \min(\rho_{i,t}\hat{A}_i, \, \text{clip}(\rho_{i,t}, 1-\epsilon, 1+\epsilon)\hat{A}_i) = \begin{cases} \hat{A}_i \, \rho_{i,t} \, \nabla_{\boldsymbol{\theta}} \ell_{\boldsymbol{\theta}}(o_{i,t}), & \text{if unclipped,} \\ 0, & \text{if clipped.} \end{cases} \quad (3)$$

**Gradient (batch-level).** Averaging over trajectories and tokens gives the batch-level gradient:

$$\nabla_{\boldsymbol{\theta}} \mathcal{J}_{\text{GRPO}}(f_{\boldsymbol{\theta}}) = \frac{1}{G} \sum_{i=1}^G \frac{1}{|o_i|} \sum_{t=1}^{|o_i|} \Big[ \underbrace{\mathbf{1}_{\text{unclipped}} \, \hat{A}_i \, \rho_{i,t}}_{\text{group-advantage surrogate}} - \underbrace{\beta\big(1 - u_{i,t}\big)}_{\text{reference KL pull}} \Big] \nabla_{\boldsymbol{\theta}} \ell_{\boldsymbol{\theta}}(o_{i,t}), \quad (4)$$

where the unclipped indicator $\mathbf{1}_{\text{unclipped}} = \mathbb{I}\Big[(\hat{A}_i \geq 0 \land \rho_{i,t} \leq 1+\epsilon) \lor (\hat{A}_i < 0 \land \rho_{i,t} \geq 1-\epsilon)\Big]$.

**Empirical setup.** To track the training dynamics of GRPO, we post-train Qwen2.5-Math-1.5B [43] with GRPO on MATH [11] and evaluate on MATH500 [20]. We compute the confidence/difficulty as $\text{Confidence}(f_{\boldsymbol{\theta}}, q, o_i) = \frac{1}{|o_i|} \sum_{t=1}^{|o_i|} \log f_{\boldsymbol{\theta}}(o_{i,t}|q, o_{i,<t})$, $\text{Difficulty}(f_{\boldsymbol{\theta}}, q, a) = 1 - \frac{1}{G} \sum_{i=1}^G r_i$. We split the training/validation trajectories into five difficulty levels using the difficulty metric: 0-0.2 (level 1), 0.2-0.4 (level 2), 0.4-0.6 (level 3), 0.6-0.8 (level 4), and 0.8-1 (level 5). [1] Next, we present the empirical findings from Figs. 2 and 3 with mathematical understanding. The detailed quantitative statistics of Figs. 2 and 3 is presented in in Tab. 8 and Tab. 9, respectively.

---

[1] In Appendix H, we verify that the model-estimated difficulty well aligns with the ground truth difficulty.

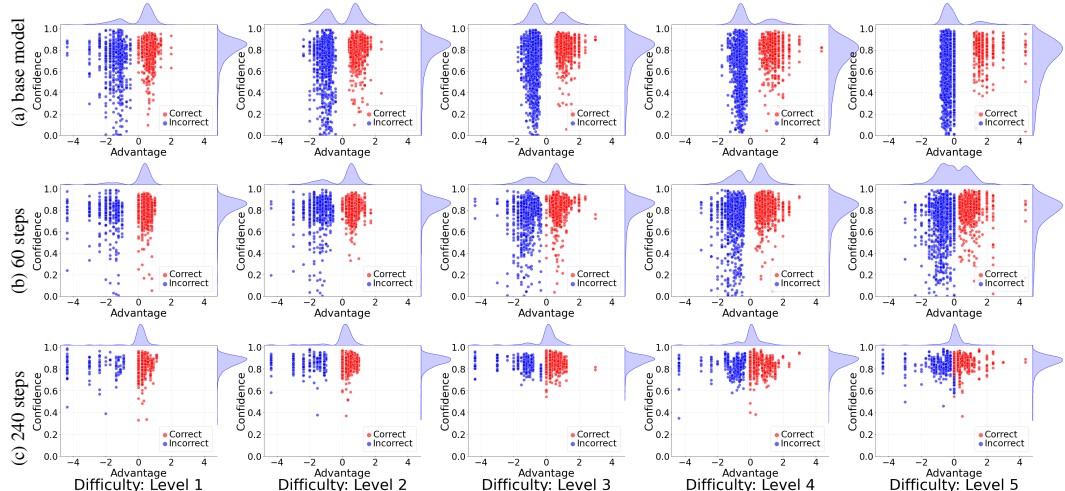

Figure 2: The confidence-advantage distribution of correct/incorrect trajectories in GRPO training. Rows 1 to 3 present the model checkpoints captured at training steps 0 (base model), 60, and 240. Columns 1 through 5 correspond to the difficulty levels of the training questions, numbered 1 to 5. We visualize confidence by exponentiating the mean token log probability for better interpretability.

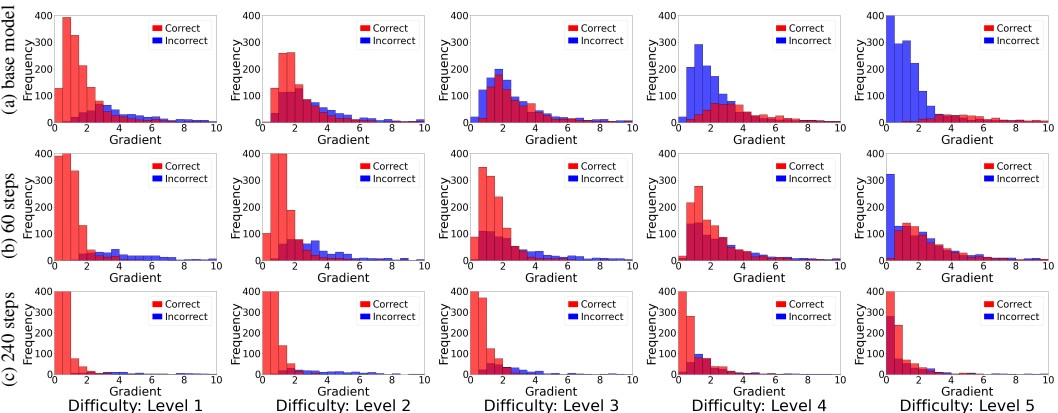

Figure 3: The gradient distribution of correct/incorrect trajectories (the same trajectories as in Fig. 2).

### 3.2 WHY PROBABILITY INCREASES (AND ENTROPY COLLAPSES)

**Empirical finding.** As can be seen from Fig. 2, during GRPO training, model confidences progressively *saturate* near 1 (i.e., $\approx 100\%$). Accuracy improves, but miscalibration emerges: the model also becomes more confident when it is *wrong*. For the base model, confidence distributions are similar across difficulty buckets—higher for correct outputs and highly variable for incorrect ones. As for the post-trained model, the probability mass shifts toward near-certain predictions for both correct and incorrect outputs, consistent with the *entropy collapse* reported by Yu et al. [45].

**Mechanism analysis.** From the per-token ascent direction, a small gradient-ascent step yields

$$\Delta\ell_{\boldsymbol{\theta}}(o_{i,t}) \;\propto\; \underbrace{\mathbf{1}_{\text{unclipped}}\,\hat{A}_i\,\rho_{i,t}}_{\text{group-advantage push}} \;-\; \underbrace{\beta\,(1-u_{i,t})}_{\text{reference-KL pull}}. \tag{5}$$

Below, we dissect how the group-advantage term concentrates probability mass, while the reference-KL term counteracts saturation only weakly when $u_{i,t} \approx 1$.

**Asymmetry from clipping (positive drift).** If $\hat{A}_i > 0$ (high-reward trajectory), the policy term stays active until $\rho_{i,t}$ hits $1 + \epsilon$; while active, it *increases* $\ell_{\boldsymbol{\theta}}$ proportionally to $\rho_{i,t}$. If $\hat{A}_i < 0$ (low-reward trajectory), the policy term is active only while $\rho_{i,t} \geq 1 - \epsilon$; once $\rho_{i,t} < 1 - \epsilon$, the policy term *shuts off* (no further downward push). Since $\rho_{i,t} = \exp(\ell_{\boldsymbol{\theta}} - \ell_{\text{old}})$, increases in the

log-probability $\ell_{\boldsymbol{\theta}}$ *inflate* $\rho_{i,t}$, which further *amplifies* the positive policy term $\hat{A}_i \rho_{i,t}$ until it is capped at $1 + \epsilon$. In contrast, decreases in $\ell_{\boldsymbol{\theta}}$ for $\hat{A}_i < 0$ are *self-limiting*: when $\rho_{i,t}$ drops below $1 - \epsilon$, the policy term vanishes. This clipping asymmetry creates a *one-way drift* that favors increasing $\ell_{\boldsymbol{\theta}}$.

**KL term cannot fully counteract the drift.** The ref-KL contribution is $-\beta(1 - u_{i,t}) = -\beta(1 - p_{\text{ref}}(o_{i,t})/p_{\boldsymbol{\theta}}(o_{i,t}))$. Here, $-\beta(1 - u_{i,t}) < 0$ if $p_{\boldsymbol{\theta}}(o_{i,t}) > p_{\text{ref}}(o_{i,t})$, and $-\beta(1 - u_{i,t}) > 0$ if $p_{\boldsymbol{\theta}}(o_{i,t}) < p_{\text{ref}}(o_{i,t})$. Hence, the KL term *opposes* probability increases when $p_{\boldsymbol{\theta}}(o_{i,t}) > p_{\text{ref}}(o_{i,t})$, and *supports* increases when $p_{\boldsymbol{\theta}}(o_{i,t}) < p_{\text{ref}}(o_{i,t})$. However, (i) when $\hat{A}_i > 0$, the active policy term $\mathbf{1}_{\text{unclipped}} \hat{A}_i \rho_{i,t}$ is positive and often dominates unless $\beta$ is tuned large or $p_{\boldsymbol{\theta}}(o_{i,t}) \gg p_{\text{ref}}(o_{i,t})$; and (ii) when $\hat{A}_i < 0$ and $\rho_{i,t} < 1 - \epsilon$, the policy term is *zero* while the KL term typically points *back toward* $p_{\text{ref}}(o_{i,t})$, preventing probabilities from drifting too low. Thus, the KL term creates a *floor* near the reference, whereas the positive policy term keeps pushing upward until the $1 + \epsilon$ cap.

**Trajectory-level credit amplifies many tokens.** GRPO applies a single scalar advantage $\hat{A}_i$ to every token in a trajectory. In high-reward rollouts, even non-essential tokens receive positive updates while unclipped, concentrating probability and reducing entropy despite limited causal contribution.

### 3.3 WHY ADVANTAGE CONTRACTS

**Empirical finding.** In Fig. 2, as GRPO increases accuracy, group-normalized advantages concentrate near zero. Early on, advantages spread widely within response groups; over training, more trajectories are correct, so most receive small nonnegative advantages, while the few incorrect ones carry large negatives. The net effect is a sharper mass near $0$ and a thinner, rarer negative tail.

**Mechanism analysis.** Denote the group correctness rate by $\bar{r} \in [0, 1]$. GRPO forms advantages $\hat{A}_i = \frac{r_i - \bar{r}}{\sqrt{\bar{r}(1 - \bar{r})} + \varepsilon}, \varepsilon \downarrow 0$. For *correct* and *incorrect* trajectories, the corresponding advantages

$$\hat{A}^{(+)}(\bar{r}) = \frac{1 - \bar{r}}{\sqrt{\bar{r}(1 - \bar{r})}} = \sqrt{\frac{1 - \bar{r}}{\bar{r}}}, \qquad \hat{A}^{(-)}(\bar{r}) = \frac{-\bar{r}}{\sqrt{\bar{r}(1 - \bar{r})}} = -\sqrt{\frac{\bar{r}}{1 - \bar{r}}}. \tag{6}$$

As accuracy improves, $\bar{r} \uparrow 1$, hence $\hat{A}^{(+)}(\bar{r}) \to 0$ while $\hat{A}^{(-)}(\bar{r}) \to -\infty$. Pooling across groups yields the mixture $\hat{A} \sim \bar{r} \delta_{\hat{A}^{(+)}(\bar{r})} + (1 - \bar{r}) \delta_{\hat{A}^{(-)}(\bar{r})}$, whose mass concentrates at $0$ because $\bar{r} \to 1$ and $\hat{A}^{(+)}(\bar{r}) \to 0$, while the heavy negative tail persists with vanishing weight $1 - \bar{r}$. This explains the observed contraction, despite per-group standardization (zero mean, unit variance) by construction.

### 3.4 WHY TRAINING CONVERGES HIERARCHICALLY

**Empirical finding.** As in Fig. 2 and Fig. 3, *easy questions* begin with high confidence, nonnegative advantages, and relatively large initial gradient norms; GRPO further raises confidence while contracting advantages and gradients toward zero. *Hard questions* start with low confidence, predominantly nonpositive advantages, and small initial gradients; training shifts the distribution upward (higher confidence, more positives), yet gradients also decay rapidly and a substantial error mass remains.

**Mechanism analysis.** Denote the per-token ascent weight $w_{i,t} = \mathbf{1}_{\text{unclipped}} \hat{A}_i \rho_{i,t} - \beta(1 - u_{i,t})$. Specifically, $w_{i,t} > 0$ increases the token's log-probability, while $w_{i,t} < 0$ decreases it. Taking expectation over trajectories for question $q$ gives

$$\mathbb{E}[w_{i,t} \mid q] \approx \bar{r} \, \mathbb{E}\left[\mathbf{1}_{\text{unclipped}}^{(+)} \hat{A}^{(+)}(\bar{r}) \rho_{i,t}\right] + (1 - \bar{r}) \, \mathbb{E}\left[\mathbf{1}_{\text{unclipped}}^{(-)} \hat{A}^{(-)}(\bar{r}) \rho_{i,t}\right] - \beta \, \mathbb{E}[1 - u_{i,t}]. \tag{7}$$

**Easy questions ($\bar{r}$ large).** When $\bar{r}$ is high, $\hat{A}^{(+)}(\bar{r})$ is small (nonnegative) while $\hat{A}^{(-)}(\bar{r})$ is large in magnitude but rare. Clipping leaves positives active until $\rho_{i,t} = 1 + \epsilon$ and deactivates negatives once $\rho_{i,t} < 1 - \epsilon$. Thus, a rare incorrect trajectory can cause a strong *instantaneous* negative push, but it is (i) down-weighted by $1 - \bar{r}$, (ii) quickly capped by clipping, and (iii) then countered by the KL pull; meanwhile, many correct trajectories accumulate small positive pushes. Net effect: $\mathbb{E}[w_{i,t} \mid q] \gtrsim 0$ and probabilities rise; as $\bar{r} \uparrow 1$, $\hat{A}^{(+)}(\bar{r}) \downarrow 0$ (implicit annealing), with advantages drifting toward $0^+$. At the same time, initial gradient norms are high, but as the model quickly converges on these easy items, gradients rapidly decay, leaving little room for further updates.

**Hard questions ($\bar{r}$ small).** When $\bar{r}$ is low, $\hat{A}^{(+)}(\bar{r})$ is large but rare, whereas $\hat{A}^{(-)}(\bar{r})$ is moderate and frequent. Clipping quickly deactivates negative updates once $\rho_{i,t} < 1 - \epsilon$, after which the KL term $-\beta(1 - u_{i,t})$ maintains a soft floor near $f_{\text{ref}}$; in contrast, rare correct trajectories deliver strong positive pushes (via large $\hat{A}^{(+)}(\bar{r})$) until $\rho_{i,t} = 1 + \epsilon$. Thus $\mathbb{E}[w_{i,t} \mid q]$ is typically positive but smaller than for easy items. Progress exhibits two phases: *discovery*, where a correct trajectory appears with probability $1 - (1 - \bar{r})^G \approx G\bar{r}$, and *amplification*, where discovered correct responses are reinforced up to the cap and become more likely subsequently. Advantages shift rightward more slowly, a nontrivial error mass persists until discovery becomes common, and confidence increases gradually. The slow improvement on hard questions is compounded by the rapid convergence of easy questions, which lowers the total gradient magnitude and constrains further updates on difficult items.

## 4 CONFIDENCE AND DIFFICULTY-ADAPTIVE POLICY OPTIMIZATION

This section introduces CoDaPO, a reinforcement learning framework for LLM reasoning that rescales per-trajectory advantages by *confidence* and *difficulty*. Confidence scaling curbs one-way probability drift by tying update size to certainty, penalizing high-confidence mistakes, and tempering already-saturated tokens. Difficulty scaling re-inflates low-contrast advantages as accuracy rises, sustaining gradients on hard items and speeding discovery. Together, these bounded reweightings direct learning to the most informative trajectories and mitigate the shortcomings identified in Sec. 3.

### 4.1 THE OPTIMIZATION FRAMEWORK

**The optimization objective.** Given $G$ rollouts from $f_{\text{old}}$, CoDaPO maximizes the objective

$$\mathcal{J}_{\text{CoDaPO}}(f_{\boldsymbol{\theta}}) \triangleq \mathbb{E}_{(q,a) \sim \mathcal{D}, \{o_i\}_{i=1}^G \sim f_{\text{old}}(\cdot|q)} \frac{1}{\sum_{i=1}^G |o_i|} \sum_{i=1}^G \sum_{t=1}^{|o_i|} \left[ \min\left(\rho_{i,t}\hat{A}_i, \ \text{clip}(\rho_{i,t}, 1 - \epsilon, 1 + \epsilon)\hat{A}_i\right) \hat{c}_i \hat{d}_q \right], \tag{8}$$

where $\rho_{i,t} := \frac{f_{\boldsymbol{\theta}}(o_{i,t}|q,o_{i,<t})}{f_{\text{old}}(o_{i,t}|q,o_{i,<t})}$ and $\hat{A}_i = \frac{r_i - \text{mean}(\{r_j\}_{j=1}^G)}{\text{std}(\{r_j\}_{j=1}^G)}$ uses the accuracy reward $r_i \in \{0, 1\}$ (as in Sec. 3). The factors $\hat{c}_i$ (confidence) and $\hat{d}_q$ (difficulty) reweight $\hat{A}_i$; detailed in following subsection.

**Sample-level normalization.** We replace GRPO's per-trajectory averaging $\frac{1}{G} \sum_i \frac{1}{|o_i|} \sum_t (\cdot)$ with a *micro-average* over tokens $\frac{1}{\sum_i |o_i|} \sum_{i,t} (\cdot)$. This gives each token equal weight and removes the length penalty induced by the extra $1/|o_i|$, which systematically down-weights longer (often harder, more explicit) solutions [45, 23]. Micro-averaging also yields an unbiased stochastic gradient for a per-token objective under variable lengths, reduces variance by aggregating all tokens in the minibatch, and avoids the pathological incentive to shorten outputs for larger updates. Crucially, it preserves the relative per-token update directions while eliminating length as a confounder in credit assignment.

**Discarding KL regularization.** GRPO's KL term stabilizes training but (i) restricts exploration, which is undesirable in reasoning tasks, and (ii) increases computation due to an extra $f_{\text{ref}}$ forward pass. We therefore omit it, as Yu et al. [45], Chu et al. [3], to reduce cost and encourage discovery of new solution paths; stability and coherence are maintained by bounded confidence/difficulty weights.

**Gradient and effective token weight.** For clarity and stability, we *stop gradient* through $\hat{A}_i, \hat{c}_i, \hat{d}_q$. The resulting gradient is $\nabla_{\boldsymbol{\theta}} \mathcal{J}_{\text{CoDaPO}} = \frac{1}{\sum_i |o_i|} \sum_{i,t} \left( \rho_{i,t} \hat{A}_i \hat{c}_i \hat{d}_q \right) \nabla_{\boldsymbol{\theta}} \ell_{\boldsymbol{\theta}}(o_{i,t})$. A small step of gradient ascent increases token log-probabilities by

$$\Delta \ell_{\boldsymbol{\theta}}(o_{i,t}) \propto w_{i,t}^{\text{CoDaPO}} := \rho_{i,t} \hat{A}_i \hat{c}_i \hat{d}_q. \tag{9}$$

Thus *all* deviations from GRPO (no clipping, no ref-KL) enter via the scalar weight $w_{i,t}^{\text{CoDaPO}}$. Taking expectation conditioned on $q$ and decomposing over correct or incorrect trajectories yields

$$\mathbb{E}\left[ w_{i,t}^{\text{CoDaPO}} \mid q \right] = \bar{r} \, \hat{d}_q \, \mathbb{E}\left[ \hat{A}^{(+)}(\bar{r}) \, \hat{c}^{(+)} \rho_{i,t} \right] + (1 - \bar{r}) \, \hat{d}_q \, \mathbb{E}\left[ \hat{A}^{(-)}(\bar{r}) \, \hat{c}^{(-)} \rho_{i,t} \right], \tag{10}$$

where $\hat{A}^{(+)}(\bar{r}) = \sqrt{\frac{1-\bar{r}}{\bar{r}}}, \hat{A}^{(-)}(\bar{r}) = -\sqrt{\frac{\bar{r}}{1-\bar{r}}}$ (cf. Eqn. (6)), and $\hat{c}^{(+)}$ and $\hat{c}^{(-)}$ denote $\hat{c}_i$ conditioned on $r_i = 1$ and $r_i = 0$, respectively. In the following section, we provide an instantiation of $\hat{c}_i$ and $\hat{d}_q$.

## 4.2 IMPLEMENTATION

Here, we present the implementation of CoDaPO (Alg. 1). The reweighting functions below deliver the strongest positive reinforcement to *correct, high-confidence* trajectories on *difficult* questions.

$$\hat{c}_i = \sigma\left(\frac{\frac{1}{G}\sum_{i=1}^{G}\frac{1}{|o_i|}\sum_{t=1}^{|o_i|}\log f_{\boldsymbol{\theta}}(o_{i,t}|q,o_{i,<t})}{\frac{1}{|o_i|}\sum_{t=1}^{|o_i|}\log f_{\boldsymbol{\theta}}(o_{i,t}|q,o_{i,<t})}\right), \quad \hat{d}_q = 1 + \delta(\frac{1}{G}\sum_{i=1}^{G}r_i - \frac{1}{2})^2, \tag{11}$$

where $\hat{c}_i$ up-weights high-confidence trajectories. The sigmoid function $\sigma(\cdot)$ bounds $\hat{c}_i$, preventing extreme weights from heavy-tailed log-probabilities. The quadratic difficulty weight $\hat{d}_q$ with scaling hyperparameter $\delta$ yields a U-shaped curriculum that up-weights very hard or easy questions. Together, the difficulty reweighting determines which questions matter most, while the confidence reweighting identifies which trajectories within each question group deserve greater emphasis. Their combined effect both penalizes overconfident errors and aligns confidence with correctness and difficulty.

**Advantage reweighting *w.r.t.* confidence.** We compare each output's average log-probability to the group average. Let $\bar{\ell}_i := \frac{1}{|o_i|}\sum_t \ell_{\boldsymbol{\theta}}(o_{i,t})$ and $\bar{\ell} := \frac{1}{G}\sum_i \bar{\ell}_i$, and define $\hat{c}_i = \sigma(\bar{\ell}/\bar{\ell}_i)$. Under the typical regime $\bar{\ell}_i, \bar{\ell} < 0$, one has $\partial\hat{c}_i/\partial\bar{\ell}_i > 0$ and $\partial\hat{c}_i/\partial\bar{\ell} < 0$, so higher-confidence outputs (less negative $\bar{\ell}_i$) receive larger $|\hat{A}_i|$-scaled updates when correct and stronger penalties when incorrect. Since large $|\hat{A}_i|$ only occurs for incorrect trajectories on easy questions and correct trajectories on hard questions, confidence reweighting ensures that these responses receive the strongest rewards or penalties, while other responses $|\hat{A}_i| \approx 0$ are minimally affected.

**Remark 4.1** (Low-confidence emphasis (optional)). *If the goal is to prioritize correct but low-confidence trajectories, an antitone alternative is $\hat{c}_i^{\star} = \sigma\big(\alpha(\bar{\ell} - \bar{\ell}_i)\big)$, or its z-score variant $\hat{c}_i^z = \sigma\big(\alpha\frac{\bar{\ell}-\bar{\ell}_i}{\mathrm{std}(\{\bar{\ell}_j\})+\epsilon}\big)$, with temperature $\alpha > 0$. These variants increase the weight when $\bar{\ell}_i$ is smaller than the group average (i.e., lower confidence), with the z-score form adding scale invariance.*

**Advantage reweighting *w.r.t.* difficulty.** The group accuracy $\bar{r}$, dynamically estimated from the model's current performance, is used to measure question difficulty. For correct trajectories, the effective multiplier is $S_{\mathrm{diff}}(\bar{r}) = \hat{A}^{(+)}(\bar{r})\hat{d}_q = \sqrt{\frac{1-\bar{r}}{\bar{r}}}\big(1 + \delta\big(\bar{r} - \frac{1}{2}\big)^2\big)$. As $\bar{r} \downarrow 0$ (hard items), $S_{\mathrm{diff}}(\bar{r}) \uparrow \infty$ (strong emphasis); as $\bar{r} \uparrow 1$ (easy items), $S_{\mathrm{diff}}(\bar{r}) \downarrow 0$ (self-annealing). For wrong trajectories, the multiplier is $S_{\mathrm{diff}}(\bar{r}) = \hat{A}^{(-)}(\bar{r})\hat{d}_q = -\sqrt{\frac{\bar{r}}{1-\bar{r}}}\big(1 + \delta\big(\bar{r} - \frac{1}{2}\big)^2\big)$, so $S_{\mathrm{diff}}(\bar{r}) \to 0^-$ as $\bar{r} \downarrow 0$ (lenient on very hard items) and $S_{\mathrm{diff}}^-(\bar{r}) \to -\infty$ as $\bar{r} \uparrow 1$ (sharp penalties on easy items). Thus the U-shaped difficulty weight amplifies rare correct signals on hard questions to speed discovery, while strongly discouraging mistakes on easy questions. Around $\bar{r} \approx \frac{1}{2}$, both effects remain moderate.

**Proposition 4.1** (U-shaped emphasis across difficulty). *For $\hat{d}_q = 1 + \delta(\bar{r} - \frac{1}{2})^2$ with $\delta \geq 0$, one has $\hat{d}_q \geq 1$ with a unique minimum at $\bar{r} = \frac{1}{2}$, and $\frac{\partial}{\partial\bar{r}}(\hat{d}_q\,\mathbb{E}[w_{i,t}^{CoDaPO} \mid q])$ changes sign at $\bar{r} = \frac{1}{2}$ (for fixed $\rho_{i,t}$ and $\hat{c}_i$). Hence, CoDaPO strictly up-weights both very hard ($\bar{r} \approx 0$) and very easy ($\bar{r} \approx 1$) items, amplifying rare correct trajectories when $\bar{r}$ is small and sharpening penalties near $\bar{r} \approx 1$.*

**Strongest positive reinforcement: correct, high-confidence, hard.** In the small-step regime with $\rho_{i,t} \approx 1$, two *correct* outputs $o_i, o_j$ on the same question satisfy $\frac{w_{i,t}^{\mathrm{CoDaPO}}}{w_{j,t}^{\mathrm{CoDaPO}}} \approx \frac{\hat{c}_i}{\hat{c}_j}$, since $\hat{A}_i = \hat{A}_j = \hat{A}^{(+)}(\bar{r})$ and $\hat{d}_q$ are shared within the group. With a confidence weight *increasing* in confidence, high-confidence correct outputs receive larger updates. Across questions, correct rollouts also gain the multiplicative boost $S_{\mathrm{diff}}(\bar{r}) = \hat{A}^{(+)}(\bar{r})\hat{d}_q$, which diverges as $\bar{r} \downarrow 0$. Consequently, $\hat{A}^{(+)}(\bar{r})\,\hat{d}_q\,\hat{c}_i$ is maximized for *hard* questions with *high-confidence*, *correct* trajectories.

## 5 EXPERIMENTS

This section evaluates CoDaPO and relevant RL methods on mathematical reasoning tasks. Selected baselines include GRPO [34] and concurrent algorithms DAPO [45], Dr. GRPO [23], and GPG [3].

**Training setup.** We post-train Qwen2.5-1.5B-Instruct, Qwen2.5-Math-1.5B, and Qwen2.5-Math-7B on MATH [20, 11, 42, 43], with TRL [37] and 4×A100 GPUs (2 for inference, 2 for optimization).

| Base Model | Algorithm | Datasets | | | | | | | Average |
|---|---|---|---|---|---|---|---|---|---|
| | | MATH 500 | AIME 2024 | AIME 2025 | AMC 2023 | Olympiad Bench | Minerva | GSM8K | |
| Qwen2.5-1.5B-Instruct | Base | 49.36 | 3.00 | 0.00 | 23.50 | 17.01 | 16.84 | 59.56 | 24.18 |
| | GRPO | 55.08 | 3.00 | **1.33** | 26.00 | 19.08 | **22.35** | 73.92 | 28.68 |
| | DAPO | 56.04 | 2.67 | 0.00 | 27.50 | 18.90 | 21.91 | 73.04 | 28.58 |
| | Dr. GRPO | **56.24** | 3.33 | 0.00 | 25.50 | 19.11 | 21.18 | 73.80 | 28.45 |
| | GPG | 54.36 | **4.00** | 1.33 | 23.00 | 19.29 | 21.54 | 74.10 | 28.34 |
| | **CoDaPO (ours)** | 55.60 | 3.33 | 1.33 | **31.00** | **19.44** | 22.21 | **74.30** | **29.60** |
| | | (6.24↑) | (0.33↑) | (1.33↑) | (7.50↑) | (2.43↑) | (5.37↑) | (14.74↑) | (5.42↑) |
| Qwen2.5-Math-1.5B | Base | 31.20 | 6.00 | 0.00 | 29.50 | 18.25 | 9.63 | 34.01 | 18.37 |
| | GRPO | 68.48 | 12.67 | 8.00 | 51.00 | 30.55 | 28.53 | 79.56 | 39.82 |
| | DAPO | 71.12 | 14.00 | 8.00 | 49.00 | 31.97 | **32.13** | 82.38 | 41.23 |
| | Dr. GRPO | 70.68 | 10.67 | 10.67 | 49.50 | 31.67 | 30.51 | 82.17 | 40.84 |
| | GPG | 70.60 | 11.33 | 5.33 | 47.00 | 31.97 | 29.49 | 82.38 | 39.73 |
| | **CoDaPO (ours)** | **71.68** | **14.67** | **10.83** | **53.50** | **32.80** | 31.69 | **83.50** | **42.67** |
| | | (40.48↑) | (8.67↑) | (10.83↑) | (24.00↑) | (14.55↑) | (22.06↑) | (49.49↑) | (24.30↑) |

Table 1: The main results of the post-training experiments (in accuracy). Note that the **boldface** numbers mean the best results, while the underlined numbers indicate the second-best results. We also show the absolute improvement of CoDaPO over the base model for each benchmark.

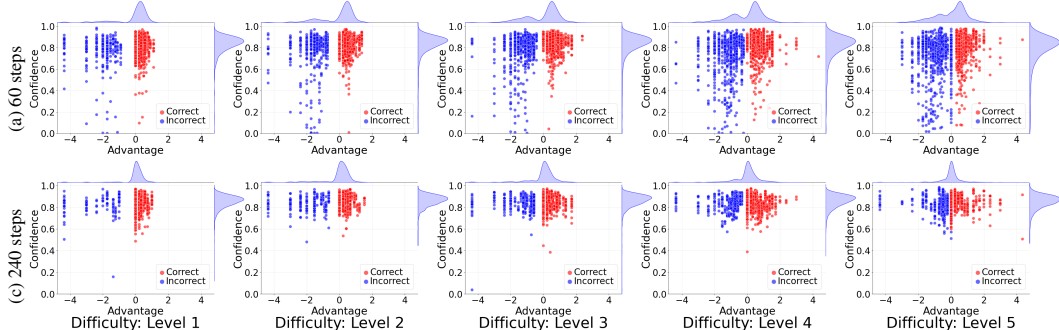

Figure 4: The confidence-advantage distribution of correct/incorrect trajectories in CoDaPO training.

With a common recipe, we sample 12 rollouts per group for each question, train for 3 epochs with learning rate $3 \times 10^{-6}$, and early-stop on reward convergence. The difficulty weight is set to $\delta = 0.4$.

**Evaluation setup.** We adopt the Qwen2.5-Math evaluation codebase for consistent and reliable measurement. For each question, we sample 5 responses at temperature 0.6 and report the mean accuracy. Evaluations span seven reasoning benchmarks: MATH500 [20, 11], AIME 2024, AIME 2025, AMC 2023, OlympiadBench [15], Minerva [18], and GSM8K [4].

**Main results.** We summarize the following observations *w.r.t.* the experimental results in Tab. 1:

- **CoDaPO can effectively improve LLMs' reasoning abilities.** Applying CoDaPO to various models trained on the MATH training set consistently yields effectively performance improvements across different benchmarks. For Qwen2.5-Math-1.5B, the average across seven benchmarks improves from 18.37% to 42.67%, with larger gains for non-instruction-tuned base models.

- **CoDaPO exhibits generalization capabilities across problem types.** Though post-trained only on the MATH training set, CoDaPO demonstrates notably strong generalization to out-of-domain datasets: accuracy rises from 18.25% to 32.80% on Olympiad Bench and from 9.63% to 31.69% on Minerva, highlighting its ability to enhance reasoning even on unseen problem types.

- **CoDaPO surpasses baseline methods in reasoning accuracy.** The comparison of CoDaPO with other RL algorithms reveals that CoDaPO outperforms these baselines on most benchmarks, improving average accuracy by 2.85% over GRPO and 1.44% over DAPO on Qwen2.5-Math-1.5B, showing the effectiveness of CoDaPO's confidence- and difficulty-adaptive design.

**The training dynamics of CoDaPO.** Fig. 4 and Fig. 5 present the confidence-advantage distribution and gradient distribution of the Qwen2.5-Math-1.5B model during CoDaPO training. We observe:

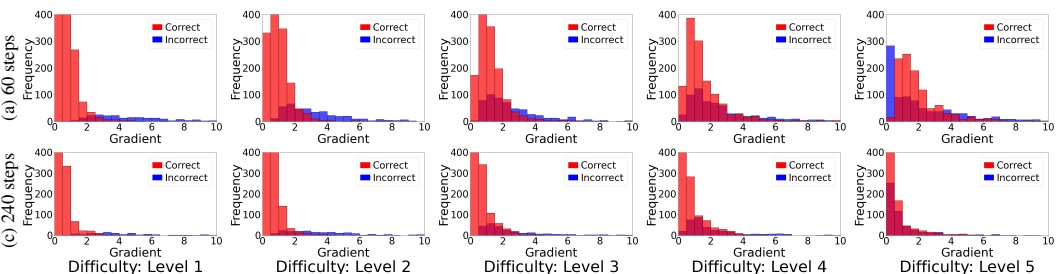

Figure 5: The gradient distribution of correct/incorrect trajectories in CoDaPO training.

| Models | MATH 500 | AIME 2024 | AIME 2025 | AMC 2023 | Olympiad Bench | Minerva | GSM8K | Average |
|---|---|---|---|---|---|---|---|---|
| Qwen2.5-Math-7B | 47.24 | 12.92 | 5.00 | 33.91 | 14.44 | 9.81 | 47.84 | 24.45 |
| GRPO | 74.62 | 26.35 | 11.25 | 59.06 | 36.51 | 37.64 | 89.40 | 47.83 |
| DAPO | 77.78 | 24.37 | 10.62 | 58.67 | 39.27 | 38.06 | 87.38 | 48.02 |
| CoDaPO | **78.41** | 24.79 | **13.33** | **60.31** | **39.96** | **40.30** | 88.28 | **49.34** |

Table 2: Results of post-training experiments on Qwen2.5-Math-7B.

- **CoDaPO enhances the model's accuracy on difficult problems.**
  On high-difficulty subsets (Levels 4-5), CoDaPO's advantage distribution is more positively concentrated than GRPO (Fig. 4, 2), indicating higher accuracy on these questions. This arises from its difficulty-adaptive mechanism (Sec. 4), which assigns greater weights to challenging questions with sparse rewards.

- **CoDaPO smooths gradient updates across difficulty levels.** Compared to GRPO (Fig. 5, Fig. 3), CoDaPO consistently reduces gradient magnitude across all levels. CoDaPO tempers updates on both correct and incorrect responses, ensuring the model learns from all samples effectively while preventing rapid convergence on easy questions from dominating updates on harder questions.

**Ablation studies.** We further validate the effectiveness of CoDaPO's reweighting design:

- **Extension to larger models.** We apply CoDaPO to Qwen2.5-Math-7B under the same settings, and the results are shown in Tab. 2. CoDaPO achieves the best average performance of $49.34\%$, demonstrating its robustness and consistent effectiveness across different base models.

- **Backbone transferability.** We train Llama-3.2-1B-Instruct with CoDaPO to evaluate the effectiveness of our method on a different model backbone. As shown in Tab. 3, CoDaPO achieves the best performance with margins similar to results in Tab. 1. This indicates that CoDaPO operates on general properties of the GRPO update rather than relying on model-specific characteristics.

- **Cross-domain generalization.** We evaluate CoDaPO on cross-domain benchmarks. As shown in Tab. 4, even when trained solely on MATH, CoDaPO achieves strong performance on science and coding tasks, surpassing other baselines and demonstrating its excellent generalization ability.

- **Individual components.** We train with each mechanism separately, as shown in Tab. 5. Both confidence and difficulty signals contribute meaningful improvements across different benchmarks. The gains are complementary, and the full CoDaPO method achieves the best overall performance.

- **Reweighting strategies.** We evaluate CoDaPO trained with different difficulty reweighting strategies. As shown in Tab. 6, U-shape reweighting generally achieves the best performance, indicating that simultaneously emphasizing rare correct answers on hard questions and rare errors on easy questions can effectively guiding the model to focus on the most informative responses.

- **KL regularization.** We train the model with a KL coefficient of $0.01$. As shown in Tab. 10 and Tab. 11, removing the KL term yields better performance without degrading language ability.

- **Hyperparameter sensitivity.** We vary the $\delta$ parameter in the difficulty reweighting term. As shown in Tab. 13, all choices yield improvements over the baseline, indicating that CoDaPO is insensitive to hyperparameters and that the difficulty reweighting term is robust.

- **Seed robustness.** We run experiments with different seeds to evaluate the impact of RL noise on the results. As shown in Tab. 14, CoDaPO consistently achieves the best performance across seeds, demonstrating that its performance gains are robust rather than coincidental.

| Model | MATH 500 | AIME 2024 | AMC 2023 | Olympiad Bench | GSM8K |
|---|---|---|---|---|---|
| Llama3.2-1B-Instruct | 21.69 | 0.83 | 10.39 | 5.04 | 5.58 |
| GRPO | 30.13 | 2.08 | 14.22 | 6.62 | 48.79 |
| CoDaPO | **31.30** | **3.96** | **15.08** | **6.81** | **51.32** |

Table 3: Results of post-training experiments on Llama3.2-1B-Instruct. Best results are shown in **bold**.

| Model | MMLU STEM | GPQA | HumanEval |
|---|---|---|---|
| Qwen2.5-Math-1.5B | 11.53 | 9.85 | 29.27 |
| GRPO | 46.85 | 19.70 | 35.98 |
| CoDaPO | **47.64** | **22.25** | **39.02** |

Table 4: Evaluation results on cross-domain benchmarks of Qwen2.5-Math-1.5B.

| Model | MATH 500 | AIME 2024 | AIME 2025 | AMC 2023 | Olympiad Bench | Minerva | GSM8K | Average |
|---|---|---|---|---|---|---|---|---|
| Qwen2.5-Math-1.5B (base) | 31.20 | 6.00 | 0.00 | 29.50 | 18.25 | 9.63 | 34.01 | 18.37 |
| + GRPO | 68.48 | 12.67 | 8.00 | 51.00 | 30.55 | 28.53 | 79.56 | 39.82 |
| + confidence reweighting | 71.52 | 14.17 | 9.79 | 51.41 | 34.10 | 32.16 | 82.88 | 42.29 |
| + difficulty reweighting | 71.32 | 14.69 | 10.42 | 51.09 | 33.88 | 31.87 | 83.05 | 42.33 |
| + both (CoDaPO) | 71.68 | 14.67 | 10.83 | 53.50 | 32.80 | 31.69 | 83.50 | 42.67 |

Table 5: Performance of Qwen2.5-Math-1.5B trained with different CoDaPO components.

| Model | MATH 500 | AIME 2024 | AIME 2025 | AMC 2023 | Olympiad Bench | Minerva | GSM8K | Average |
|---|---|---|---|---|---|---|---|---|
| Qwen2.5-Math-1.5B | 31.20 | 6.00 | 0.00 | 29.50 | 18.25 | 9.63 | 34.01 | 18.37 |
| CoDaPO (linear reweighting) | 71.16 | **15.62** | 9.58 | 52.03 | **33.05** | 30.66 | 80.48 | 41.80 |
| CoDaPO (U-shape reweighting) | **71.68** | 14.67 | **10.83** | **53.50** | 32.80 | **31.69** | **83.50** | **42.67** |

Table 6: Performance of CoDaPO trained with different difficulty reweighting strategies.

- **Sampling settings.** We evaluate under different decoding temperatures. As shown in Tab. 15, CoDaPO consistently outperforms the baseline across all settings. Moreover, performance shifts caused by temperature are largely synchronous across methods (i.e., accuracy rises or falls together).
- **Test-time scaling.** To ensure that CoDaPO genuinely improves reasoning ability rather than merely narrowing the scope of solvable problems, we evaluate on AIME25 using Pass@K metrics. As shown in Fig. 14 and Fig. 15, CoDaPO consistently surpasses baselines across all $k$. This confirms that CoDaPO strengthens general reasoning rather than overfitting to a subset of problems.

In summary, our ablation studies confirm that each component of CoDaPO contributes meaningfully to its performance, and the combination of confidence- and difficulty-aware reweighting achieves the strongest results. CoDaPO's effectiveness is robust across different model sizes, backbones, seeds, hyperparameters, decoding settings, and cross-domain tasks, and it consistently improves reasoning ability under Pass@K evaluation. These results demonstrate that CoDaPO reliably enhances the underlying reasoning distribution rather than merely optimizing specific datasets or configurations.

**Case Studies.** We present several representative case studies in Appendix I. Compared to GRPO, CoDaPO demonstrates more coherent and logically structured reasoning trajectories.

# 6 CONCLUSION

In this work, we use the PRAG lens to GRPO's core failure modes, *probability inflation*, *advantage contraction*, and *hierarchical convergence*. We introduce *CoDaPO*, which rescales per-trajectory advantages by *confidence* (tempering overconfident errors and one-way drift) and *difficulty* (re-inflating signals and sustaining learning on hard items). Across multiple benchmarks, CoDaPO delivers consistent gains for small–mid-scale models, improving both in-domain and out-of-domain mathematical reasoning. These results suggest that bounded, information-aware reweighting stabilizes RL post-training and allocates credit more effectively. Future work includes more confidence-difficulty schedules, integration with tools, and extensions to larger models and broader domains.

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

# Appendix

## A    ETHIC STATEMENT

The study does not involve human subjects, data set releases, potentially harmful insights, applications, conflicts of interest, sponsorship, discrimination, bias, fairness concerns, privacy or security issues, legal compliance issues, or research integrity issues.

## B    IMPACT STATEMENT

This work aims to advance the field of machine learning and large language models, especially the capabilities of machine reasoning. By making step-by-step reasoning more accurate and sample-efficient, CoDaPO can underpin safer autonomous code assistants, more reliable tutoring systems, and faster hypothesis generation in science, each with clear public benefits. However, the same improvements lower the technical barrier for malicious actors to automate sophisticated scams or targeted disinformation, and large-scale post-training increases energy use and may widen the gap between industry and resource-constrained labs. Therefore, we believe that careful alignment, monitoring, and investment in greener training practices are necessary to ensure that the benefits outweigh the risks. We do not find any negative societal consequences of our work.

## C    REPRODUCTION STATEMENT

The experimental setups for training and evaluation are described in detail in Appendix G, and the experiments are all conducted using public datasets. We provide the link to our source codes to ensure the reproducibility of our experimental results: `https://anonymous.4open.science/r/CoDaPO-submission-code-DE28`.

## D    LLM USAGE DISCLOSURE

This submission was prepared with the assistance of LLMs, which were utilized for polishing content and checking grammar. The authors assume full responsibility for the entire content of the manuscript. It is confirmed that no LLM is listed as an author.

## E    FURTHER DISCUSSIONS

**Limitation.**    While our results are promising, several important limitations should be noted. First, our evaluation is limited to 1.5B parameter models, and the scaling behavior of CoDaPO to models with tens or hundreds of billions of parameters remains unknown. This is particularly important as larger models may exhibit different advantage-confidence dynamics. Second, although we benchmarked on seven reasoning tasks, post-training was confined to the MATH dataset, so the robustness of CoDaPO to other domains (*e.g.*, coding) remains unclear. Third, our experiments were conducted under moderate compute, typical for small-scale models, and we have yet to quantify resource usage or optimize for large-scale or production-scale efficiency in RL-based post-training.

## F    RELATED WORK

**Supervised Fine-tuning (SFT)** finetunes the policy model to predict the next token on data that is more relevant to the downstream task. The objective of SFT is to maximize the token-wise log probability of dataset-collected outputs $\mathcal{O}$, which are treated as the ground truth for training. Namely,

$$\mathcal{J}_{\text{SFT}}(f_{\boldsymbol{\theta}}) \triangleq \mathbb{E}_{(q,a)\sim\mathcal{D},o\sim\mathcal{O}(q)} \left( \frac{1}{|o|} \sum_{t=1}^{|o|} \log f_{\boldsymbol{\theta}}(o_t|q, o_{<t}) \right). \tag{12}$$

Although simple in optimization, SFT has several significant drawbacks. SFT focuses on exploiting (memorizing, to some extent) the dataset-collected outputs, resulting in limited generalization power, especially in the out-of-distribution scenarios [2]. Besides, collecting and annotating the output data can be expensive and often requires domain-specific knowledge in solving particular questions.

**Proximal Policy Optimization (PPO)** [33] is an actor-critic RL algorithm that is widely used in the RL fine-tuning stage of LLMs. Simplifying the TRPO [31], PPO maximizes the advantage $A_t$ of the model-generated output $o$ without the need to collect ground truth outputs. Here, the advantage $A_t$ is computed by the Generalized Advantage Estimation (GAE) [32], taking 1) the output value estimated by a trainable value model and 2) the KL penalty between $f_\theta$ and $f_{\text{ref}}$. PPO maximizes the objective:

$$\mathcal{J}_{\text{PPO}}(f_\theta) \triangleq \mathbb{E}_{(q,a)\sim\mathcal{D},o\sim f_{\text{old}}(\cdot|q)} \frac{1}{|o|} \sum_{t=1}^{|o|} \min\left[ \frac{f_\theta(o_t|q,o_{<t})}{f_{\text{old}}(o_t|q,o_{<t})} A_t, \text{clip}\left( \frac{f_\theta(o_t|q,o_{<t})}{f_{\text{old}}(o_t|q,o_{<t})}, 1-\epsilon, 1+\epsilon \right) A_t \right].$$

(13)

Although widely used in alignment tasks, PPO has several limitations. Its learning process is unstable, computationally expensive, and requires extensive hyperparameter tuning. The clipped objective can slow convergence and yield suboptimal policies [6]. Notably, training the value model is challenging due to high variance and poor generalization [1]. Besides, PPO is also prone to reward hacking, struggles with long-term credit assignment, and suffers from the issue of sample inefficiency [10].

**Group Relative Policy Optimization (GRPO)** [34] simplifies PPO via removing the learnable value model. Instead, GRPO uses the average reward of multiple sampled outputs for the same question. Specifically, given a question $q$, GRPO requires to sample $G$ outputs from the old policy as $\{o_i\}_{i=1}^G \sim f_{\text{old}}(\cdot|q)$. Then, it computes the reward $r_i$ for each output $o_i$ (through deterministic reward functions) and obtains a group of rewards $\{r_i\}_{i=1}^G$. The advantage $\hat{A}_i$ of GRPO is estimated as:

$$\hat{A}_i = \tilde{r}_i = \frac{r_i - \text{mean}(\{r_i\}_{i=1}^G)}{\text{std}(\{r_i\}_{i=1}^G)}.$$

(14)

The objective of GRPO, shown below, is to maximize the advantage (the first term) while ensuring that the policy model remains close to the reference policy (the second term of KL divergence):

$$\mathcal{J}_{\text{GRPO}}(f_\theta) \triangleq \mathbb{E}_{(q,a)\sim\mathcal{D},\{o_i\}_{i=1}^G\sim f_{\text{old}}(\cdot|q)} \frac{1}{G} \sum_{i=1}^G \frac{1}{|o_i|} \sum_{t=1}^{|o_i|}$$

(15)

$$\left[ \min\left( \frac{f_\theta(o_{i,t}|q,o_{i,<t})}{f_{\text{old}}(o_{i,t}|q,o_{i,<t})} \hat{A}_i, \text{clip}\left( \frac{f_\theta(o_{i,t}|q,o_{i,<t})}{f_{\text{old}}(o_{i,t}|q,o_{i,<t})}, 1-\epsilon, 1+\epsilon \right) \hat{A}_i \right) - \beta \mathbb{D}_{\text{KL}}\left[ f_\theta || f_{\text{ref}} \right] \right].$$

Here, the $\text{clip}(\cdot, 1-\epsilon, 1+\epsilon)$ ensures that updates do not deviate excessively from the old policy by bounding the policy ratio between $1-\epsilon$ and $1+\epsilon$. Besides, the KL divergence is estimated as:

$$\mathbb{D}_{\text{KL}}\left[ f_\theta || f_{\text{ref}} \right] = \frac{f_{\text{ref}}(o_{i,t}|q,o_{i,<t})}{f_\theta(o_{i,t}|q,o_{i,<t})} - \log \frac{f_{\text{ref}}(o_{i,t}|q,o_{i,<t})}{f_\theta(o_{i,t}|q,o_{i,<t})} - 1.$$

(16)

Nonetheless, GRPO can be challenging to implement because it sometimes produces outputs with unintended token distributions or incoherent language patterns [8]. It also demands careful reward function design to balance fairness constraints and meaningful group-based advantage estimation [35]. The RL training process in GRPO can be unstable due to its reliance on group-based relative advantages, and it remains computationally expensive, especially for large-scale implementations [33, 28]. Furthermore, while GRPO introduces optimizations to post-training, it does not consistently outperform simpler methods like SFT, particularly in small-scale training or with smaller models. This highlights the trade-offs between complexity, computational cost, interpretability, and practical effectiveness [28].

In addition, several RL algorithms have been developed primarily for alignment tasks. Therein, DPO [29], CPO [41], and their variants [19, 9, 27, 12, 39] rely on pairs of outputs labeled by human preference. In contrast, KTO [7] and BCO [16] require only a single binary label (like or dislike) for each output. Besides, the PRM [36, 20] and Step-KTO [21] offer step-by-step guidance by incorporating feedback at each reasoning step rather than focusing solely on the final outputs. Recently, the follow-up work of GRPO improves the optimization objective, *e.g.*, DAPO [45], Dr. GRPO [23], REINFOECE++ [13], CPPO [22], and GPG [3]. Another line of research generalizes GRPO to broader applications such as multimodal reasoning [48, 14, 3, 25, 47] and logical reasoning [40].

We present CoDaPO algorithm in Alg. 1 and provide Tab. 7 to compare the fine-grained components in CoDaPO and relevant RL-based algorithms.

---

**Algorithm 1 CoDaPO**: **Co**nfidence and **D**ifficulty-**A**daptive **P**olicy **O**ptimization

---

**Input:** Initial policy model $\pi_\theta$, training set $\mathcal{D}$, group size $G$

1: **for** step $= 1, \cdots, M$ **do**
2:     Sample a batch $\mathcal{D}_b$ from $\mathcal{D}$
3:     Update the old policy model: $\pi_{\theta_{\text{old}}} \leftarrow \pi_\theta$
4:     // Group-based Rollout Sampling
5:     **for** $i = 1, \cdots, G$ **do**
6:         Sample response $o_i \sim \pi_{\theta_{\text{old}}}(\cdot \mid q)$ for each training sample $q \in \mathcal{D}_b$
7:     **end for**
8:     Compute rewards $\{r_i\}_{i=1}^G$ for each sampled response $o_i$
9:     // Confidence and difficulty-adaptive reweighting
10:     **for** every training sample $q$'s response group $\{o_i\}_{i=1}^G$ **do**
11:         Compute confidence reweighting factor $\{\hat{c}_i\}_{i=1}^G$ via Equation 11
12:         Compute difficulty reweighting factor $\hat{d}_q$ via Equation 11
13:         For each $o_i$, compute $\hat{A}_{i,t}$ for the $t$-th token of $o_i$
14:         Apply reweighting to the advantage: $\hat{A}_{i,t} \leftarrow \hat{A}_{i,t} \cdot \hat{c}_i \cdot \hat{d}_q$
15:     **end for**
16:     // Update policy
17:     **for** iteration $= 1, \cdots, \mu$ **do**
18:         Update the policy model $\pi_\theta$ by maximizing the CoDaPO objective via Equation 8
19:     **end for**
20: **end for**

**Output:** $\pi_\theta$

---

| Normalization | Advantage | Regularization | Note |
|---|---|---|---|
| | **GRPO [34]** | | |
| $\mathbb{E}_{(q,a)\sim\mathcal{D},\{o_i\}_{i=1}^G\sim f_{\text{old}}(\cdot\mid q)}$ $\frac{1}{G}\sum_{i=1}^G\frac{1}{\lvert o_i\rvert}\sum_{t=1}^{\lvert o_i\rvert}\Big[$ | $\min\Big(\frac{f_{\boldsymbol\theta}(o_{i,t}\mid q,o_{i,<t})}{f_{\text{old}}(o_{i,t}\mid q,o_{i,<t})}\hat{A}_i,$ $\text{clip}\big(\frac{f_{\boldsymbol\theta}(o_{i,t}\mid q,o_{i,<t})}{f_{\text{old}}(o_{i,t}\mid q,o_{i,<t})},1-\epsilon,1+\epsilon\hat{A}_i\big)$ | $-\beta\Big(\frac{f_{\text{ref}}(o_{i,t}\mid q,o_{i,<t})}{f_{\boldsymbol\theta}(o_{i,t}\mid q,o_{i,<t})}$ $-\log\frac{f_{\text{ref}}(o_{i,t}\mid q,o_{i,<t})}{f_{\boldsymbol\theta}(o_{i,t}\mid q,o_{i,<t})}-1\Big)\Big].$ | $\hat{A}_i=\frac{r_i-\text{mean}(\{r_i\}_{i=1}^G)}{\text{std}(\{r_i\}_{i=1}^G)}.$ |
| | **Dr. GRPO [23]** | | |
| $\mathbb{E}_{(q,a)\sim\mathcal{D},\{o_i\}_{i=1}^G\sim f_{\text{old}}(\cdot\mid q)}$ $\frac{1}{G\cdot c}\sum_{i=1}^G\sum_{t=1}^{\lvert o_i\rvert}\Big[$ | $\min\Big(\frac{f_{\boldsymbol\theta}(o_{i,t}\mid q,o_{i,<t})}{f_{\text{old}}(o_{i,t}\mid q,o_{i,<t})}\hat{A}_i,$ $\text{clip}\big(\frac{f_{\boldsymbol\theta}(o_{i,t}\mid q,o_{i,<t})}{f_{\text{old}}(o_{i,t}\mid q,o_{i,<t})},1-\epsilon,1+\epsilon\big)\hat{A}_i\big)$ | $-\beta\Big(\frac{f_{\text{ref}}(o_{i,t}\mid q,o_{i,<t})}{f_{\boldsymbol\theta}(o_{i,t}\mid q,o_{i,<t})}$ $-\log\frac{f_{\text{ref}}(o_{i,t}\mid q,o_{i,<t})}{f_{\boldsymbol\theta}(o_{i,t}\mid q,o_{i,<t})}-1\Big)\Big].$ | $\hat{A}_i=r_i-\text{mean}(\{r_i\}_{i=1}^G).$ |
| | **DAPO [45]** | | |
| $\mathbb{E}_{(q,a)\sim\mathcal{D},\{o_i\}_{i=1}^G\sim f_{\text{old}}(\cdot\mid q)}$ $\frac{1}{\sum_{i=1}^G\lvert o_i\rvert}\sum_{i=1}^G\sum_{t=1}^{\lvert o_i\rvert}\Big[$ | $\min\Big(\frac{f_{\boldsymbol\theta}(o_{i,t}\mid q,o_{i,<t})}{f_{\text{old}}(o_{i,t}\mid q,o_{i,<t})}\hat{A}_i,$ $\text{clip}\big(\frac{f_{\boldsymbol\theta}(o_{i,t}\mid q,o_{i,<t})}{f_{\text{old}}(o_{i,t}\mid q,o_{i,<t})},1-\epsilon_{\text{low}},1+\epsilon_{\text{high}}\big)\hat{A}_i\big)\Big].$ | **None** | $\hat{A}_i=\frac{r_i-\text{mean}(\{r_i\}_{i=1}^G)}{\text{std}(\{r_i\}_{i=1}^G)}.$ |
| | **GPG [3]** | | |
| $\mathbb{E}_{(q,a)\sim\mathcal{D},\{o_i\}_{i=1}^G\sim f_{\text{old}}(\cdot\mid q)}$ $\frac{1}{\sum_{i=1}^G\lvert o_i\rvert}\sum_{i=1}^G\sum_{t=1}^{\lvert o_i\rvert}\Big[$ | $\log f_\theta(o_{i,t}\mid q,o_{i,<t})\hat{A}_i\Big].$ | **None** | $\hat{A}_i=\alpha\cdot\big(r_i-\text{mean}(\{r_i\}_{i=1}^G)\big).$ |
| | **CoDaPO (ours)** | | |
| $\mathbb{E}_{(q,a)\sim\mathcal{D},\{o_i\}_{i=1}^G\sim f_{\text{old}}(\cdot\mid q)}$ $\frac{1}{\sum_{i=1}^G\lvert o_i\rvert}\sum_{i=1}^G\sum_{t=1}^{\lvert o_i\rvert}\Big[$ | $\min\Big(\frac{f_{\boldsymbol\theta}(o_{i,t}\mid q,o_{i,<t})}{f_{\text{old}}(o_{i,t}\mid q,o_{i,<t})}\hat{A}_i\cdot\hat{c}_i\hat{d}_q,$ $\text{clip}\big(\frac{f_{\boldsymbol\theta}(o_{i,t}\mid q,o_{i,<t})}{f_{\text{old}}(o_{i,t}\mid q,o_{i,<t})},1-\epsilon,1+\epsilon\big)\hat{A}_i\cdot\hat{c}_i\hat{d}_q\big)\Big].$ | **None** | $\hat{A}_i=\frac{r_i-\text{mean}(\{r_i\}_{i=1}^G)}{\text{std}(\{r_i\}_{i=1}^G)},\hat{c}_i=\sigma\Big(\frac{1}{G}\sum_{i=1}^G\frac{1}{\lvert o_i\rvert}\sum_{t=1}^{\lvert o_i\rvert}\log f_\theta(o_{i,t}\mid q,o_{i,<t})$ $-\frac{1}{\lvert o_i\rvert}\sum_{t=1}^{\lvert o_i\rvert}\log f_\theta(o_{i,t}\mid q,o_{i,<t})\Big),$ $\hat{d}_q=1+\delta(\frac{1}{G}\sum_{i=1}^G r_i-\frac{1}{2})^2.$ |

Table 7: Comparing the components in different RL algorithms for reasoning. Note that the **highlighted** contents indicate the differences with the original implementation of the GRPO algorithm. Specifically, (1) the advantage term evaluates the quality of model-generated outputs, (2) the regularization term measures the divergence between the policy model and a frozen reference model, and (3) the normalization term scales the final optimization signal across multiple problems and outputs.

## G    IMPLEMENTATION DETAILS

**Benchmarks.** We evaluate our proposed method and other baselines on the following diverse benchmarks.

1. **MATH 500 [11].** A curated set of 500 challenging problems from the MATH dataset, focusing on high school-level mathematics across algebra, geometry, number theory, and combinatorics.

2. **AIME 2024** [2]**.** A benchmark based on the 2024 American Invitational Mathematics Examination, testing advanced problem-solving skills with 15 short-answer math problems designed for top high school students.

3. **AIME 2025** [3]**.** The 2025 version of the AIME benchmark is similarly structured, providing a fresh set of high-difficulty pre-Olympiad level math problems.

4. **AMC 2023** [4]**.** Based on the 2023 American Mathematics Competitions (AMC 10/12), this benchmark assesses middle-to-advanced high school math across a range of topics in a multiple-choice format.

5. **Olympiad Benchmark [15].** A collection of problems from various math olympiads (*e.g.*, USAMO, IMO), aimed at evaluating models on deep mathematical reasoning and multi-step proofs.

6. **Minerva [18].** A benchmark and model suite by Google DeepMind that tackles math and science questions (from grade school to graduate level) using CoT reasoning and LLMs.

7. **GSM8K [4].** A dataset of 8,500 grade-school level math word problems designed to test models' ability to perform multi-step numerical reasoning in natural language.

**Experiment framework.** In this work, we utilize Transformer Reinforcement Learning (TRL) as the training backbone, specifically version 0.16.1. To ensure consistency with the vanilla GRPO algorithm, we revise the original GRPO trainer in TRL accordingly since it has some discrepancies in loss computation from Shao et al. [34]. Building upon this implementation, we develop several baseline methods evaluated in this work, including DAPO, Dr.GRPO, GPG, and our proposed approach, CoDaPO. The evaluation adapts Qwen2.5-Math's evaluation codebase[5], ensuring consistent and reliable measurement across all experiments.

**General hyperparameters.** To ensure fair comparisons, we train all algorithms using the same set of hyperparameters.

- **Sampling setting.** For each question, we sample 12 responses to form a response group. The sampling temperature is set to 0.9, and we use a top-p value of 1.0 to consider the full token distribution.

- **Learning rate.** We use a learning rate of $3.0e - 6$ and apply a warm-up over the first $10\%$ of global training steps. A cosine learning rate scheduler is employed to gradually reduce the learning rate to zero throughout training.

- **Batch size.** We set the batch size to 6 to evenly distribute the generated responses across training devices. To mitigate the variance introduced by different question samples, we use a gradient accumulation step of 12.

- **Randomness control.** To ensure reproducibility, we set the random seed to 42 and enable full determinism.

**Prompt template.** We use the prompt template shown in Fig. 6 for both training and evaluation. Furthermore, we apply chat template shown in Fig. 7 when processing the raw data.

**Reward setting.** We observe that although using a format reward can improve the readability of LLM outputs to some extent, it may lead to reward hacking in the later stages of training, potentially resulting in irreversible training collapse. This observation is consistent with the findings of [46].

---

[2]`https://huggingface.co/datasets/HuggingFaceH4/aime_2024`
[3]`https://huggingface.co/datasets/opencompass/AIME2025`
[4]`https://huggingface.co/datasets/math-ai/amc23`
[5]`https://github.com/QwenLM/Qwen2.5-Math/tree/main/evaluation`

---

**Prompt Template**

```
    Please reason step by step, and put your final answer within
                          \boxed{}.
```

Figure 6: Prompt template used during training and evaluation.

---

**Chat Template**

```
<|im_start|>system
Please reason step by step, and put your final answer within
\boxed{}.
<|im_end|>
<|im_start|>user
Find the sum of all integer bases $b>9$ for which $17_b$ is a
divisor of $97_b$.
<|im_end|>
<|im_start|>assistant
```

Figure 7: Example of a prompt used after applying the chat template.

Therefore, in all experiments conducted in this work, we rely solely on the accuracy reward, defined as follows:

$$r_i(y_i, \hat{y}) = \begin{cases} 1, & \text{is\_equivalent}(y_i, \hat{y}) \\ 0, & \text{otherwise} \end{cases}.$$

Here, $y_i$ denotes the answer produced for the $i$-th output and $\hat{y}$ represents the corresponding ground truth. To accurately determine whether $y_i$ and $\hat{y}$ are equivalent, we employ Math-Verify[6] as our reward evaluation system, where all LaTeX expressions are parsed and compared for mathematical equivalence.

**Completion length constraint.** Following the work of [46], which argues that overly lenient length constraints—or any heuristics that encourage the model to produce excessively long chains of thought—can, in many cases, lead to overthinking, we impose a strict maximum completion length of 1024 tokens during both training and evaluation. While this constraint may appear intuitively unreasonable, our experimental results demonstrate that reinforcement learning post-training under such a short-CoT constraint can still yield strong performance (see Fig. 10).

## H FULL EXPERIMENTS

**Difficulty estimation.** Accurate estimation of question difficulty plays a crucial role in our proposed algorithm, CoDaPO, as it directly influences the computation of difficulty-adaptive weights in the optimization objective. However, relying solely on pre-existing difficulty annotations presents significant limitations. First, not all datasets contain ground-truth difficulty labels. Second, since the model's performance evolves during training, the perceived difficulty of a question may vary over time. Consequently, fixed difficulty labels may fail to reflect the dynamic nature of the model's learning process.

To explore a more adaptive and robust difficulty estimation approach, we conduct preliminary experiments on the MATH dataset, which includes human-annotated difficulty levels. Using GRPO on Qwen2.5-Math-1.5B, we analyze the accuracy trajectories for different difficulty levels throughout training. As shown in Fig. 8, model accuracy aligns well with the ground-truth difficulty labels: easier questions correspond to higher accuracy, and harder questions to lower accuracy. Moreover, the accuracy gap between different difficulty levels increases and stabilizes as training progresses, indicating a consistent difficulty signal.

---

[6]https://github.com/huggingface/Math-Verify

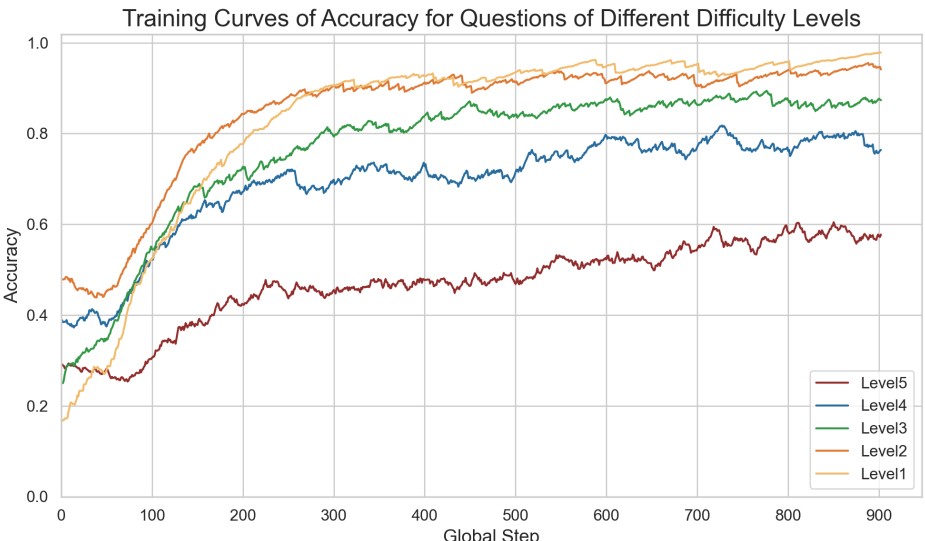

Figure 8: The GRPO training accuracy curves for questions of different difficulty levels on Qwen2.5-Math-1.5B.

These observations motivate us to estimate question difficulty based on the model's own accuracy. This approach is naturally integrated into the online RL process without requiring any additional evaluation model, as multiple samples per question are generated during training. It generalizes well to datasets without difficulty annotations and provides a dynamic estimation mechanism that adapts to the model's evolving capabilities over time, overcoming the limitations of static difficulty labels.

**Analysis details.** We show the quantitative details of Fig. 2 and Fig. 3 in Tab. 8 and Tab. 9, respectively. For each difficulty level, we fix a set of 100 validation questions and sample 20 trajectories per question from the model at each checkpoint. This yields exactly 2000 points per subplot. We repeat this procedure for the base model, for step 60, and for step 240 using the same questions and sampling protocol. Thus every subplot at every checkpoint is based on the full (and identical) validation subset, with 2000 points each. During GRPO training, model confidence progressively saturates toward 1, with both correct and incorrect responses clustering near high-confidence values, leading to miscalibration. Confidence differences across difficulty levels diminish over time, while tail behavior (kurtosis) reveals structural changes in the distribution. Gradients initially exhibit heavy-tailed bursts for easy questions, but training rapidly suppresses magnitudes and contracts distributions across all difficulty levels. By the end of training, gradient contraction becomes more uniform, though hard questions retain residual difficulty. Overall, GRPO induces saturation in confidence and global contraction in gradients, progressively reducing differences across difficulty levels.

| Confidence | Step 0 | | | Step 60 | | | Step 240 | | |
|---|---|---|---|---|---|---|---|---|---|
| | Level 1 | Level 3 | Level 5 | Level 1 | Level 3 | Level 5 | Level 1 | Level 3 | Level 5 |
| Minimum | 0.00 | 0.00 | 0.00 | 0.01 | 0.00 | 0.00 | 0.32 | 0.54 | 0.36 |
| Maximum | 0.99 | 0.99 | 0.99 | 0.99 | 0.98 | 0.99 | 0.98 | 0.97 | 0.98 |
| Mean | 0.78 | 0.75 | 0.69 | 0.82 | 0.81 | 0.76 | 0.86 | 0.87 | 0.86 |
| Std | 0.15 | 0.18 | 0.21 | 0.10 | 0.14 | 0.17 | 0.06 | 0.05 | 0.06 |
| Median | 0.83 | 0.81 | 0.75 | 0.84 | 0.84 | 0.81 | 0.87 | 0.88 | 0.87 |
| Kurtosis | 6.69 | 2.53 | 0.60 | 16.10 | 8.41 | 3.32 | 10.02 | 2.67 | 5.47 |

Table 8: Confidence statistics at different steps and difficulty levels.

**Completion length.** In Figs. 9 and 10, we illustrate how the length of model responses evolves throughout the training process. Contrary to prior studies that advocate for encouraging longer responses as a means of enhancing model performance, our observations reveal a different perspective:

| Gradient | Step 0 | | | Step 60 | | | Step 240 | | |
|---|---|---|---|---|---|---|---|---|---|
| | Level 1 | Level 3 | Level 5 | Level 1 | Level 3 | Level 5 | Level 1 | Level 3 | Level 5 |
| Minimum | 0.49 | 0.75 | 0.56 | 0.66 | 0.63 | 0.49 | 0.65 | 0.74 | 0.48 |
| Maximum | 1723.08 | 954.45 | 213.61 | 193.70 | 425.39 | 875.58 | 21.10 | 19.25 | 11.85 |
| Mean | 5.26 | 5.30 | 5.60 | 2.84 | 3.42 | 5.06 | 2.26 | 1.95 | 2.19 |
| Standard deviation | 42.90 | 26.05 | 11.51 | 6.78 | 12.75 | 25.66 | 1.14 | 0.89 | 0.90 |
| Median | 2.40 | 2.57 | 2.95 | 2.08 | 2.03 | 2.57 | 2.01 | 1.81 | 2.08 |
| Kurtosis | 1325.53 | 988.33 | 95.16 | 434.23 | 666.90 | 795.30 | 46.17 | 102.37 | 17.61 |

Table 9: Gradient statistics at different steps and difficulty levels.

namely, that even under stringent response length constraints (*e.g.*, 1024 tokens), models can still achieve substantial performance gains. In fact, although we explicitly impose a hard stop when a model's response exceeds 1024 tokens, the model never actually reaches this limit during training.

Interestingly, performance improvements are often accompanied by a decrease in output length. We observe a similar trend in vanilla GRPO as well. We hypothesize that this phenomenon stems from the nature of the math questions, which typically do not require excessively long reasoning chains. Instead, longer responses from the base model often reflect overthinking, including the generation of unnecessary reasoning steps or placeholder verification logic (*e.g.*, writing pseudo Python code to "verify" results without external execution support).

RL mitigates these tendencies by discouraging redundant generation and encouraging the model to produce concise, solution-oriented answers. As a result, we observe a consistent downward trend in average completion length, which aligns with improved performance.

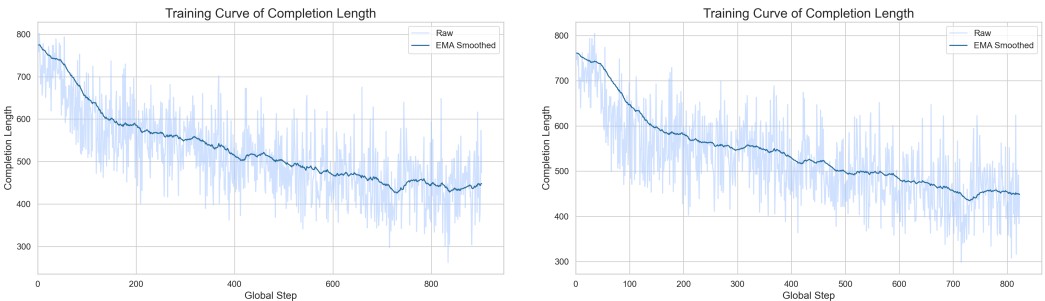

Figure 9: The GRPO training curves for completion length on Qwen2.5-Math-1.5B.

Figure 10: The CoDaPO training curves for completion length on Qwen2.5-Math-1.5B.

**Reward curves.** We present the reward curve and its standard deviation throughout the training process in Figs. 11 and 12, where Qwen2.5-Math-1.5B is trained on the MATH dataset using CoDaPO. Note that since we adopt accuracy as the sole reward, the reward values are equivalent to the model's average accuracy. Overall, CoDaPO consistently improves model performance, exhibiting a stable upward trend in accuracy across training steps.

Despite removing the KL regularization term, the training process remains stable, and we do not observe any catastrophic degradation in accuracy—an issue sometimes encountered in RL-based training. Notably, the most significant performance gain occurs within the first 200 steps, which corresponds to our designated warm-up phase, highlighting the early-stage effectiveness of CoDaPO in guiding the model toward better reasoning behaviors.

**Confidence curves.** We present the average confidence curve of model responses, along with separate confidence curves for correct and incorrect responses, in Fig. 13. To compute the confidence for each output, we first calculate the log probability of each token in the response and take the average at the token level. Given the unbounded nature of log probabilities, we exponentiate the average log probability to obtain a normalized confidence score for each response.

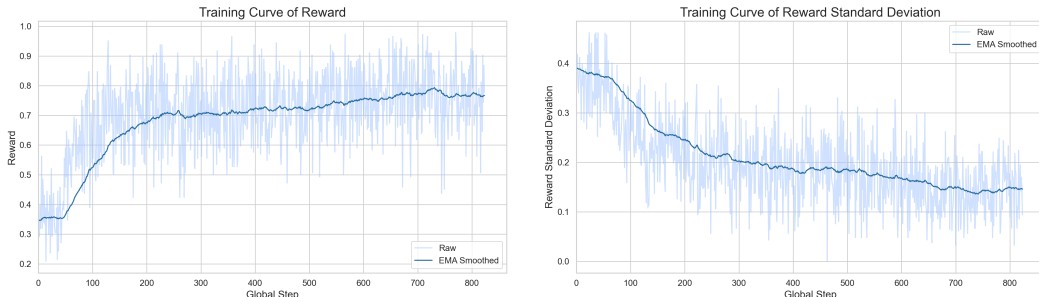

Figure 11: The CoDaPO training curves for reward on Qwen2.5-Math-1.5B.

Figure 12: The CoDaPO training curve for reward standard deviation on Qwen2.5-Math-1.5B.

From the confidence curves, we observe a clear and consistent gap between correct and incorrect responses throughout the CoDaPO training process: the model assigns significantly higher confidence to correct outputs compared to incorrect ones. This demonstrates that CoDaPO not only improves accuracy, but also enhances the model's ability to calibrate its confidence in a more meaningful and discriminative manner.

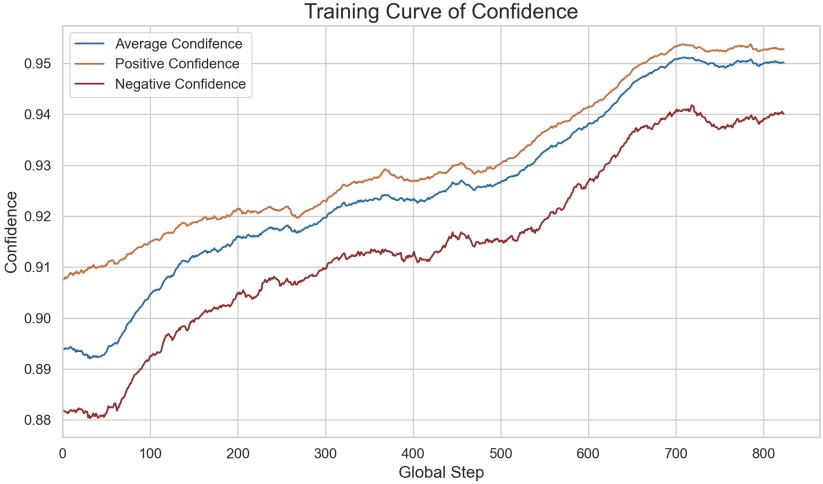

Figure 13: The CoDaPO training curves for response confidence on Qwen2.5-Math-1.5B.

**Extension to larger models.** We apply CoDaPO to Qwen2.5-Math-7B, and the results are shown in Tab. 2. CoDaPO achieves $78.41\%$ accuracy on MATH500, outperforming both the base model ($47.24\%$) and GRPO ($74.62\%$), and achieves the best average performance of $49.34\%$, demonstrating its robustness and consistent effectiveness across different base models.

**Discarding KL regularization.** In CoDaPO, we remove the KL regularization term to encourage the model to explore more freely, without being overly constrained by the base model's output distribution. Further, to evaluate whether the removal of the KL term negatively affects the linguistic quality of the model outputs, we perform additional ablation studies where we reapply the KL regularization. The evaluation results on the MATH500 dataset confirm that removing KL improves reasoning accuracy. To assess general language modeling ability, we also evaluate the model's perplexity on a generic corpus, wikitext-2 [7]. The results are summarized in Tab. 10 and show that removing KL does not lead to any notable increase in perplexity, suggesting that the model's linguistic fluency remains intact. Additionally, we evaluate the impact of using or not using KL on the model's confidence

---

[7] https://huggingface.co/datasets/mikasenghaas/wikitext-2

and accuracy across different difficulty levels, as shown in Table 11. We find that including the KL term reduces overall confidence for both correct and incorrect outputs, which is consistent with our analysis in Section 3. However, this reduction in confidence comes at a cost: by constraining the model probabilities, the KL term also limits the upward adjustment on correct outputs, thereby restricting the model's ability to fully leverage positive reward signals.

| Models | Qwen2.5-Math-1.5B | CoDaPO w/ KL | CoDaPO wo/ KL |
|---|---|---|---|
| Accuracy on MATH500 | 31.20 | 69.20 | 71.68 |
| Perplexity on wikitext-2 | 35.62 | 36.10 | 36.40 |

Table 10: Results of ablation study on KL regularization.

| Level | Split | w/o KL | | | w/ KL | | | Difference (w/o-w/) | |
|---|---|---|---|---|---|---|---|---|---|
| | | Count | Mean | Std | Count | Mean | Std | Confidence | Accuracy |
| 1 | Correct | 1958 | 0.9093 | 0.0398 | 1914 | 0.8564 | 0.0866 | +0.0529 (+5.82%) | |
| 1 | Incorrect | 42 | 0.9026 | 0.0517 | 86 | 0.7714 | 0.1938 | +0.1312 (+14.53%) | +0.0220 (+2.30%) |
| 1 | All | 2000 | 0.9091 | 0.0401 | 2000 | 0.8527 | 0.0953 | +0.0564 (+6.20%) | |
| 2 | Correct | 1903 | 0.9137 | 0.0357 | 1842 | 0.8682 | 0.0750 | +0.0454 (+4.97%) | |
| 2 | Incorrect | 97 | 0.8952 | 0.0460 | 158 | 0.8167 | 0.1343 | +0.0785 (+8.77%) | +0.0305 (+3.31%) |
| 2 | All | 2000 | 0.9128 | 0.0365 | 2000 | 0.8642 | 0.0824 | +0.0486 (+5.33%) | |
| 3 | Correct | 1839 | 0.9135 | 0.0377 | 1799 | 0.8675 | 0.0809 | +0.0460 (+5.04%) | |
| 3 | Incorrect | 161 | 0.9011 | 0.0465 | 201 | 0.8170 | 0.1360 | +0.0840 (+9.33%) | +0.0200 (+2.22%) |
| 3 | All | 2000 | 0.9125 | 0.0386 | 2000 | 0.8624 | 0.0893 | +0.0501 (+5.49%) | |
| 4 | Correct | 1743 | 0.9090 | 0.0458 | 1669 | 0.8597 | 0.0903 | +0.0494 (+5.43%) | |
| 4 | Incorrect | 257 | 0.8813 | 0.0614 | 331 | 0.8310 | 0.1039 | +0.0503 (+5.71%) | +0.0370 (+4.43%) |
| 4 | All | 2000 | 0.9055 | 0.0489 | 2000 | 0.8549 | 0.0933 | +0.0506 (+5.58%) | |
| 5 | Correct | 1550 | 0.9079 | 0.0403 | 1498 | 0.8604 | 0.0939 | +0.0474 (+5.23%) | |
| 5 | Incorrect | 450 | 0.8786 | 0.0514 | 502 | 0.8044 | 0.1317 | +0.0742 (+8.45%) | +0.0260 (+3.47%) |
| 5 | All | 2000 | 0.9013 | 0.0447 | 2000 | 0.8464 | 0.1075 | +0.0549 (+6.09%) | |

Table 11: Comparison of confidence statistics with and without KL across difficulty levels.

**Confidence dynamic.** Due to the large number of samples, overlapping points in the confidence-advantage distribution may have obscured fine-grained details. To address this, we provide more detailed level-wise statistics demonstrating that CoDaPO mitigates the overconfidence issue. The comparison level-wise probability is shown in Tab. 12. These results show that CoDaPO consistently reduces the model's confidence across all difficulty levels, thereby alleviating overconfidence. This is particularly important in the RL setting, where excessive confidence leads to higher sampling probability, which in turn reduces intra-group variance, ultimately causing group-wise advantage collapse. In addition, we observe that CoDaPO induces greater reductions in confidence for incorrect responses on harder problems, e.g., for Level 5, the log probability of incorrect responses decreased by a much larger $17.01\%$. These observations indicate that CoDaPO produces more conservative (i.e., lower-confidence) predictions than GRPO, with the effect being particularly pronounced for incorrect responses at higher difficulty levels.

| Level | Prob (GRPO) | Prob (CoDaPO) | Difference in Prob (%) | Log Prob (GRPO) | Log Prob (CoDaPO) | Difference in Log Prob (%) |
|---|---|---|---|---|---|---|
| Overall | 0.892243 | 0.878914 | -1.49% | -0.114878 | -0.130072 | -13.23% |
| Level 1 | 0.888831 | 0.876010 | -1.44% | -0.118636 | -0.133318 | -12.38% |
| Level 2 | 0.897471 | 0.884609 | -1.43% | -0.108690 | -0.123172 | -13.32% |
| Level 3 | 0.898096 | 0.884135 | -1.55% | -0.108254 | -0.123972 | -14.52% |
| Level 4 | 0.892173 | 0.878437 | -1.54% | -0.115166 | -0.130887 | -13.65% |
| Level 5 | 0.884646 | 0.871380 | -1.50% | -0.123643 | -0.139012 | -12.43% |

Table 12: The comparison level-wise probability between CoDaPo and GRPO.

**Reweighting strategies.** To compare different difficulty reweighting strategies, such as upweighting difficult samples, we adopt the difficulty term $1 - \hat{r}$, and train the model under the same setting. The results are shown in the table below. We observe that simply upweighting difficult samples performs worse than our U-shaped difficulty design. This is likely because focusing solely on difficult samples produces highly unbalanced gradient signals. Hard problems typically yield very sparse positive advantages (correct responses are rare), causing the model to overfit these few signals while largely ignoring errors on easy problems. As a result, the model fails to correct mistakes on easy items, which constitute a substantial portion of the training distribution and are crucial for stable improvement.

**Hyperparameter sensitivity.** In our work, $\delta$ is an integral part of our difficulty-based reweighting mechanism, which controls the degree of reward or penalty applied to problems of varying difficulty during training. To demonstrate the effectiveness and robustness of this design, we conducted a series of experiments under different $\delta$. The results in Tab. 13 show that CoDaPO consistently benefits from a range of settings, indicating that our method is not sensitive to the selection of this hyperparameter.

| $\delta$ | 0 | 0.2 | 0.4 | 0.6 | 0.8 |
|---|---|---|---|---|---|
| Accuracy on MATH500 | 71.52 | 72.00 | 71.68 | 71.60 | 71.80 |

Table 13: CoDaPO's Performance under different $\delta$.

**Decoding Strategies.** To explore the impact of evaluation settings on performance, we evaluate the models with different decoding settings. As shown in Tab. 15, different sampling temperatures do affect absolute performance. However, these effects are mostly relative: the relative ranking between methods remains stable. Across all decoding temperatures, CoDaPO consistently outperforms GRPO, demonstrating the robustness and reliability of the improvements brought by our method.

**Seed Robustness.** We report results across multiple random seeds, as shown in Tab. 14. We observe that CoDaPO consistently outperforms other baselines, and across the 7 benchmarks, it achieves the best performance on at least 6 of them for each seed, indicating that the improvements are robust rather than due to chance.

| Seed | Model | MATH 500 | AIME 2024 | AIME 2025 | AMC 2023 | Olympiad Bench | Minerva | GSM8K | Average |
|---|---|---|---|---|---|---|---|---|---|
| 42 | Qwen2.5-Math-1.5B | 31.20 | 6.00 | 0.00 | 29.50 | 18.25 | 9.63 | 34.01 | 18.37 |
| 0 | GRPO | 69.11 | 11.87 | 6.67 | 51.25 | 31.55 | 29.27 | 80.03 | 39.96 |
| 0 | DAPO | 71.91 | 13.13 | 13.33 | 50.00 | 32.95 | 31.59 | 82.92 | 42.26 |
| 0 | CoDaPO | **72.21** | **14.58** | 10.83 | **52.19** | **33.58** | **31.99** | **83.39** | **42.68** |
| 42 | GRPO | 68.48 | 12.67 | 8.00 | 51.00 | 30.55 | 28.53 | 79.56 | 39.82 |
| 42 | DAPO | 71.12 | 14.00 | 8.00 | 49.00 | 31.97 | 32.13 | 82.38 | 41.23 |
| 42 | CoDaPO | **71.68** | **14.67** | **10.83** | **53.50** | **32.80** | 31.69 | **83.50** | **42.67** |
| 2025 | GRPO | 68.97 | 12.08 | 7.08 | 52.81 | 31.37 | 29.07 | 80.38 | 40.25 |
| 2025 | DAPO | 71.50 | 12.50 | 11.25 | 52.50 | 33.99 | 31.60 | 82.81 | 42.31 |
| 2025 | CoDaPO | **72.06** | **15.21** | **12.50** | **52.50** | 33.88 | **31.71** | **83.46** | **43.05** |
| 3373 | GRPO | 69.17 | 11.25 | 4.58 | 50.00 | 32.06 | 29.32 | 79.92 | 39.47 |
| 3373 | DAPO | 71.80 | 12.50 | 11.25 | 52.66 | 33.21 | 31.11 | 83.06 | 42.22 |
| 3373 | CoDaPO | **72.86** | **15.00** | **12.08** | 52.03 | **33.77** | **31.62** | **83.37** | **42.96** |

Table 14: Performance of Qwen2.5-Math-1.5B and different CoDaPO variants across multiple seeds and benchmarks. Maximum values for each seed are highlighted in bold.

| Temp. | 0 | 0.2 | 0.4 | 0.6 | 0.8 | 1.0 |
|---|---|---|---|---|---|---|
| Base Model | 27.80 | 30.00 | 31.00 | 31.20 | 30.20 | 23.40 |
| GRPO | 69.20 | 69.40 | 68.40 | 68.48 | 67.80 | 64.20 |
| CoDaPO | 69.80 | 72.40 | 70.60 | 71.68 | 69.60 | 68.60 |

Table 15: Qwen2.5-Math-1.5B performance on the MATH500 under different sampling temperatures.

**Test-time scaling.** We present the test-time scaling results in Fig. 14 and 15. Specifically, we report the pass@k accuracy on AIME25 before and after CoDaPO training, where $k \in$

$\{1, 2, 4, 8, 16, 32, 64, 128\}$. Across all values of $k$, CoDaPO consistently improves the model's performance. Notably, for Qwen2.5-Math-1.5B, the pass@8 accuracy after CoDaPO training surpasses the pass@32 accuracy of the model before training, effectively reducing the number of required responses by 24 while achieving better results. This result demonstrates that CoDaPO effectively improves the quality and reliability of model outputs under constrained sampling budgets. The fact that fewer samples are needed to achieve a given level of accuracy indicates that the model becomes more confident and consistent in producing correct answers.

Furthermore, for the instruction-tuned Qwen2.5-1.5B-Instruct model, CoDaPO also leads to substantial improvements in pass@k accuracy. The performance gain becomes more pronounced as $k$ increases. This indicates that CoDaPO also improves the diversity and calibration of the output distribution. The model is not only better at generating the correct answer in the top few samples but also more likely to include it somewhere in a wider sampling set, which further confirms the effectiveness of CoDaPO in enhancing both single-shot reliability and multi-sample robustness.

To compare against other methods, we extend the response length to 4096 tokens to further probe the model's reasoning boundaries. As shown in Tab. 16, we find that GRPO mainly improves performance at small $k$ but gradually saturates as $k$ increases, indicating that GRPO primarily sharpens high-probability outputs rather than expanding the set of correct solutions. In contrast, DAPO and especially CoDaPO consistently outperform the base model across the entire range of $k$, including the large-$k$ regime , with CoDaPO achieving the strongest and most stable gains overall.

DAPO's dynamic resampling and related mechanisms encourage broader exploration. CoDaPO complements this picture by using confidence- and difficulty-aware reweighting of advantages so that confident correct answers and rare correct solutions to hard questions receive stronger updates, while overconfident errors are penalized. The persistent pass@$k$ gains for CoDaPO across $k$ align with this design: it improves the underlying reasoning distribution rather than sharpening the top few samples.

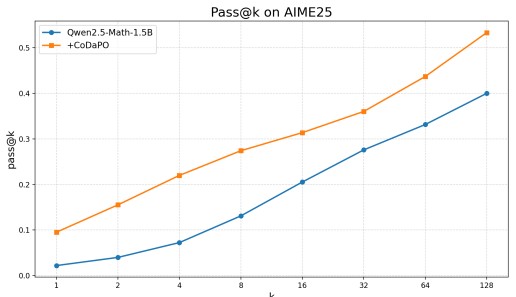 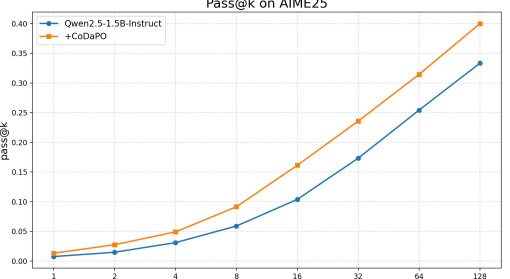

Figure 14: Pass@k result of Qwen2.5-Math-1.5B with CoDaPO on AIME25.

Figure 15: Pass@k result of Qwen2.5-1.5B-Instruct with CoDaPO on AIME25.

| K | 1 | 2 | 4 | 8 | 16 | 32 | 64 | 128 |
|---|---|---|---|---|---|---|---|---|
| Qwen2.5-Math-1.5B | 2.98 | 5.50 | 9.57 | 15.70 | 21.75 | 27.46 | 33.70 | 40.00 |
| GRPO | 6.88 | 12.85 | 19.79 | 27.56 | 35.18 | 43.17 | 46.55 | 46.67 |
| DAPO | 7.76 | 18.26 | 25.17 | 32.48 | 38.98 | 44.76 | 50.48 | 53.33 |
| CoDaPO | 11.02 | 17.80 | 24.81 | 31.56 | 38.79 | 46.15 | 53.93 | 60.00 |

Table 16: Pass@k result of Qwen2.5-Math-1.5B with different methods on AIME25.

**The full training dynamics.** In Figs. 16 and 17, we present the full confidence-advantage distribution and gradient distribution of different checkpoints during CoDaPO training on the MATH training.

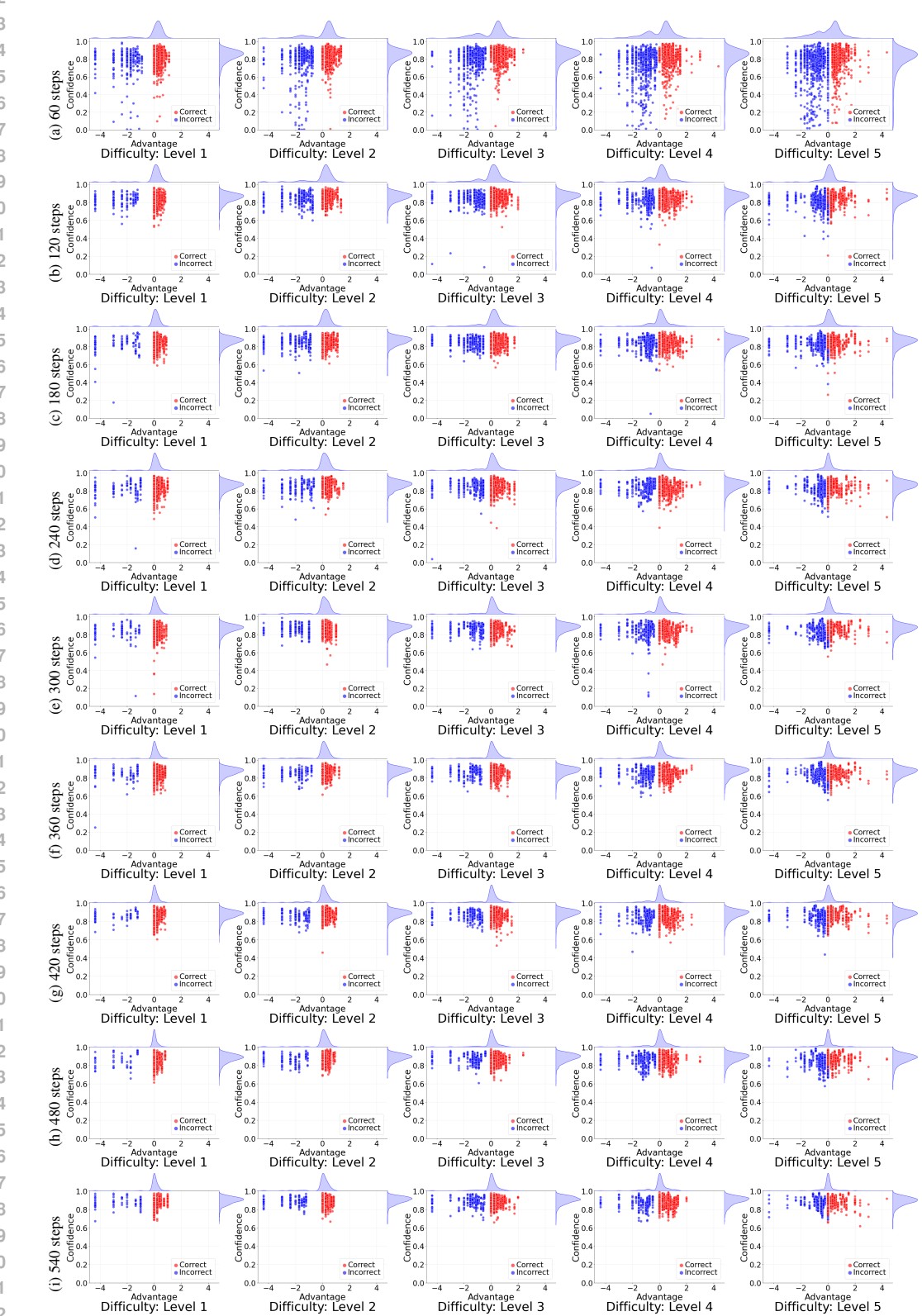

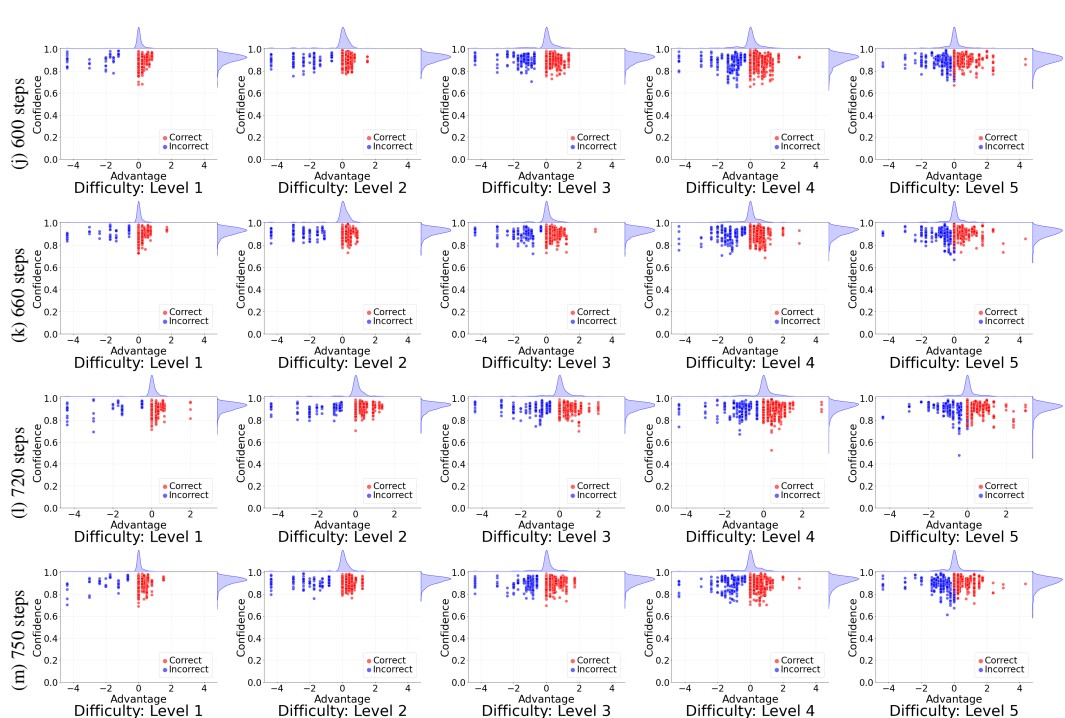

Figure 16: The confidence-advantage distribution of the Qwen2.5-Math-1.5B model, post-training on the MATH dataset with the CoDaPO algorithm. We present the distribution of model checkpoints captured at every 60 training steps. Columns 1 through 5 correspond to difficulty levels 1 through 5. Additionally, the marginal distributions (*i.e.*, the density distributions) of advantage and confidence across all samples are shown above and to the right of each confidence-advantage distribution.

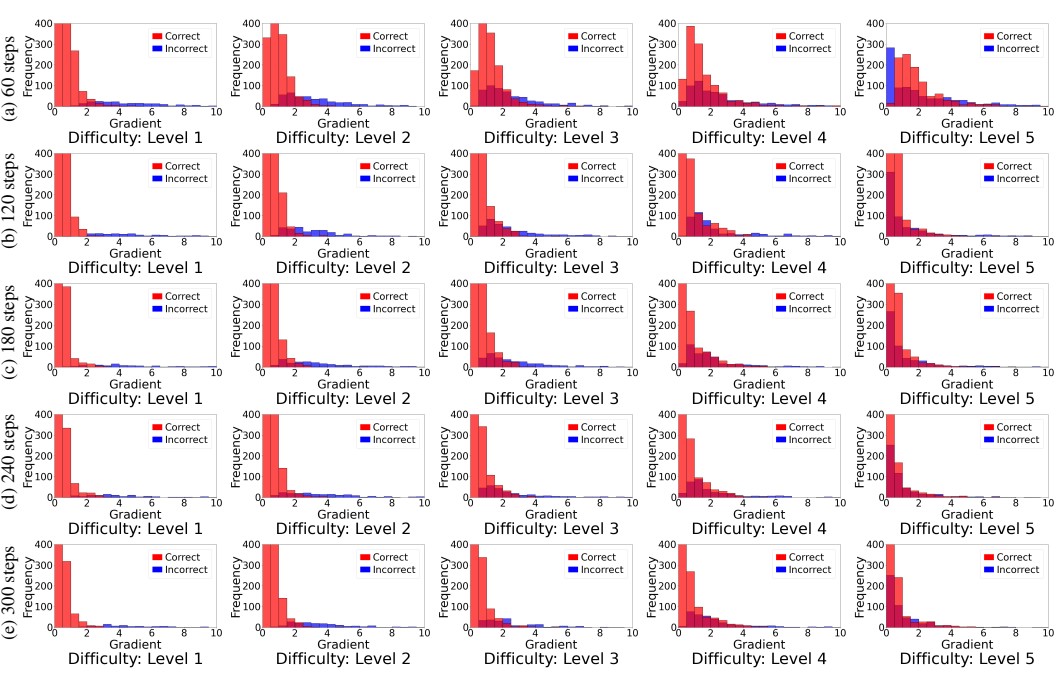

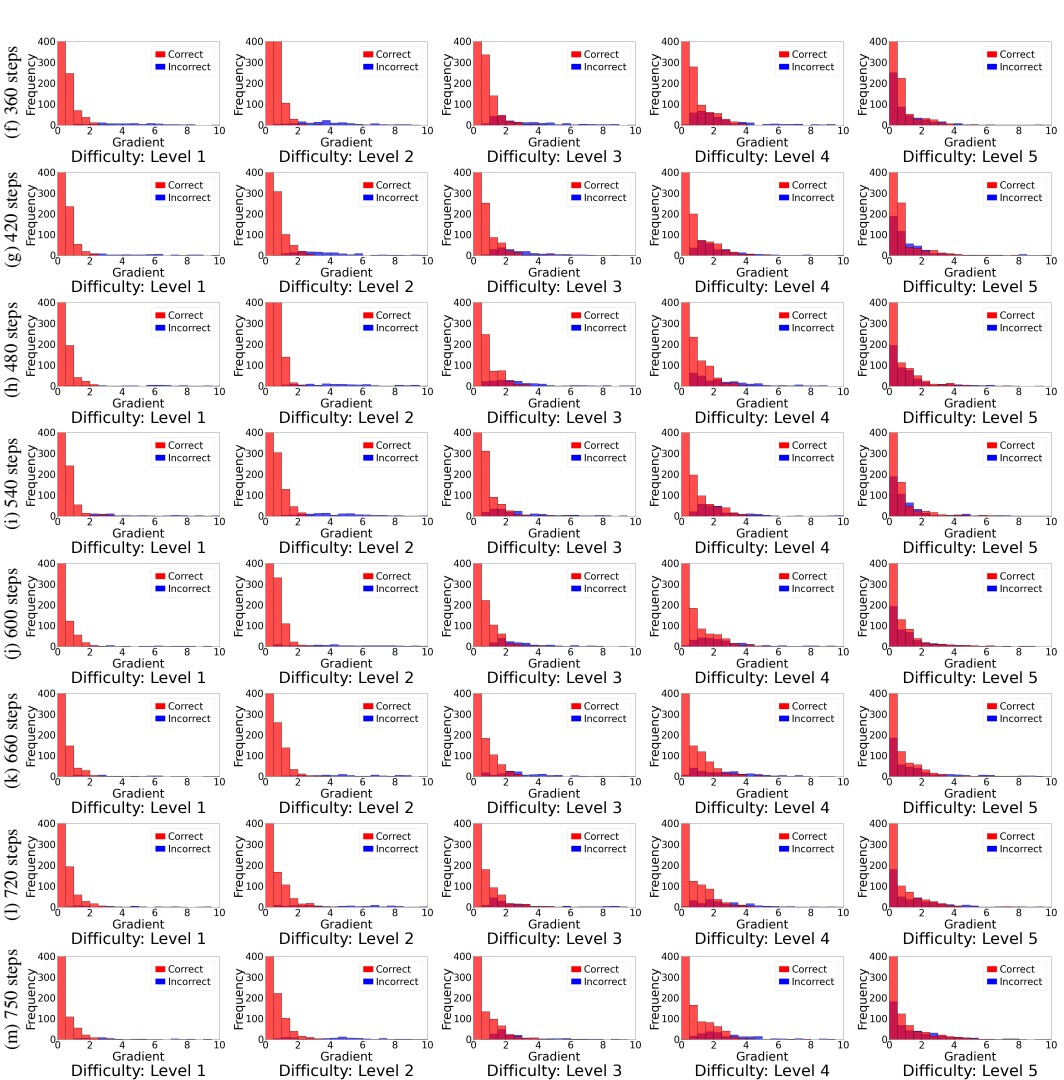

Figure 17: The gradient distribution of the Qwen2.5-Math-1.5B model, post-training on the MATH dataset with the CoDaPO algorithm. We present the distribution of model checkpoints captured at every 60 training steps. Columns 1 through 5 correspond to difficulty levels 1 through 5. (consistent with Fig. 16).

# I   CASE STUDIES

**In-domain case.** We analyze the performance gap between CoDaPO and GRPO through a symbolic reasoning task that requires algebraic manipulation, base conversion, and number-theoretic reasoning in Fig. 18. While both methods adopt similar initial steps, only CoDaPO successfully arrives at the correct and complete solution. This discrepancy reveals a broader insight: CoDaPO demonstrates a stronger capacity for maintaining symbolic consistency and handling algebraic constraints, whereas GRPO is more susceptible to local errors and brittle logic execution.

A key distinction lies in how the two models approach intermediate decision points. CoDaPO tends to preserve the symbolic structure of the problem throughout the reasoning process, producing interpretable and logically coherent derivations. In contrast, GRPO is more prone to heuristic or trial-based reasoning patterns, which may yield superficially plausible but ultimately incorrect results—especially in cases requiring discrete enumeration or careful constraint satisfaction.

This case exemplifies a common challenge in reinforcement learning for language models: small reasoning errors in early steps often cascade into incorrect final answers, and methods lacking robust symbolic understanding struggle to recover. CoDaPO mitigates this through more structured reasoning and better alignment between confidence and difficulty signals, which enhances its robustness in solving multi-step, discrete, and mathematically grounded problems.

These findings suggest that CoDaPO is not only effective in improving accuracy but also in enhancing reasoning fidelity and interpretability, especially in domains like mathematics, programming, and logic that require precise, symbolic manipulation.

**Out-of-domain case.** In tasks that require translating natural language descriptions into symbolic formulations grounded in domain knowledge—such as those involving physical systems or scientific laws (Fig. 19)—we observe a consistent performance gap between the two methods. CoDaPO demonstrates stronger semantic alignment between the problem description and the constructed solution model. Its responses tend to follow a coherent chain of physical and mathematical reasoning, with clear transitions from qualitative understanding to quantitative execution.

In contrast, GRPO is more prone to semantic drift: it often introduces irrelevant concepts, misapplies formulas, or incorrectly anchors symbolic steps to the problem's actual objective. This can lead to answers that are numerically plausible but logically invalid, due to disconnects between intermediate computations and the final question. Such behavior suggests difficulties in maintaining global consistency across multi-step derivations—particularly when domain-specific reasoning is required.

More generally, these observations highlight the importance of model capabilities in selective abstraction and structured model-building. An effective llm solver must not only recall relevant formulas or operations, but also identify the appropriate scope of reasoning, exclude irrelevant detail, and maintain logical consistency through multi-step symbolic transformations. CoDaPO exhibits greater stability along these dimensions, making it more reliable on tasks that blend symbolic reasoning with physical interpretation.

**AIME25 Question**

Find the sum of all integer bases $b > 9$ for which $17_b$ is a divisor of $97_b$.

Ground Truth: $\boxed{70}$

**CoDaPO Response**

To solve this problem, we need to convert the numbers $17_b$ and $97_b$ from base $b$ to base 10 and then find the values of $b$ for which $17_b$ divides $97_b$.
1. Convert $17_b$ to base 10:

$$17_b = 1 \cdot b + 7 = b + 7.$$

2. Convert $97_b$ to base 10:

$$97_b = 9 \cdot b + 7 = 9b + 7.$$

We need $17_b$ to be a divisor of $97_b$, so:

$$9b + 7 \equiv 0 \pmod{b + 7}.$$

This means:
$$9b + 7 = k(b + 7)$$

for some integer $k$. Simplifying this, we get:

$$9b + 7 = kb + 7k$$
$$9b - kb = 7k - 7$$
$$b(9 - k) = 7(k - 1)$$

So:
$$b = \frac{7(k - 1)}{9 - k}.$$

Since $b > 9$, $k$ must be chosen such that $b$ is an integer greater than 9. Let's find the valid values of $k$ and corresponding $b$.
We check the values of $k$ for which the right-hand side is an integer and $b > 9$:

- For $k = 2$: $b = \frac{7(2-1)}{9-2} = \frac{7}{7} = 1$, not greater than 9.
- For $k = 3$: $b = \frac{7(3-1)}{9-3} = \frac{14}{6} = \frac{7}{3}$, not an integer.
- For $k = 4$: $b = \frac{7(4-1)}{9-4} = \frac{21}{5}$, not an integer.
- For $k = 5$: $b = \frac{7(5-1)}{9-5} = \frac{28}{4} = 7$, not greater than 9.
- For $k = 6$: $b = \frac{7(6-1)}{9-6} = \frac{35}{3}$, not an integer.
- For $k = 7$: $b = \frac{7(7-1)}{9-7} = \frac{42}{2} = 21$, valid.
- For $k = 8$: $b = \frac{7(8-1)}{9-8} = \frac{49}{1} = 49$, valid.
- For $k = 9$: $b = \frac{7(9-1)}{9-9}$ is undefined.
- For $k = 10$: $b = \frac{7(10-1)}{9-10} = \frac{63}{-1} = -63$, not greater than 9.

So the valid values for $b$ are 21 and 49. The sum of these values is:
$$21 + 49 = 70.$$

Thus, the final answer is:

$$\boxed{70}$$

**GRPO Response**

To solve the problem, we need to translate the numbers $17_b$ and $97_b$ from base $b$ to base 10.
The number $17_b$ in base 10 is:

$$1 \cdot b + 7 = b + 7$$

The number $97_b$ in base 10 is:

$$9 \cdot b + 7 = 9b + 7$$

We need to find the values of $b$ such that $17_b$ (which is $b + 7$) is a divisor of $97_b$ (which is $9b + 7$). This means that:
$$b + 7 \mid 9b + 7$$

or equivalently:

$$9b + 7 \equiv 0 \pmod{b + 7}$$

We can express this congruence as an equation:

$$9b + 7 = k(b + 7)$$

for some integer $k$. Rearranging the equation:

$$9b + 7 = kb + 7k$$
$$9b - kb = 7k - 7$$
$$b(9 - k) = 7(k - 1)$$
$$b = \frac{7(k - 1)}{9 - k}$$

For $b$ to be an integer greater than 9, the right-hand side must be an integer. This means $7(k - 1)$ must be divisible by $9 - k$. We can iterate through possible integer values of $k$ to find valid solutions for $b$:

```
# Python code to find valid bases
valid_bases = []
for k in range(10): # Try k from 0 to 9
    if (9 - k) != 0:
        b = 7 * (k - 1) / (9 - k)
        if b.is_integer() and b > 9:
            valid_bases.append(int(b))

sum_of_bases = sum(valid_bases)
print((valid_bases, sum_of_bases))
```

Running this code yields:

valid bases: $[16, 21] \Rightarrow$ sum $= 16 + 21 = \boxed{37}$

Therefore, the final answer is:

$$\boxed{37}$$

Figure 18: Case study of CoDaPO and GRPO Responses to an AIME25 Question.

**Minerva Question**

An eclipsing binary consists of two stars of different radii and effective temperatures. Star 1 has radius $R_1$ and $T_1$, and Star 2 has $R_2 = 0.5R_1$ and $T_2 = 2T_1$. Find the change in bolometric magnitude of the binary, $\Delta m_{bol}$, when the smaller star is behind the larger star. (Consider only bolometric magnitudes so you don't have to worry about color differences.)

Ground Truth: $\boxed{1.75}$

**CoDaPO Response**

To solve this problem, we need to calculate the change in bolometric magnitude of the binary system when the smaller star (Star 2) is behind the larger star (Star 1). The bolometric magnitude of a star is given by:

$$m_{bol} = -2.5 \log_{10}(L_{bol})$$

where $L_{bol}$ is the bolometric luminosity of the star. The bolometric luminosity of a star is proportional to its surface area and its effective temperature:

$$L_{bol} \propto R^2 T^4$$

So, the bolometric luminosities of Star 1 and Star 2 are:

$$L_{bol,1} \propto R_1^2 T_1^4$$

$$L_{bol,2} \propto R_2^2 T_2^4 = (0.5R_1)^2 (2T_1)^4$$
$$= 0.25R_1^2 \cdot 16T_1^4 = 4R_1^2 T_1^4$$

The total bolometric luminosity of the binary system when both stars are visible is:

$$L_{bol,total} = L_{bol,1} + L_{bol,2} = kR_1^2 T_1^4 + k \cdot 4R_1^2 T_1^4$$
$$= 5kR_1^2 T_1^4$$

When Star 2 is behind Star 1, the total bolometric luminosity is just the luminosity of Star 1:

$$L_{bol,total} = L_{bol,1} = kR_1^2 T_1^4$$

The change in bolometric magnitude is given by:

$$\Delta m_{bol} = m_{bol,after} - m_{bol,before}$$
$$= -2.5 \log_{10}\left(\frac{L_{bol,total,after}}{L_{bol,total,before}}\right)$$
$$= -2.5 \log_{10}\left(\frac{kR_1^2 T_1^4}{5kR_1^2 T_1^4}\right)$$
$$= -2.5 \log_{10}\left(\frac{1}{5}\right)$$
$$= -2.5 \log_{10}(5^{-1})$$
$$= -2.5(-\log_{10}(5))$$
$$= 2.5 \log_{10}(5)$$

Using the value $\log_{10}(5) \approx 0.6990$, we get:

$$\Delta m_{bol} = 2.5 \times 0.6990 = 1.7475 \approx 1.75$$

Thus, the change in bolometric magnitude is:

$$\boxed{1.75}$$

**GRPO Response**

To find the change in bolometric magnitude of the binary when the smaller star (Star 2) is behind the larger star (Star 1), we need to consider the effect of the limb darkening and the phase angle of the eclipsing binary.

The bolometric magnitude of a star can be approximated using the Stefan-Boltzmann law, which states that the luminosity of a star is proportional to the fourth power of its temperature and the square of its radius. The bolometric magnitude $m_B$ of a star is related to its luminosity by:

$$m_B = -2.5 \log_{10}(L) + C$$

where $L$ is the luminosity of the star and $C$ is a constant. For a binary system, the change in bolometric magnitude when one star passes in front of the other can be approximated by considering the change in the effective area of the star that is visible during the eclipse.

The effective area of a star that is visible during an eclipse is given by:

$$A = R^2 \left(1 - \left(\frac{R}{d}\right)^2 \sin^2(\theta)\right)$$

where $R$ is the radius of the star, $d$ is the distance to the star, and $\theta$ is the phase angle. The change in the effective area when Star 2 is behind Star 1 is given by the difference in the areas of the two stars:

$$\Delta A = \pi R_1^2 - \pi R_2^2 \left(1 - \left(\frac{R_2}{d}\right)^2 \sin^2(\theta)\right)$$

Since we are considering only the bolometric magnitude, we can simplify the problem by considering the ratio of the luminosities of the two stars. The change in bolometric magnitude is given by:

$$\Delta m_{bol} = -2.5 \log_{10}\left(\frac{L_2}{L_1}\right)$$

where $L_1$ and $L_2$ are the luminosities of Star 1 and Star 2, respectively. Using the Stefan-Boltzmann law, we have:

$$\frac{L_2}{L_1} = \left(\frac{T_2}{T_1}\right)^4 \left(\frac{R_2}{R_1}\right)^2 = \left(\frac{2T_1}{T_1}\right)^4 \left(\frac{0.5R_1}{R_1}\right)^2$$
$$= 2^4 \cdot 0.5^2 = 16 \cdot 0.25 = 4$$

Thus, the change in bolometric magnitude is:

$$\Delta m_{bol} = -2.5 \log_{10}(4) = -2.5 \cdot 0.60206$$
$$= -1.50515 \approx -1.51$$

Therefore, the change in bolometric magnitude of the binary when the smaller star is behind the larger star is:

$$\boxed{-1.51}$$

Figure 19: Case study of CoDaPO and GRPO Responses to a Minerva Question.

