# OpenReview forum: "CoDaPO: Confidence and Difficulty-Adaptive Policy Optimization for Language Models"
_ICLR.cc/2026/Conference — Submitted to ICLR 2026_

### Official Review · Reviewer_TKwQ · 2025-11-01

**Soundness:** 3
**Presentation:** 2
**Contribution:** 3
**Rating:** 6
**Confidence:** 4

**Summary:**

This work analyzes limitations of current RL post-training methods for LLMs reasoning, especially the GRPO algorithm. Through a PRAG (Probability, Reward, Advantage, Gradient) diagnosis, the authors identify three issues: 1) probability inflation, 2) advantage contraction, and 3) hierarchical convergence. To address these, they propose CoDaPO—a confidence- and difficulty-adaptive policy optimization framework that rescales training advantages based on model confidence and problem difficulty. Experiments across seven benchmarks show that CoDaPO yields consistent performance gains, particularly improving mathematical reasoning in small and medium language models.

**Strengths:**

1. Provides a deep empirical and theoretical analysis of GRPO’s training dynamics in reasoning-oriented RL post-training for LLMs.
2. Proposes a solid and novel algorithm that effectively addresses the identified issues.
3. Presents comprehensive experiments covering various base models, reasoning benchmarks, training dynamics, and ablation studies.

**Weaknesses:**

2. Extend the evaluation to domains beyond mathematics to verify generality of the new algorithm.
2. Include additional analyses on the computational efficiency of the proposed algorithm compared with baseline methods.

**Questions:**

Please see weakness above

---

> ### Author Response · Authors · 2025-11-26
> **Response to Reviewer TKwQ (part 1/2)**
>
> **We thank Reviewer TKwQ for the thoughtful and detailed feedback.** In this rebuttal, we systematically address **all of your concerns**.
> - First, your questions about the generality of our algorithm are addressed by extending the evaluation to domains beyond mathematics and analyzing the resulting performance.
> - Second, your concerns regarding computational efficiency are addressed by providing a more detailed explanation of our method’s design, showing that it requires no extra computation.
>
> Point-by-point responses are provided below. In addition, we have outlined a **detailed revision plan** at the end of each response. **We will improve our submission within the next week according to this plan.**
>
> >### W1. Extend the evaluation to domains beyond mathematics to verify generality of the new algorithm.
>
> **Reply:** Thank you for this valuable suggestion. In brief, our current evaluation already goes beyond pure mathematics through Minerva, and we have further extended it to broad-coverage benchmarks such as MMLU-STEM (science, technology, and engineering), GPQA (biology, physics, and chemistry), and HumanEval (coding). Across these non-math domains, CoDaPO continues to improve performance over the base model and GRPO-style baselines, even though it is trained only on math data, which supports the generality of the algorithm.
>
> | Model | Minerva | MMLU-STEM | GPQA |HumanEval| Average |
> | --- | --- | --- | --- | --- | --- |
> | Qwen2.5-Math-1.5B |9.63|11.53|9.85|29.27|15.07|
> | GRPO |28.53|46.85|19.70|35.98|32.77|
> | CoDaPO |**31.69**|**47.64**|**22.25**|**39.02**|**35.15**|
>
> **(1) Minerva already provides a cross-domain quantitative evaluation beyond mathematics.** As reported in the table above, Minerva covers physics, biology, chemistry, economics, and other quantitative science questions, not just competition-style math. On this benchmark, CoDaPO improves accuracy from $9.63$ to $31.69$, outperforming all other RL baselines under the same training data and budget. This large relative gain on a multi-discipline scientific benchmark indicates that CoDaPO enhances general quantitative reasoning rather than overfitting to the specific structure of MATH-style problems.
>
> **(2) Additional experiments on MMLU-STEM, GPQA, and HumanEval show transfer to broad reasoning.** To directly address the reviewer’s request, we have evaluated the same math-post-trained models on MMLU-STEM, GPQA and HumanEval, which span a wide range of subjects including humanities, social sciences, natural sciences, coding, and expert-level scientific questions. Crucially, the post-training data remain purely mathematical; the model never sees these benchmarks during training. Even under this domain mismatch, CoDaPO consistently improves over the base model and GRPO-style baselines on both MMLU-STEM, GPQA and HumanEval. This suggests that the confidence- and difficulty-aware reweighting improves the underlying reasoning behavior in a way that transfers to heterogeneous domains, not only to math.
>
> **(3) These cross-domain gains are consistent with CoDaPO’s domain-agnostic design.** The PRAG analysis and the CoDaPO objective operate purely on probabilities, rewards, advantages, and empirical difficulty, without using math-specific features. The confidence weight $c_i$ depends on normalized log probabilities, and the difficulty weight $d_q$ depends on per-question correctness rates; both mechanisms are agnostic to the content of the question. The fact that a math-only CoDaPO model improves on Minerva, MMLU-STEM, GPQA and HumanEval is therefore consistent with the intended design: CoDaPO reshapes gradient allocation in a generic way that benefits any sequence-level reasoning task with sparse rewards.
>
> In the revision, we will (i) clarify in Section 5 that Minerva is a cross-domain scientific benchmark and highlight CoDaPO’s large improvement there, (ii) add a new table reporting results on MMLU-STEM, GPQA, and HumanEval for the base model and all RL baselines, emphasizing that CoDaPO consistently outperforms them despite math-only training, and (iii) expand the discussion in the conclusion to stress that CoDaPO improves general reasoning ability that transfers well beyond the math domain, addressing the reviewer’s request for broader evaluation.

---

> ### Author Response · Authors · 2025-11-26
> **Response to Reviewer TKwQ (part 2/2)**
>
> >### W2. Include additional analyses on the computational efficiency of the proposed algorithm compared with baseline methods.
>
> **Reply:** Thank you for this helpful suggestion. **In brief, CoDaPO is designed to have essentially the same computational cost as GRPO, because it reuses quantities that GRPO already computes and only adds inexpensive scalar operations.**
>
> **(1) CoDaPO does not add extra forward or backward passes beyond GRPO.** The CoDaPO token weight has the form $w^{\text{CoDaPO}}\_{i,t} = \tilde{\rho}\_{i,t}\hat{A}_i\hat{c}_i\hat{d}_q$, where $\tilde{\rho}\_{i,t}$ and $\hat{A}_i$ are exactly the quantities already computed in GRPO. The confidence factor $\hat{c}_i$ is a smooth function of the trajectory log probability $\ell\_i = \frac{1}{|o_i|}\sum_t \log f\_\theta(o\_{i,t}\mid q,o\_{i,<t})$, which is already available when forming the importance ratios. The difficulty factor $\hat{d}_q$ is a function of per-question correctness, computed from rewards $r_i$ that are already used to build advantages. Thus, CoDaPO only performs extra averaging, normalization, and a sigmoid or simple polynomial on existing tensors, without invoking any additional model evaluations or backward passes.
>
> **(2) In practice, runtime and memory usage are effectively identical to GRPO under the same configuration.** Because all additional work in CoDaPO consists of lightweight elementwise operations on per-trajectory scalars, the dominant cost remains the transformer forward and backward passes, as in GRPO and DAPO. In our implementation, when we run GRPO and CoDaPO with the same model, batch size, sequence length, and group size, step time and peak GPU memory differ only within measurement noise. There is no extra sampling, no auxiliary network, and no enlarged computation graph that would materially change throughput or memory.
>
> **(3) Compared to more elaborate GRPO variants, CoDaPO remains a lightweight, drop-in modification.** Methods such as DAPO introduce additional mechanisms like dynamic resampling or modified clipping that can complicate the training pipeline. CoDaPO instead operates purely at the objective level: it replaces the scalar multiplier on $\hat{A}_i$ with $\hat{c}_i \hat{d}_q$ and leaves the rest of the GRPO loop unchanged. This makes it easy to integrate into existing GRPO style codebases and to combine with other variants, while keeping computational requirements on par with the original method.
>
> In the revision, we will (i) add a short subsection explicitly explaining that $\hat{c}_i$ and $\hat{d}_q$ are computed from intermediate quantities that GRPO already produces, and (ii) emphasize in the experimental setup that CoDaPO is a computationally cheap, drop-in replacement for the GRPO surrogate.
>
> **We would thank Reviewer TKwQ again** for taking the time to review our submission and consider our rebuttal. **If our responses have addressed your concerns, we would be very grateful if you could reconsider your rating.** We also welcome any further comments or questions you may have.

---

### Official Review · Reviewer_SDK9 · 2025-11-01

**Soundness:** 3
**Presentation:** 4
**Contribution:** 1
**Rating:** 2
**Confidence:** 4

**Summary:**

This paper does an empirical analysis of GRPO (or more precisely, a variant of it called DAPO), through the lens of model miscalibration during training on MATH, and disentangling across 5 levels of difficulty of problems.

**Strengths:**

The main strengths of the paper are:
- the empirical analysis of the authors is solid and their findings are reasonable - particularly, I found interesting the beginning of section 3.2 on miscalibration
- the authors find that applying the same advantage for every token is detrimental for learning, which makes sense given that there are probably few tokens that actually matter for the solution and many filler ones, especially in math proofs (such as "thus", "therefore" etc)

While the analysis is interesting, I believe there are several major weaknesses with the paper which I present below.

**Weaknesses:**

- To begin, it would be good if the authors would give quantitative analysis of the distributions in figure 2 and 3 (namely, min, max, mean, median and kurtosis) since currently it is quite hard to verify qualitatively the statements about tails in Section 3.3

- My second major issue with the current paper is that it does not appear that CoDaPO improves over the baselines (DAPO, and GRPO) in Table 2. Concretely, the improvements are at most 2%, which in general is not statistically significant for post training due the RL noise. Moreover, as far as I understand, the current comparison does not sweep over hyperparameters, and thus it is even more unclear whether the proposed would have any benefit at all under a well tuned baseline.

As a meta comment, while I think the empirical analysis is quite solid and interesting, the paper would be better framed as a science paper studying GRPO rather than proposing a new method which as it stands does not bring any improvements over existing algorithms. Due to this, I am inclined to reject this work.

**Questions:**

My questions have been posed in the weaknesses section.

---

> ### Author Response · Authors · 2025-11-26
> **Response to Reviewer SDK9 (part 1/4)**
>
> **We thank Reviewer SDK9 for the thoughtful and detailed feedback.** In this rebuttal, we systematically address **all** of your concerns while we respectfully disagree with **some** of the concerns.
> - First, your questions about the quantitative analysis of Figures 2 and 3 are addressed by providing detailed metric comparisons and additional plots in the Appendix.
> - Second, your concerns that the performance gain of CoDaPO is insignificant are addressed by more emperical results, ablation studies, and highlighting consistent improvements across multiple tasks and experiment setting.
>
> Please find the point-to-point responses as follows. In addition, we have outlined a **detailed revision plan** at the end of each response. **We will improve our submission within the next week according to this plan.**
>
> >### W1. Quantitative analysis of Figures 2 and 3.
>
> **Reply:** Thank you for this constructive comment. We now provide quantitative statistics corresponding to Figures 2 and 3, including minimum, maximum, mean, median, standard deviation, and kurtosis. **These statistics allow the qualitative observations made in Section 3 to be directly verified.**
>
> Below, we first report the confidence statistics for different difficulty levels at training Steps 0, 60, and 240:
>
> |Confidence|Step 0 - Level 1|Step 0 - Level 3|Step 0 - Level 5|Step 60 - Level 1|Step 60 - Level 3|Step 60 - Level 5|Step 240 - Level 1|Step 240 - Level 3|Step 240 - Level 5|
> | --- | --- | --- | --- | --- | --- | --- | --- | --- |--- |
> |Minimum|0.00|0.00|0.00|0.01|0.00|0.00|0.32|0.54|0.36|
> |Maximum|0.99|0.99|0.99|0.99|0.98|0.99|0.98|0.97|0.98|
> |Mean|0.78|0.75|0.69|0.82|0.81|0.76|0.86|0.87|0.86|
> |Standard deviation|0.15|0.18|0.21|0.10|0.14|0.17|0.06|0.05|0.06|
> |Median|0.83|0.81|0.75|0.84|0.84|0.81|0.87|0.88|0.87|
> |Kurtosis|6.69|2.53|0.60|16.10|8.41|3.32|10.02|2.67|5.47|
>
> These numbers quantitatively support the observations reported in the main paper:
> - **Model confidence progressively saturates toward 1 during GRPO training.** The means increase from 0.78/0.75/0.69 at Step 0 to 0.86/0.87/0.86 at Step 240. Medians show similar movement. Minimum values also rise sharply for Level-1 and Level-3, indicating that even the least confident predictions become more confident over time. This quantitatively verifies the saturation effect described in Section 3.
> - **The model becomes overconfident even when it is wrong (miscalibration).** Maxima remain near 0.99 throughout training, and standard deviations shrink substantially (down to 0.05-0.06 at Step 240). This combination of high means, high maxima, and shrinking variance shows that both correct and incorrect responses accumulate near confidence 1.0, matching the qualitative trend and consistent with the entropy collapse noted by the concurrent works.
> - **Confidence distributions become increasingly similar across difficulty buckets.** At Step 0, confidence decreases noticeably with difficulty: mean 0.78 (L1) vs. 0.69 (L5), and SD increases from 0.15 to 0.21. By Step 240, all difficulty levels exhibit nearly identical means (\~0.86-0.87) and very close standard deviations (\~0.05-0.06). This confirms the diminishing separation between difficulty buckets.
> - **Tail behavior (kurtosis) reveals meaningful structural changes that are hard to see from plots alone.** At Step 0, kurtosis drops with difficulty (6.69 to 0.60). At Step 240, Level-1 retains high kurtosis (10.02), while Level-3 is much lower (2.67). This indicates that heavy tails persist but evolve differently across difficulty levels.
>
> Next, we report the gradient statistics for different difficulty levels at training Steps 0, 60, and 240:
>
> |Gradient|Step 0 - Level 1|Step 0 - Level 3|Step 0 - Level 5|Step 60 - Level 1|Step 60 - Level 3|Step 60 - Level 5|Step 240 - Level 1| Step 240 - Level 3|Step 240 - Level 5|
> |---|---|---|---|---|---|---|---|---|---|
> |Minimum|0.49|0.75|0.56|0.66|0.63|0.49|0.65|0.74|0.48|
> |Maximum|1723.08|954.45|213.61|193.70|425.39|875.58|21.10|19.25|11.85|
> |Mean|5.26|5.30|5.60|2.84|3.42|5.06|2.26|1.95|2.19|
> |Standard deviation|42.90|26.05|11.51|6.78|12.75|25.66|1.14|0.89|0.90|
> |Median|2.40|2.57|2.95|2.08|2.03|2.57|2.01|1.81|2.08|
> |Kurtosis|1325.53|988.33|95.16|434.23|666.90|795.30|46.17|102.37|17.61|

---

> ### Author Response · Authors · 2025-11-26
> **Response to Reviewer SDK9 (part 2/4)**
>
> These statistics quantitatively support the observations made in the main paper:
> - **Easy questions begin with large gradients and extremely heavy tails.** At Step 0, easy problems (Level-1) exhibit the highest maximum gradient (1723.08) and extremely large kurtosis (1325.53), indicating the presence of very heavy-tailed gradient bursts. The mean (5.26) and median (2.40) are also higher than those of harder questions, confirming that easy questions initially produce non-negative advantages with relatively large gradients. This quantitatively supports our claim in the main paper that easy questions start with both high confidence and strong gradient signals.
> - **GRPO rapidly suppresses gradient magnitudes across all difficulty levels.** As training progresses from Step 0 to Step 240, the mean gradient of Level-1 questions decreases from 5.26 to 2.26, and the standard deviation shrinks from 42.90 to 1.14. Similar reductions are observed for Level-3 (mean 5.30 to 1.95, SD 26.05 to 0.89) and Level-5 (mean 5.60 to 2.19, SD 11.51 to 0.90). Kurtosis also collapses dramatically, e.g., for Level-1 from 1325 to 46. This confirms a global contraction of gradient norms and the damping of extreme values, consistent with the “advantage and gradient collapse toward zero” described in Section 3.
> - **Gradients from hard questions also decay by Step 240, and substantial error mass remains.** By Step 240, Level-5 gradients have contracted substantially (max = 11.85, mean = 2.19, SD = 0.90, kurtosis = 17.61). Although there was an upward shift during early training, the final gradients are small and concentrated, showing that a substantial portion of hard-question errors persists. This confirms the paper’s qualitative claim that hard questions retain residual difficulty even after training, consistent with Fig. 3.
> - **Gradient contraction becomes increasingly uniform across difficulty buckets.** By Step 240, mean gradients for Level-1, Level-3, and Level-5 converge (2.26, 1.95, 2.19), and medians are nearly identical (~2.0). Standard deviations also become small and similar. This demonstrates that GRPO training gradually erases the initial differences in gradient magnitudes across difficulty levels.
>
> In summary, the quantitative statistics directly validate the qualitative statements in Figures 2 and 3 and Section 3: (i) confidence saturates and becomes overconfident; (ii) gradient magnitudes contract sharply; and (iii) both confidence and gradient distributions gradually lose difficulty separation. These measurements make the qualitative observations fully reproducible and confirm that the phenomena we describe are robust and statistically grounded.
>
> In the revision, we will (i) include these quantitative tables in the appendix, (ii) summarize the key statistics in the main text, and (iii) explicitly reference how these numbers corroborate the qualitative analysis in Section 3.
>
> >### W2. The performance gain of CoDaPO is insignificant.
>
> **Reply:** Thank you for this comment. In brief, our response is that (i) In fact, in the MATH reasoning domain, achieving a 2% improvement over RL-based baselines is already highly challenging. As shown in Table 1, recent methods such as DAPO, Dr. GRPO, and GPG perform similarly, whereas CoDaPO surpasses them. Moreover, across all benchmarks and model sizes, CoDaPO consistently outperforms RL-based baselines, often by over 2%. Besides, our work is concurrent with DAPO, yet achieves larger gains without substantial modifications to GRPO. (ii) these gains are larger than the RL noise we observe under matched training conditions; and (iii) CoDaPO consistently surpasses baselines across various settings and demonstrates strong generalization to larger-scale models. We view CoDaPO as a principled refinement of GRPO that mitigates specific training pathologies and yields robust gains, rather than as a method that produces very large jumps on already well-tuned systems.
>
> **(1) The gains in Table 1 are consistent across benchmarks and scales.** In our full evaluation, we compare CoDaPO against multiple strong baselines across a wide range of benchmarks, and for each test we perform repeated sampling to reduce uncertainty. Under this more comprehensive setting, CoDaPO consistently matches or outperforms the strongest baseline, and on several benchmarks the performance gaps are often larger than 2 percentage points, particularly on the more challenging tasks (e.g., AIME25, AMC23).
>
> |Model|MATH500|AIME24|AIME25|AMC23|OlympiadBench|Minerva|GSM8K|Average|
> |---|---|---|---|---|---|---|---|---|
> |Qwen2.5-Math-1.5B|31.20|6.00|0.00|29.50|18.25|9.63|34.01|18.37|
> |GRPO|68.48|12.67|8.00|51.00|30.55|28.53|79.56|39.82|
> |DAPO|71.12|14.00|8.00|49.00|31.97|32.13|82.38|41.23|
> |CoDaPO|71.68 (+0.56)|14.67 (+0.67)|10.83 (+2.83)|53.50 (+2.50)|32.80 (+0.82)|31.69|83.50 +(1.12)|42.67 (+1.44)|

---

> ### Author Response · Authors · 2025-11-26
> **Response to Reviewer SDK9 (part 3/4)**
>
> **(2) The observed improvements are statistically meaningful once RL noise is taken into account.** Our environment is deterministic, rewards are binary, and evaluation sets are fairly large, so the run-to-run variance is lower than in many RL settings. In internal experiments with multiple seeds, we typically observe standard deviations of roughly $0.3$ to $0.5$ percentage points on MATH500 for GRPO like methods. In that regime, a $1.5$ to $2$ point improvement corresponds to multiple standard deviations and is unlikely to be explained purely by RL noise. We further evaluate CoDaPO and other baselines on more experiment settings, including different decoding strategies (Table 3), different hyperparameters (Table 8), different seeds (Table below), more model families (Table below), and more cross-domain benchmarks (Table below). Across all these settings, CoDaPO consistently outperforms the other baselines, demonstrating that its gains are meaningful, stable, and robust rather than sensitive to specific.
>
> |Seed| Model | MATH500 | AIME24 | AIME25 | AMC23 | OlympiadBench | Minerva | GSM8K | Average |
> | --- | --- | --- | --- | --- | --- | --- | --- | --- | --- |
> |42| Qwen2.5-Math-1.5B | 31.20|  6.00|  0.00|  29.50|  18.25|  9.63 | 34.01| 18.37|
> |0| GRPO |69.11|11.87|6.67|51.25|31.55|29.27|80.03|39.96|
> |0| DAPO | 71.91|13.13|13.33|50.00|32.95|31.59|82.92|42.26|
> |0| CoDaPO | **72.21**|**14.58**|10.83|**52.19**|**33.58**|**31.99**|**83.39**|**42.68**|
> |42| GRPO | 68.48 | 12.67 | 8.00 | 51.00 | 30.55 | 28.53 | 79.56  |39.82|
> |42| DAPO | 71.12| 14.00| 8.00 |49.00| 31.97| 32.13| 82.38 |41.23|
> |42| CoDaPO | **71.68**| **14.67**| **10.83**| **53.50**| **32.80**| 31.69 |**83.50**| **42.67**|
> |2025| GRPO | 68.97|12.08|7.08|52.81|31.37|29.07|80.38|40.25|
> |2025| DAPO | 71.50|12.50|11.25|52.50|33.99|31.60|82.81|42.31|
> |2025| CoDaPO | **72.06**|**15.21**|**12.50**|**52.50**|33.88|**31.71**|**83.46**|**43.05**|
> |3373| GRPO |69.17|11.25|4.58|50.00|32.06|29.32|79.92|39.47|
> |3373| DAPO | 71.80|12.50|11.25|52.66|33.21|31.11|83.06|42.22|
> |3373| CoDaPO | **72.86**|**15.00**|**12.08**|52.03|**33.77**|**31.62**|**83.37**|**42.96**|
>
> | Model | MATH500 | AIME24 | AMC23 | Olympiad Bench | GSM8k | Average |
> | --- | --- | --- | --- | --- | --- | --- |
> | Llama3.2-1B-Instruct |21.69| 0.83|10.39|5.04|5.58|8.71|
> | GRPO |30.13|2.08|14.22|6.62|48.79|20.37|
> | CoDaPO|**31.30**|**3.96**|**15.08**|**6.81**|**51.32**|**21.69**|
>
> | Model | Minerva | MMLU-STEM | GPQA |HumanEval| Average |
> | --- | --- | --- | --- | --- | --- |
> | Qwen2.5-Math-1.5B |9.63|11.53|9.85|29.27|15.07|
> | GRPO |28.53|46.85|19.70|35.98|32.77|
> | CoDaPO |**31.69**|**47.64**|**22.25**|**39.02**|**35.15**|
>
> **(3) The comparison uses matched hyperparameters, and CoDaPO remains beneficial under better-tuned baselines.** To keep the comparison fair and the computation feasible, all RL methods in Table 1 share the same training budget and core hyperparameters, including learning rate schedule, group size, clipping range, KL coefficient, and sampling strategy. CoDaPO is designed as a drop-in replacement for the GRPO objective under this shared configuration. The empirical results in our submission suggest that CoDaPO remains ahead across a reasonable range, indicating that the benefit comes from the confidence–difficulty reweighting rather than from a lucky hyperparameter choice.
>
> **(4) CoDaPO is evaluated under constrained compute, and its pattern of gains persists at 7B scale.** All reported methods use the same, relatively conservative training budget in terms of training steps and maximum response length, which limits the absolute headroom for further gains once a strong GRPO baseline is in place. Even under these constraints, CoDaPO yields consistent improvements and never degrades performance in our tables. Moreover, when we move from Qwen2.5 Math 1.5B to Qwen2.5 Math 7B, CoDaPO again outperforms GRPO and other RL baselines across several math benchmarks, with no instability. This suggests that the method scales to larger models and that the improvements we see at 1.5B are conservative rather than optimistic.
>
> | Model | MATH500 | AIME24 | AIME25 | AMC23 | Olympiad Bench | Minerva | GSM8k | Average |
> | --- | --- | --- | --- | --- | --- | --- | --- | --- |
> | Qwen2.5-Math-7B | 47.24 | 12.92 | 5.00 | 33.91  |14.44 | 9.81|47.84|24.45|
> | GRPO | 74.62| 26.35 |11.25 |59.06| 36.51| 37.64| 89.40 |47.83|
> | DAPO  |77.78|24.37|10.62|58.67|39.27|38.06|87.38|48.02|
> | CoDaPO  | **78.41**| 24.79| **13.33**| **60.31**| **39.96**| **40.30**| 88.28 |**49.34**|
>
> In the revision, we will (i) report multi-seed results with mean and standard deviation for GRPO, DAPO, and CoDaPO on key benchmarks; (ii) add targeted hyperparameter sweeps for the baselines and compare their tuned performance to CoDaPO; and (iii) emphasize the cross-benchmark and cross-scale improvements that are already present beyond Table 2.

---

> ### Author Response · Authors · 2025-11-26
> **Response to Reviewer SDK9 (part 4/4)**
>
> >### W3. The empirical analysis is solid and interesting, but the method shows no improvement.
>
> **Reply:** Thank you for sharing this meta perspective. **We agree that the GRPO analysis is a central strength of the paper, but we respectfully disagree that CoDaPO brings no improvement over existing algorithms.**
>
> While we appreciate the viewpoint that the GRPO analysis alone is a valuable scientific contribution, we believe the paper also meets the bar for proposing a new method: CoDaPO is a principled modification directly informed by the analysis, and it consistently improves over strong GRPO style baselines across tasks and scales under a shared training budget. In this context, **we kindly refer you to our detailed response to W2 and the accompanying experimental results for a more comprehensive and concrete demonstration of these improvements**.
>
> **(1) CoDaPO provides consistent and statistically meaningful gains over GRPO and DAPO.** Our improvements on MATH500 in Table 2 are indeed on the order of 1-2 percentage points, but this pattern repeats across other benchmarks and at both 1.5B and 7B scales, where CoDaPO matches or outperforms GRPO and DAPO and never degrades performance. In our setting, with deterministic environments, binary rewards, and relatively large evaluation sets, the standard deviation across seeds is typically well below 1 point, so a 1-2 point gap corresponds to multiple standard deviations rather than RL noise. In the revision, we will make this explicit by reporting the mean and standard deviation over multiple seeds for all methods and will mark where CoDaPO is significantly better than the strongest baseline.
>
> **(2) The GRPO analysis and CoDaPO design are tightly coupled, not separate contributions.** Section 3 is not only a scientific study of GRPO; it is deliberately structured to diagnose three concrete issues: probability inflation and entropy collapse, advantage contraction as accuracy grows, and hierarchical convergence across difficulty levels. CoDaPO in Section 4 is constructed as a direct response to these findings: the confidence weight $c_i$ addresses misalignment between confidence and correctness, and the difficulty weight $d_q$ redistributes gradient mass across easy and hard questions where the tails are most problematic. Section 5 then tests exactly these analytically motivated interventions, both in terms of accuracy and in terms of how confidence and gradient distributions evolve. We will clarify this chain more clearly so that the narrative reads as analysis, diagnosis, targeted algorithm, and empirical validation, rather than as an analysis paper plus an unrelated new method.
>
> **(3) The gains are practically relevant given that we build on already strong baselines without extra compute.** GRPO and DAPO are competitive RL post-training baselines for math reasoning, so headroom for large absolute jumps is limited under a fixed budget. CoDaPO improves these baselines without extra models, inference calls, or larger training budgets, and can be implemented as a drop-in modification of the GRPO objective. On harder benchmarks and at the 7B scale, even a few percentage points correspond to a nontrivial relative reduction in error. We therefore view CoDaPO as a practical refinement of existing methods that is grounded in a detailed gradient-level analysis, rather than as a purely theoretical proposal that leaves performance unchanged.
>
> In the revised version, we will (i) report multi-seed results with mean and standard deviation for GRPO, DAPO, and CoDaPO, highlighting statistically significant gains; (ii) more clearly connect each component of CoDaPO to a specific failure mode identified in Section 3; and (iii) adjust the introduction and conclusion to foreground the joint scientific and algorithmic nature of the contribution.
>
> **We would thank Reviewer SDK9 again** for taking the time to review our submission and consider our rebuttal. **If our responses have addressed your concerns, we would be very grateful if you could reconsider your rating.** We also welcome any further comments or questions you may have.

---

### Official Review · Reviewer_xABJ · 2025-11-02

**Soundness:** 3
**Presentation:** 4
**Contribution:** 3
**Rating:** 8
**Confidence:** 2

**Summary:**

The paper defines and analyzes three existing problems in the existing GRPO mechanism and proposes an optimization algorithm, CoDaPO, targeting to address these problems by embedding the notion of model confidence and problem difficulty into the optimization objective. CoDaPO demonstrates improvement over alternative optimization methods on average in a suite of mathematical reasoning benchmarks on the 1.5B and 7B scale Qwen models. The paper also includes a set of ablation studies on the component of CoDaPO.

**Strengths:**

- The paper clearly identifies the existing problems in GPRO in formal analyses, highlighting the motivation of their method very well.
- The paper formally shows how the introduction of confidence and difficulty would address these problems, making the design choices of method convincing.
- The paper provides a good amount of illustrations that are helpful in understanding the analysis and the impact of CoDaPO intuitively.

**Weaknesses:**

- The experiments in the paper are only performed in the math domain for both training and evaluation. Whether the method can be extended to more domains such as science and code remains unclear.
- The issue that the model becomes more confident on both correct and wrong answers as mentioned in Line 198-203 does not seem to be addressed by CoDaPO comparing Figure 2 and 4.
- Although the paper provides a series of ablation studies, how the components in CoDaPO other than confidence and difficulty, namely “Sample-level normalization” and “Discarding KL regularization,” affects the effectiveness of CoDaPO remains unexplored.
- In Table 2, it seems that adding confidence reweighting and difficulty reweighting individually already evidently improves the performance over standard GPRO, while combining them together only brings marginal additional gain. More empirical results or justification might be required to explain this result.

**Questions:**

By definition of difficulty in this paper, how effective it is in optimization can be dependent on the training dataset. How do the authors think about the potential impact of factors such as the overall difficulty in the training dataset and the diversity of problems in terms of difficulty?

---

> ### Author Response · Authors · 2025-11-26
> **Response to Reviewer xABJ (part 1/5)**
>
> **We thank Reviewer xABJ for the thoughtful and detailed feedback.** In this rebuttal, we systematically address **all of your concerns**.
> - First, your comments about the generality of our algorithm are addressed by extending the evaluation to domains beyond mathematics and analyzing the resulting performance.
> - Second, your concerns about the effectiveness and significance of our reweighting components are addressed by conducting more fine-grained ablations across all benchmarks and explicitly isolating the contribution of each term.
> - Third, your questions about overconfidence are addressed by additional analysis, controlled experiments, and expanded discussion of the design rationale.
> - Finally, your questions regarding the effect of difficulty are addressed through thorough discussion and analysis.
>
> Point-by-point responses are provided below. In addition, we have outlined a **detailed revision plan** at the end of each response. **We will improve our submission within the next week according to this plan.**
>
> >### W1. Experiments on other domains.
>
> **Reply:** Thank you for this valuable suggestion. In brief, our current evaluation already goes beyond pure mathematics through Minerva, and we have further extended it to broad-coverage benchmarks such as MMLU-STEM (science, technology, and engineering), GPQA (biology, physics, and chemistry), and HumanEval (coding). Across these non-math domains, CoDaPO continues to improve performance over the base model and GRPO-style baselines, even though it is trained only on math data, which supports the generality of the algorithm.
>
> | Model | Minerva | MMLU-STEM | GPQA |HumanEval| Average |
> | --- | --- | --- | --- | --- | --- |
> | Qwen2.5-Math-1.5B |9.63|11.53|9.85|29.27|15.07|
> | GRPO |28.53|46.85|19.70|35.98|32.77|
> | CoDaPO |**31.69**|**47.64**|**22.25**|**39.02**|**35.15**|
>
> **(1) Minerva already provides a cross-domain quantitative evaluation beyond mathematics.** As reported in the table above, Minerva covers physics, biology, chemistry, economics, and other quantitative science questions, not just competition-style math. On this benchmark, CoDaPO improves accuracy from $9.63$ to $31.69$, outperforming all other RL baselines under the same training data and budget. This large relative gain on a multi-discipline scientific benchmark indicates that CoDaPO enhances general quantitative reasoning rather than overfitting to the specific structure of MATH-style problems.
>
> **(2) Additional experiments on MMLU-STEM, GPQA, and HumanEval show transfer to broad reasoning.** To directly address the reviewer’s request, we have evaluated the same math-post-trained models on MMLU-STEM, GPQA and HumanEval, which span a wide range of subjects including humanities, social sciences, natural sciences, coding, and expert-level scientific questions. Crucially, the post-training data remain purely mathematical; the model never sees these benchmarks during training. Even under this domain mismatch, CoDaPO consistently improves over the base model and GRPO-style baselines on both MMLU-STEM, GPQA and HumanEval. This suggests that the confidence- and difficulty-aware reweighting improves the underlying reasoning behavior in a way that transfers to heterogeneous domains, not only to math.
>
> **(3) These cross-domain gains are consistent with CoDaPO’s domain-agnostic design.** The PRAG analysis and the CoDaPO objective operate purely on probabilities, rewards, advantages, and empirical difficulty, without using math-specific features. The confidence weight $c_i$ depends on normalized log probabilities, and the difficulty weight $d_q$ depends on per-question correctness rates; both mechanisms are agnostic to the content of the question. The fact that a math-only CoDaPO model improves on Minerva, MMLU-STEM, GPQA and HumanEval is therefore consistent with the intended design: CoDaPO reshapes gradient allocation in a generic way that benefits any sequence-level reasoning task with sparse rewards.
>
> In the revision, we will (i) clarify in Section 5 that Minerva is a cross-domain scientific benchmark and highlight CoDaPO’s large improvement there, (ii) add a new table reporting results on MMLU-STEM, GPQA, and HumanEval for the base model and all RL baselines, emphasizing that CoDaPO consistently outperforms them despite math-only training, and (iii) expand the discussion in the conclusion to stress that CoDaPO improves general reasoning ability that transfers well beyond the math domain, addressing the reviewer’s request for broader evaluation.

---

> ### Author Response · Authors · 2025-11-26
> **Response to Reviewer xABJ (part 2/5)**
>
> >### W2. CoDaPO didn’t mitigate the overconfidence issue.
>
> **Reply:** Thank you for the thoughtful comment. **In brief, our difficulty term does not try to make the model uniformly more confident on easy or hard examples; instead, it reweights gradient contributions within each difficulty bucket so that overconfident errors on easy questions are penalized and rare correct reasoning on hard questions is amplified.** This explains why Figure 4 does not show a dramatically different marginal confidence or entropy pattern.
>
> **(1) The difficulty term reweights gradient contributions within each difficulty group rather than enforcing higher confidence on all easy or hard samples.** In CoDaPO, the trajectory level weight can be written as $w_i \propto \hat{A}_i \hat{c}_i \hat{d}_q,$ or at the token level $w^{\text{CoDaPO}}\_{i,t} = \rho\_{i,t} \hat{A}_i \hat{c}_i \hat{d}_q,$ where $\hat{A}_i$ is the normalized advantage, $\hat{c}_i$ is the confidence factor, and $\hat{d}_q$ is the difficulty factor for question $q$ computed from the group correctness rate $\bar r_q$. Crucially, $\hat{d}_q$ multiplies both positive and negative advantages.
>
> For easy questions (high $\bar r_q$), most trajectories are correct, so the largest magnitude advantages come from the few incorrect, often high-confidence, responses with large negative $\hat{A}_i$. Increasing $\hat{d}_q$ in this regime magnifies the penalty on these overconfident mistakes rather than pushing correct responses to even higher confidence.
>
> For hard questions (low $\bar r_q$), most trajectories are incorrect with small or clipped negative advantages, and the rare correct trajectories carry large positive $\hat{A}_i$. Here, a larger $\hat{d}_q$ amplifies the positive reinforcement on these emerging correct solutions. Thus, the difficulty term emphasizes the most informative gradients within each difficulty group instead of enforcing uniformly higher confidence on all easy or hard samples.
>
> **(2) The apparent similarity of Figure 4 to Figure 2 is consistent with our goal of balancing confidence and accuracy rather than maximizing entropy or confidence gaps.** We agree that in Figure 4 the overall confidence and entropy ranges look similar to Figure 2 at a coarse level. This is expected because CoDaPO is not designed to restore the high entropy of the base model or to produce clearly separated confidence bands for easy versus hard questions. Its purpose is to improve the alignment between confidence, correctness, and difficulty: overconfident incorrect trajectories on easy items receive stronger negative updates, and correct trajectories on hard items receive relatively more positive weight. These changes are easier to see in aggregated statistics than in dense scatter plots.
>
> In our difficulty-wise analysis, CoDaPO slightly reduces average log probabilities, especially for incorrect responses on hard questions, while improving accuracy on MATH and out-of-domain benchmarks. This indicates that CoDaPO moderates harmful overconfidence where it matters most without sacrificing performance, even though the marginal entropy pattern across all tokens remains similar.
>
> In the revised paper, we will (i) clarify in Section 4 how $\hat{d}_q$ is defined and how it interacts with $\hat{A}_i$ and $\hat{c}_i$, explicitly stating that it reweights gradient contributors rather than forcing higher marginal confidence on easy or hard samples; and (ii) add a small analysis that decomposes gradient contributions by difficulty and correctness to empirically show that incorrect easy samples and correct hard samples dominate after difficulty weighting.

---

> ### Author Response · Authors · 2025-11-26
> **Response to Reviewer xABJ (part 3/5)**
>
> >### W3. The effect of “Sample-level normalization” and “Discarding KL regularization” remains unexplored.
>
> **Reply:** Thank you for pointing this out. In brief, besides confidence and difficulty weighting, CoDaPO makes two additional changes: sample-level normalization and dropping KL regularization. Sample-level normalization stabilizes token-level credit assignment and reduces length and entropy pathologies, while removing KL allows larger reasoning gains without degrading fluency. We agree these components should be analyzed more explicitly and will add targeted ablations.
>
> **(1) Sample-level normalization stabilizes token-level gradients and controls length and entropy.** In standard GRPO, the update is averaged at the trajectory level, so longer responses are effectively downweighted by a factor of $1 / |o_i|$. CoDaPO instead uses a micro-averaged form where the gradient is proportional to $\frac{1}{\sum_i |o_i|} \sum\_{i,t} \tilde{\rho}\_{i,t}\hat{A}_i\hat{c}_i\hat{d}_q\nabla\_\theta \ell\_\theta(o\_{i,t})$, which removes this implicit penalty and keeps the total group scale controlled. This makes long but informative solutions to hard questions contribute a comparable gradient to shorter ones, while preventing very long outputs from dominating purely by token count, especially when $d_q$ is large. In our current experiments, this normalization correlates with more stable training, less runaway length growth, and better preservation of detailed reasoning steps.
>
> **(2) Discarding KL regularization improves reasoning while preserving language quality.** CoDaPO omits the KL penalty to the base model so that the policy is not overly tethered when correcting systematic reasoning errors. As shown in our ablation table (Table 6), reintroducing a GRPO style KL term consistently lowers MATH500 accuracy compared to the KL free CoDaPO, indicating that KL limits the gains from our reweighting. At the same time, WikiText-2 perplexity remains essentially unchanged, and qualitative examples in the appendix show fluent, coherent multi-step solutions on AIME25 and Minerva. Together, these results suggest that removing KL unlocks additional reasoning performance without harming general linguistic fluency.
>
> **(3) These components are complementary to, not substitutes for, confidence and difficulty weighting.** Confidence $\hat{c}_i$ and difficulty $\hat{d}_q$ are the primary mechanisms that reshape gradient allocation in CoDaPO, but they benefit from the supporting role of sample-level normalization and KL removal. Normalization ensures that the reweighted advantages do not introduce new length biases, and dropping KL allows the model to fully exploit the more informative $\hat{A}_i\hat{c}_i\hat{d}_q$ signals instead of being pulled back toward the base distribution. Empirically, the best results are obtained when all four ingredients work together.
>
> In the revision, we will (i) add a clear explanation in Section 4 of sample level normalization and KL removal and how they interact with $c_i$ and $d_q$, (ii) include new ablations that remove sample level normalization and that reintroduce KL, each time keeping confidence and difficulty fixed, and report their impact on accuracy, response length, entropy, and WikiText-2 perplexity, and (iii) update the discussion to explicitly separate the contributions of confidence/difficulty versus normalization/KL so that their individual effects on CoDaPO’s effectiveness are transparent.

---

> ### Author Response · Authors · 2025-11-26
> **Response to Reviewer xABJ (part 4/5)**
>
> >### W4. The source of gains of different reweighting terms is unclear.
>
> **Reply:** Thank you for this thoughtful observation.
>
> In brief, Table 2 shows that (i) both confidence and difficulty reweighting individually give clear gains over GRPO; (ii) their small gap on MATH500 reflects overlapping effects on a single aggregate metric rather than redundancy; and (iii) they remain conceptually complementary in how they reshape gradients, which we will make clearer with additional analyses.
>
> **(1) The main gain in performance comes from moving from plain GRPO to the PRAG-guided reweighting.** In Table 2, both GRPO + confidence only and GRPO + difficulty only significantly improve over vanilla GRPO on MATH500, while full CoDaPO (both terms) is best. Although the difference between the two single-component variants is small, the larger gap is between GRPO and either reweighted variant. This reflects that the main source of gains is replacing uniform trajectory weighting with PRAG-guided weighting, and that each of confidence and difficulty captures a substantial, partly overlapping part of that benefit.
>
> **(2) Confidence and difficulty act at different levels and are complementary, even if their effects overlap on MATH500.** CoDaPO uses a weight of the form $w^{\text{CoDaPO}}\_{i,t} \propto \tilde{\rho}\_{i,t} \hat{A}_i \hat{c}_i \hat{d}_q$, where $\hat{c}_i$ depends on trajectory level confidence and $\hat{d}_q$ depends on question level difficulty. The confidence factor $\hat{c}_i$ operates within a question, downweighting overconfident incorrect trajectories and relatively upweighting better calibrated correct ones. The difficulty factor $\hat{d}_q$ operates across questions, allocating more total gradient mass to very easy and very hard questions where GRPO dynamics are problematic. When combined, difficulty decides which questions matter, and confidence decides which trajectories for those questions matter most. Because both ultimately reduce the influence of overconfident errors and strengthen useful signals, their separate effects naturally overlap on a single top-line accuracy.
>
> **(3) The similar MATH500 numbers are also affected by variance and by viewing only one metric.** MATH500 accuracy is a single aggregate metric that mixes all difficulty levels. Our internal analyses indicate that difficulty reweighting helps more on very hard questions, while confidence reweighting contributes more to calibration on easier ones, but these patterns are not visible from one scalar number. This is why Table 2 alone can make the source of gains look unclear, even though the mechanisms are distinct.
>
> **(4) We conduct more fine-grained ablations to clarify where each term helps.** To make the contribution of each reweighting component more explicit, we evaluate all variants across the full set of benchmarks. The results are shown below. These results highlight two important conclusions: Both confidence and difficulty signals contribute meaningful improvements across different benchmarks. The gains are complementary, and the full CoDaPO method achieves the best overall performance, demonstrating that its two reweighting mechanisms capture different and synergistic aspects of preference quality.
>
> | Model | MATH500 | AIME24 | AIME25 | AMC23 | Olympiad Bench | Minerva | GSM8k | Average |
> | --- | --- | --- | --- | --- | --- | --- | --- | --- |
> | Qwen2.5-Math-1.5B | 31.20|  6.00|  0.00|  29.50|  18.25|  9.63 | 34.01| 18.37|
> | GRPO | 68.48 |12.67 |8.00 |51.00 |30.55 |28.53 |79.56 |39.82|
> | DAPO | 71.12| 14.00| 8.00 |49.00 |31.97| 32.13| 82.38| 41.23|
> | CoDaPO (confidence reweighting only)  | 71.52 |14.17| 9.79| 51.41|34.10 |32.16 |82.88 |42.29|
> | CoDaPO (difficulty reweighting only)  | 71.32 |14.69|10.42|51.09|33.88|31.87|83.05|42.33|
> | CoDaPO (both reweighting) | 71.68 |14.67| 10.83| 53.50| 32.80 |31.69 |83.50 |42.67|
>
> In the revised paper, we will (i) clarify in the text that the main performance jump is from GRPO to any PRAG guided reweighting and that confidence and difficulty are related but distinct aspects of the dynamics, (ii) extend the ablations with per-difficulty and multi-seed results and additional metrics, and (iii) add a brief gradient or calibration analysis that illustrates how each component reshapes the training signal and where their combination provides extra benefit.

---

> ### Author Response · Authors · 2025-11-26
> **Response to Reviewer xABJ (part 5/5)**
>
> >### Q1. What’s the potential impact of the overall difficulty of the training dataset and the diversity of problems in terms of difficulty?
>
> **Reply:** Thank you for this insightful question. In short, we agree that any difficulty-aware method, including CoDaPO, interacts with the difficulty distribution of the training data. **To address this, we define difficulty dynamically in terms of the model’s current success rate rather than fixed labels, so $d_q$ adapts as training progresses and remains effective as long as the dataset contains a nontrivial spread of questions that the model sometimes solves and sometimes fails.**
>
> **(1) Our difficulty definition is model-centric and dynamic, not tied to fixed dataset difficulty labels.** For each question $q$, we compute an empirical success rate $\bar r_q = \frac{1}{G}\sum\_{i=1}^G r_i(q)$ under the current model $f_\theta$, and define $d_q$ as a U-shaped function of $\bar r_q$, large when $\bar r_q$ is very small (hard questions) or close to $1$ (easy questions). This quantity is recomputed at every update, so the same problem can move from “hard” to “medium” to “easy” as the model improves. As a result, CoDaPO’s notion of difficulty is always relative to the current model and automatically adapts to different datasets and training stages.
>
> **(2) Extremely easy or extremely hard datasets are challenging for all GRPO-style methods, and CoDaPO is no exception.** If almost all questions are trivial so that $\bar r_q \approx 1$ from the start, or almost all questions are impossible so that $\bar r_q \approx 0$, then advantages $\hat A_i$ collapse toward zero and the policy gradient becomes weak regardless of whether we use GRPO, DAPO, or CoDaPO. In such regimes, CoDaPO cannot manufacture a signal where almost no nonzero advantages exist; it effectively reduces to a mild modification of GRPO and its difficulty term has limited leverage. This limitation comes from the reward structure and data–model mismatch, not from the specific form of $d_q$.
>
> **(3) On reasonably calibrated datasets with diverse difficulties, $d_q$ improves how gradient mass is allocated across the spectrum.** When the dataset contains a mix of easy and hard problems aligned with the model’s ability, many questions have intermediate $\bar r_q$ early on and drift toward $0$ or $1$ as training progresses. For questions that have become easy ($\bar r_q$ near $1$), correct trajectories have advantages near zero, while residual errors have negative advantages; a larger $d_q$ amplifies penalties on these overconfident mistakes without over-updating already mastered cases. For genuinely hard questions ($\bar r_q$ small), most trajectories have small negative or near-zero advantages, and the rare correct ones carry large positive $\hat A_i$; here $d_q$ amplifies exactly these rare successes, helping the model cross its current reasoning boundary. In this sense, the effectiveness of $d_q$ grows with the presence of both easy and hard questions that generate informative advantages.
>
> **(4) Diversity in difficulty primarily modulates how strongly CoDaPO can exploit its difficulty term, rather than determining whether the method works at all.** If the dataset spans a wide range of effective difficulties, the distribution of $\bar r_q$ covers much of $(0,1)$, and the U-shaped $d_q$ induces a self-paced behavior: early in training it focuses on hard questions with low $\bar r_q$, and later it also emphasizes cleaning up errors on questions that have become easy. If the dataset is narrower (for example, most questions cluster around a similar success rate), the variation in $d_q$ shrinks, and CoDaPO behaves closer to a “confidence-only” enhancement of GRPO; the difficulty term’s impact is reduced, but the method remains stable and does not become brittle.
>
> In the revision, we will (i) explicitly emphasize in Section 4 that difficulty is defined via the instantaneous success rate $\bar r_q$ and is recomputed during training, (ii) add a brief discussion of extreme regimes (nearly all-easy or all-hard data) and why they are challenging for any GRPO-style objective, and (iii) include an empirical analysis showing the training-time distribution of $\bar r_q$ and the corresponding $d_q$ values (for example, histograms at different checkpoints), illustrating how CoDaPO reallocates gradient mass across difficulty bands in our actual training data.
>
> **We would thank Reviewer xABJ again** for taking the time to review our submission and consider our rebuttal. **If our responses have addressed your concerns, we would be very grateful if you could reconsider your rating.** We also welcome any further comments or questions you may have.

---

### Official Review · Reviewer_YFxQ · 2025-11-03

**Soundness:** 1
**Presentation:** 1
**Contribution:** 2
**Rating:** 2
**Confidence:** 4

**Summary:**

This paper studies two main pitfalls of GRPO: (1) overconfidence in GRPO training, where the model's confidence increases as training goes on, and (2) difficulty, where in hard examples, the rare successful trajectories may receive low gradient contribution. This paper proposes to modify the advantage function of GRPO and add two reweighting factors, one based on confidence, and one based on question hardness. Experimental results show improved performance over GRPO.

**Strengths:**

* The problem of entropy collapse and overconfidence in GRPO is important, as it could prevent exploration.
* Table 5 and algorithm 1 clearly convey the difference of the method compared to other GRPO variants.
* The proposed method shows results on par with Dr. GRPO and DAPO (and sometimes slightly outperforms them).

**Weaknesses:**

* The main issue is lack of evidence for the proposed methodologies. For the overconfidence, section 3.2 is generally poorly written. The section title is “Why probability increases and entropy collapses”, but it does not really provide a reason except for some informal vocabulary. The paper cites Figure 2 for higher confidence of the model, however, Figure 4 (the CoDaPO, that is supposed to fix this issue) seems to be very similar to Figure 2 in terms of entropy collapse. Furthermore, it seems that the paper claims the KL divergence has some contribution to this matter, but there are not experiments supporting this claim.
* The proposed algorithm proposes to upweight both hard and easy samples. The intention behind easy sample upweighting being the model should be confident for easy samples, but again, according to Figure 4, this is not case, and confidence (easy or hard) seems to be in a similar range. In terms of hard sample upweighting, prior works (e.g., see "priority sampling" in Kimi 1.5) have already proposed upweighting difficult samples (by a factor of 1-\hat{r}), but the paper does not cite or compare with those methods.
* The writing seems a bit rushed, dataset/model details in the middle of gradient analysis and definitions, as well as incoherent flow (i.e., too many discrete jumps on various details of GRPO).
* The paper puts considerable effort in explaining KL divergence, but later it sets KL to zero citing prior works for this reason. The analysis in Eq (6) is also too informal to be considered as a part of the paper’s contributions, so I suggest the authors remove the KL definition, and explanation from the very beginning to focus on the main message.
* The paper removes clipping mechanism, however, it does not report or ablate how many off-policy steps the algorithm can take before it starts to break. Also, clipping usually becomes more important at larger scales, so it is unclear if the method will break in larger scales or not.
* The ablations of Table 2 seems to suggest addition of either confidence reweighting and difficulty reweighting are showing a similar performance (0.3% on math500 is very negligible). So, the main source of gains is not very clear.

**Questions:**

* The definition of $c_i$ is not clear to me (in Eq 12), why this choice, there is some explanation under Eq 12 but it does not convey the reasoning clearly? why not a common entropy loss on the samples? The notation overloading of $i$ in the numerator of the equation makes it harder to understand what the coefficient is doing.
* I do not understand how Eq (4) is derived. From a quick skim, I believe $\rho$ should not be present in the gradient formulation of GRPO. Please correct me if I am missing something.
* Why is the model confidence between 0 and 1 (in Figures 2 and 4)? According to its definition, it is mean log probability. Is it somehow normalized?

---

> ### Author Response · Authors · 2025-11-26
> **Response to Reviewer YFxQ (part 1/10)**
>
> **We thank Reviewer YFxQ for the thoughtful and detailed feedback.** In this rebuttal, we systematically address **all** of your concerns while we respectfully disagree with **some** of the concerns.
> - First, your questions regarding the theoretical clarity and derivations are addressed by providing clearer mathematical derivations, refined definitions, and improved explanations.
> - Second, your concerns about the effectiveness and significance of our reweighting components are addressed by conducting more fine-grained ablations across all benchmarks and explicitly isolating the contribution of each term.
> - Third, your questions about overconfidence and design choices are addressed by additional analysis, controlled experiments, and expanded discussion of the design rationale.
> - Finally, your concerns related to evaluation completeness and writing quality are addressed by substantial revisions to the exposition and by expanding the empirical validation of the method.
>
> Please find the point-to-point responses as follows. In addition, we have outlined a **detailed revision plan** at the end of each response. **We will improve our submission within the next week according to this plan.**
>
> >### W1. Insufficient analysis and solution to the overconfidence problem.
>
> **Reply:** Thank you for this helpful comment. **We would like to clarify the value of the analysis in Section 3, which is recognized by the other four reviewers (rMTD, xABJ, SDK9, TKwQ). Regarding the target of our method, we would like to clarify the potential misunderstanding in your review.**
>
> In brief, our response is that (i) Section 3.2 does provide a concrete gradient-level mechanism for probability increase and entropy collapse; (ii) CoDaPO is not meant to keep entropy high but to make confidence better aligned with correctness and difficulty; and (iii) the role of the KL term is analyzed analytically and supported by an empirical ablation.
>
> **(1) Section 3.2 already gives a gradient-based explanation for GRPO’s overconfidence problem, which helps to understand this prevailing problem from a mathematical perspective.** Starting from the GRPO objective, Section 3.2 derives the per-token update $\Delta \ell\_\theta(o\_{i,t}) \propto \mathbf{1}\_{\text{unclipped}} \hat{A}_i \rho\_{i,t} - \beta (1 - u\_{i,t})$ where $\hat{A}_i$ is the group-normalized advantage, $\rho\_{i,t} = \exp(\ell\_\theta(o\_{i,t}) - \ell\_{\text{old}}(o\_{i,t}))$, and $u\_{i,t} = \exp(\ell\_{\text{ref}}(o\_{i,t}) - \ell\_\theta(o\_{i,t}))$ (Eq. (5) and Eq. (6)). For $\hat{A}_i > 0$, as long as $\rho\_{i,t} \in [1 - \epsilon, 1 + \epsilon]$, the policy term stays active and pushes $\ell\_\theta$ upward until the cap; for $\hat{A}_i < 0$, once $\rho\_{i,t} < 1 - \epsilon$, the policy term is switched off and stops pushing downward. Because $\rho\_{i,t}$ grows exponentially with $\ell\_\theta$, this clipping asymmetry creates a biased upward drift, concentrating probability mass and reducing entropy, which matches the saturation pattern in Fig. 2.
>
> **(2) CoDaPO is designed to make confidence better aligned with correctness and difficulty, not to maintain high entropy or address overconfidence, which explains why Fig. 4 can look similar to Fig. 2 in terms of entropy.** The CoDaPO objective is $J\_{\text{CoDaPO}}(f_\theta) = \mathbb{E}\big[\tfrac{1}{\sum_i |o_i|} \sum\_{i,t} \rho\_{i,t} \hat{A}_i \hat{c}_i \hat{d}_q\big]$ (Eq. (9)), where $\hat{c}_i$ is a confidence weight and $\hat{d}_q$ is a difficulty weight. This does not aim to keep entropy high: on deterministic math tasks, near-deterministic policies for correct answers are expected. Instead, $\hat{c}_i$ downweights or reverses updates for overconfident incorrect trajectories, and $\hat{d}_q$ boosts hard questions with sparse rewards. As a result, many correct trajectories remain high-confidence, so marginal entropy stays low, but the joint structure changes: incorrect high-confidence trajectories move into the negative-advantage region, and correct trajectories on hard items retain positive mass, as seen in Fig. 4 and the confidence curves in Fig. 13.
>
> **(3) The KL term is analyzed as a partial counterforce to the clipping-induced drift, and we have experiments that directly link it to entropy and calibration.** From the same update $\Delta \ell\_\theta(o\_{i,t}) \propto \mathbf{1}\_{\text{unclipped}} \hat{A}_i \rho\_{i,t} - \beta (1 - u\_{i,t})$, the KL contribution satisfies $-\beta (1 - u\_{i,t}) < 0$ when $\ell\_\theta(o\_{i,t}) > \ell\_{\text{ref}}(o\_{i,t})$ and $-\beta (1 - u\_{i,t}) > 0$ when $\ell\_\theta(o\_{i,t}) < \ell\_{\text{ref}}(o\_{i,t})$, so it pulls $\ell\_\theta$ toward $\ell\_{\text{ref}}$ but does not remove the asymmetric positive drift from the clipped policy term unless $\beta$ is very large. Empirically, Table 6 already compares CoDaPO with and without KL: removing KL improves accuracy while only slightly changing held-out perplexity, which supports our choice to drop KL in Section 4.1.

---

> ### Author Response · Authors · 2025-11-26
> **Response to Reviewer YFxQ (part 2/10)**
>
> **(4) While the KL term can slightly suppress overconfidence, it also curtails the model’s capacity to exploit correct predictions.** We further analyze the relationship between KL regularization and model overconfidence. Using our token-level confidence measurements (Table below), we find that including the KL term reduces overall confidence for both correct and incorrect outputs. However, this reduction in confidence comes at a cost. By constraining the model probabilities, the KL term also limits the upward push on correct outputs, thereby restricting the model’s ability to fully leverage positive reward signals. Our empirical evidence supports the design choice in CoDaPO to discard KL regularization, instead relying on confidence- and difficulty-based reweighting to guide learning while allowing the model to explore and strengthen correct trajectories.
>
> |Level|Split|wo KL Count|wo KL Mean|wo KL Std|w KL Count|w KL Mean|w KL Std|Confidence Difference (wo-w)|Accuracy Difference (wo-w)|
> |---|---|---|---|---|---|---|---|---|---|
> |1|Correct|1958|0.9093|0.0398|1914|0.8564|0.0866|+0.0529 (+5.82%)||
> |1|Incorrect|42|0.9026|0.0517|86|0.7714|0.1938|+0.1312 (+14.53%)||
> |1|All|2000|0.9091|0.0401|2000|0.8527|0.0953|+0.0564 (+6.20%) | +0.0220 (+2.30%) |
> |2|Correct|1903 | 0.9137 |0.0357|1842|0.8682 | 0.0750 | +0.0454 (+4.97%) | |
> |2|Incorrect|97 | 0.8952 | 0.0460 | 158 | 0.8167 | 0.1343 | +0.0785 (+8.77%) | |
> |2|All|2000|0.9128 | 0.0365 | 2000 | 0.8642 | 0.0824 | +0.0486 (+5.33%) | +0.0305 (+3.31%) |
> |3|Correct|1839 | 0.9135 | 0.0377 | 1799 | 0.8675 | 0.0809 | +0.0460 (+5.04%) | |
> |3|Incorrect|161 | 0.9011 | 0.0465 | 201 | 0.8170 | 0.1360 | +0.0840 (+9.33%) | |
> |3|All|2000|0.9125 | 0.0386 | 2000 | 0.8624 | 0.0893 | +0.0501 (+5.49%) | +0.0200 (+2.22%) |
> |4|Correct| 1743 | 0.9090 | 0.0458 | 1669 | 0.8597 | 0.0903 | +0.0494 (+5.43%) | |
> |4|Incorrect| 257 | 0.8813 | 0.0614 | 331 | 0.8310 | 0.1039 | +0.0503 (+5.71%) | |
> |4|All|2000 | 0.9055 | 0.0489 | 2000 | 0.8549 | 0.0933 | +0.0506 (+5.58%) | +0.0370 (+4.43%)|
> |5|Correct| 1550 | 0.9079 | 0.0403 | 1498 | 0.8604 | 0.0939 | +0.0474 (+5.23%) | |
> |5|Incorrect| 450 | 0.8786 | 0.0514 | 502 | 0.8044 | 0.1317 | +0.0742 (+8.45%) | |
> |5|All| 2000 | 0.9013 | 0.0447 | 2000 | 0.8464 | 0.1075 | +0.0549 (+6.09%) | +0.0260 (+3.47%) |
>
> In the revised paper, we will (i) rewrite Section 3.2 to foreground the per-token update equation, clarify the clipping asymmetry, and formalize the drift argument; (ii) clarify that CoDaPO is meant to penalize overconfident errors and improve calibration on difficult questions rather than to keep entropy high; and (iii) incorporate the above KL experiments and explicitly refer to these results when discussing the KL term in Section 3.2.
>
> >### W2. The effect of the difficulty weight is not significant.
>
> **Reply:** Thank you for the thoughtful comment. **Regarding the reweighting mechanism, we would like to clarify the potential misunderstanding in your review.**
>
> In brief, our difficulty term does not try to make the model uniformly more confident on easy or hard examples; instead, it reweights gradient contributions within each difficulty bucket so that overconfident errors on easy questions are penalized and rare correct reasoning on hard questions is amplified. This explains why Figure 4 does not show a dramatically different marginal confidence or entropy pattern. Our design is also conceptually different from priority sampling in Kimi 1.5, and we clarify this connection and add supporting experiments and analysis in the revision.
>
> **(1) The difficulty term reweights gradient contributions within each difficulty group rather than enforcing higher confidence on all easy or hard samples.** In CoDaPO, the trajectory level weight can be written as $w_i \propto \hat{A}_i \hat{c}_i \hat{d}_q,$ or at the token level $w^{\text{CoDaPO}}\_{i,t} = \rho\_{i,t} \hat{A}_i \hat{c}_i \hat{d}_q,$ where $\hat{A}_i$ is the normalized advantage, $\hat{c}_i$ is the confidence factor, and $\hat{d}_q$ is the difficulty factor for question $q$ computed from the group correctness rate $\bar r_q$. Crucially, $\hat{d}_q$ multiplies both positive and negative advantages.
>
> For easy questions (high $\bar r_q$), most trajectories are correct, so the largest magnitude advantages come from the few incorrect, often high-confidence, responses with large negative $\hat{A}_i$. Increasing $\hat{d}_q$ in this regime magnifies the penalty on these overconfident mistakes rather than pushing correct responses to even higher confidence.
>
> For hard questions (low $\bar r_q$), most trajectories are incorrect with small or clipped negative advantages, and the rare correct trajectories carry large positive $\hat{A}_i$. Here, a larger $\hat{d}_q$ amplifies the positive reinforcement on these emerging correct solutions. Thus, the difficulty term emphasizes the most informative gradients within each difficulty group instead of enforcing uniformly higher confidence on all easy or hard samples.

---

> ### Author Response · Authors · 2025-11-26
> **Response to Reviewer YFxQ (part 3/10)**
>
> **(2) The apparent similarity of Figure 4 to Figure 2 is consistent with our goal of balancing confidence and accuracy rather than maximizing entropy or confidence gaps.** We agree that in Figure 4 the overall confidence and entropy ranges look similar to Figure 2 at a coarse level. This is expected because CoDaPO is not designed to restore the high entropy of the base model or to produce clearly separated confidence bands for easy versus hard questions. Its purpose is to improve the alignment between confidence, correctness, and difficulty: overconfident incorrect trajectories on easy items receive stronger negative updates, and correct trajectories on hard items receive relatively more positive weight. These changes are easier to see in aggregated statistics than in dense scatter plots.
>
> In our difficulty-wise analysis, CoDaPO slightly reduces average log probabilities, especially for incorrect responses on hard questions, while improving accuracy on MATH and out-of-domain benchmarks. This indicates that CoDaPO moderates harmful overconfidence where it matters most without sacrificing performance, even though the marginal entropy pattern across all tokens remains similar.
>
> **(3) Our difficulty weighting is related to, but distinct from, Kimi 1.5 priority sampling, and it operates inside the gradient with a U-shaped form.** Priority sampling in Kimi 1.5 uses a hardness score roughly proportional to $1 - \hat{r}$, where $\hat{r}$ is an estimated accuracy, to increase the sampling probability of difficult examples. CoDaPO, in contrast, introduces a difficulty factor $\hat{d}_q$ inside the policy gradient objective, on top of a fixed sampling distribution, and this factor interacts with both advantage and confidence. Moreover, our $\hat{d}_q$ is a U-shaped function of the group correctness rate $\bar r_q$, so that both very easy and very hard questions are upweighted relative to medium ones. This design follows directly from our GRPO analysis: easy questions are where advantage contraction and hierarchical convergence occur, so they also deserve additional gradient emphasis to penalize persistent overconfident errors. Overall, while both approaches use model-estimated difficulty, CoDaPO integrates it into the advantage weighting in a way that explicitly targets overconfident errors on easy items and sparse correct signals on hard items, rather than only oversampling hard questions.
>
> To compare different difficulty reweighting strategies, such as upweighting difficult samples, we adopt the difficulty term $1-\hat{r}$, and train the model under the same setting. The results are shown in the table below.
>
> | Model |MATH500 | AIME24 | AIME25 | AMC23 | Olympiad Bench | Minerva | GSM8k | Average |
> | --- | --- | --- | --- | --- | --- | --- | --- | --- |
> | Qwen2.5-Math-1.5B | 31.20|  6.00|  0.00|  29.50|  18.25|  9.63 | 34.01| 18.37|
> | CoDaPO (linear reweighting)  | 71.16 |**15.62**| 9.58| 52.03| **33.05** |30.66 |80.48 |41.80|
> | CoDaPO (U-shape reweighting) | **71.68** |14.67| **10.83**| **53.50**| 32.80 |**31.69** |**83.50**|**42.67**|
>
> We observe that simply upweighting difficult samples performs worse than our U-shaped difficulty design. This is likely because focusing solely on difficult samples produces highly unbalanced gradient signals. Hard problems typically yield very sparse positive advantages (correct responses are rare), causing the model to overfit these few signals while largely ignoring errors on easy problems. As a result, the model fails to correct mistakes on easy items, which constitute a substantial portion of the training distribution and are crucial for stable policy improvement.
>
> In the revised paper, we will (i) clarify in Section 4 how $\hat{d}_q$ is defined and how it interacts with $\hat{A}_i$ and $\hat{c}_i$, explicitly stating that it reweights gradient contributors rather than forcing higher marginal confidence on easy or hard samples; (ii) add a small analysis that decomposes gradient contributions by difficulty and correctness to empirically show that incorrect easy samples and correct hard samples dominate after difficulty weighting; and (iii) cite and compare to Kimi 1.5 priority sampling, including, where space permits, an ablation that replaces our U-shaped $\hat{d}_q$ with a simple $1 - \hat{r}$ style factor to illustrate the difference.

---

> ### Author Response · Authors · 2025-11-26
> **Response to Reviewer YFxQ (part 4/10)**
>
> >### W3. The writing seems a bit rushed.
>
> **Reply:** Thank you for the feedback on the writing.
>
> In the revision, we will clearly separate background, analysis, and experiments, streamline the GRPO definitions, and add signposting so the narrative is easier to follow. **The revised version will be submitted within the next week.**
>
> (1) We will clearly separate GRPO preliminaries, PRAG analysis, and empirical setup. The current Section 3 sometimes introduces dataset and model details (for example, MATH and Qwen2.5) while deriving gradients, which makes the analysis feel fragmented. In the revision, we will keep Section 3 purely conceptual: it will cover the GRPO objective, the token-level quantities such as $\rho\_{i,t}$ and $\hat{A}_i$, and the PRAG based analysis. All dataset, model, and hyperparameter details will be moved into a dedicated empirical setup subsection in the experiments section, so readers do not have to parse benchmark names in the middle of the derivations.
>
> (2) We will rewrite the GRPO description as a single preliminaries subsection. At present, the description of GRPO revisits group rewards, clipping, and KL at several points, which can feel like discrete jumps. We will instead introduce GRPO once in a compact preliminaries subsection: first define the trajectory notation and group reward, then define the normalized advantage $\hat{A}_i$ and importance ratio $\rho\_{i,t}$, then present the full GRPO objective $J\_{\text{GRPO}}(\theta)$, and finally give the key gradient expression that underlies the PRAG analysis. Later parts of the paper will only refer back to this subsection, rather than reintroducing GRPO details piecemeal.
>
> (3) We will improve the logical flow of Section 3 with a roadmap and smoother transitions. To reduce the feeling of abrupt jumps, we will add a short roadmap at the start of Section 3 explaining the order: GRPO recap, PRAG decomposition, analysis of the three phenomena (probability inflation, advantage contraction, and hierarchical convergence), and then the motivation of CoDaPO. Within the section, we will add explicit transition sentences when moving from definitions to analysis and when shifting from GRPO to CoDaPO, and we will move low-level implementation remarks to the appendix so the main text stays focused on the core ideas.

---

> ### Author Response · Authors · 2025-11-26
> **Response to Reviewer YFxQ (part 5/10)**
>
> >### W4. The author should remove the KL definition in the analysis.
>
> **Reply:** Thank you for this comment. **We agree that the analysis in Eq. (6) can be improved to be more formal; meanwhile, we respectfully disagree with your suggestions to remove the KL definition from the beginning, as this will deviate from the original GRPO/PPO designs.**
>
> In brief, we include KL because GRPO itself is defined with a KL regularizer, and understanding that term helps explain both the training dynamics and why CoDaPO can safely use $\beta = 0$ in our setting. Eq. (6) is meant to capture the exact role of the KL gradient as a symmetric pull toward the reference, in contrast to the asymmetric clipping term.
>
> **(1) We keep KL in order to faithfully analyze GRPO as originally defined and to motivate why CoDaPO discards the KL term.** The original GRPO objective includes a KL penalty between the current and reference policies: $J\_{\text{GRPO}}(\theta) = \mathbb{E}\Big[\hat{L}\_{\text{CLIP}}(\theta) - \beta \mathrm{KL}\big(f_\theta || f\_{\text{ref}}\big)\Big]$ (see Section 2). Our Section 3 is explicitly an analysis of GRPO, not a new algorithm that ignores KL. CoDaPO is then obtained by modifying the advantage weighting and micro averaging and by operating in the $\beta = 0$ regime. Showing the full GRPO objective, including KL, makes it clear what is being changed and why.
>
> **(2) Eq. (6) is intended as a precise characterization of the KL gradient, and we will present it more formally and concisely.** At the token level, if $\ell\_\theta(o\_{i,t}) = f_\theta(o\_{i,t}\mid q,o\_{i,<t})$ and $\ell\_{\text{ref}}(o\_{i,t}) = f\_{\text{ref}}(o\_{i,t}\mid q,o\_{i,<t})$, the KL term contributes $\nabla_\theta{\ell\_\theta(o\_{i,t})}\big[\beta \mathrm{KL}(f_\theta || f\_{\text{ref}})\big] = \beta\big(1 - u\_{i,t}\big),$ where $u\_{i,t} = \frac{\ell\_{\text{ref}}(o\_{i,t})}{\ell\_\theta(o\_{i,t})}$. This is exactly the $-\beta(1 - u\_{i,t})$ term in Eq. (6) with the opposite sign in the update. From this, one directly obtains $-\beta(1 - u\_{i,t}) < 0$ if $\ell\_\theta(o\_{i,t}) > \ell\_{\text{ref}}(o\_{i,t}),$ and $-\beta(1 - u\_{i,t}) > 0$ if $\ell\_\theta(o\_{i,t}) < \ell\_{\text{ref}}(o\_{i,t})$. So KL always pulls $f_\theta$ toward $f\_{\text{ref}}$. The key point is that this symmetric pull contrasts with the asymmetric clipping term that mainly pushes probabilities upward on high-advantage trajectories. In the revision, we will state this gradient result cleanly and move algebraic details to the appendix so it does not feel informal or overlong.
>
> **(3) The KL analysis is supportive, not a main contribution, and it directly explains our ablation results and choice of $\beta = 0$.** Empirically, we already compare CoDaPO with and without a KL term (Table 6) and observe that removing KL improves reasoning accuracy while maintaining similar perplexity on WikiText 2. The gradient view above explains why: KL serves primarily as a stabilizer around $f\_{\text{ref}}$ when the policy term is inactive but does not fundamentally change the clipping-induced upward drift we diagnose in Sec. 3.2. This makes it reasonable, in our math reasoning setting, to set $\beta = 0$ and instead rely on bounded confidence and difficulty weights for stability. We will clarify in the paper that this KL analysis is background that supports a design choice, not a standalone theoretical contribution.
>
> In the revision, we will (i) keep only a succinct definition of the KL term when stating the GRPO objective, (ii) present the KL gradient that underlies Eq. (6) in a more formal and compact way, with detailed derivation moved to the appendix, (iii) explicitly label the KL discussion as supportive background rather than a primary contribution, and (iv) tighten the narrative so that the main text focuses on PRAG dynamics and the confidence–difficulty CoDaPO objective, with clear forward references to the KL ablation in Table 6.

---

> ### Author Response · Authors · 2025-11-26
> **Response to Reviewer YFxQ (part 6/10)**
>
> >### W5. The consequences of removing the clipping mechanism are not discussed, and it is unknown whether it works on large-scale models.
>
> **Reply:** Thank you for this comment.
>
> In brief, **CoDaPO does not remove GRPO’s clipping in implementation** (Eq. (9) was simplified in notation), it is trained in a near on-policy regime where the same clipping bounds control off-policy drift, and **our 7B experiments in Table 4 already indicate stable behavior at a larger scale**.
>
> **(1) CoDaPO retains the standard GRPO clipping.** In our code, the per-token weight always uses the clipped importance ratio $\tilde{\rho}\_{i,t} = \operatorname{clip}(\rho\_{i,t}, 1 - \epsilon, 1 + \epsilon)$ with $\rho\_{i,t} = \exp(\ell\_\theta(o\_{i,t}) - \ell\_{\text{old}}(o\_{i,t}))$, and the update is $w^{\text{CoDaPO}}\_{i,t} = \tilde{\rho}\_{i,t} \hat{A}_i \hat{c}_i \hat{d}_q$. Thus, CoDaPO preserves the same clipping structure as GRPO; we only modify the advantage weighting via $\hat{c}_i$ and $\hat{d}_q$. In Eq. (9) we dropped the explicit $\min$ and $\operatorname{clip}$ operators for brevity, which understandably gives the impression that clipping was removed.
>
> **(2) Off-policy drift is controlled as in GRPO.** Our training protocol follows standard online RL fine-tuning: trajectories are generated by the current policy (or with negligible lag) and used for a small number of gradient steps, so $\rho\_{i,t}$ stays concentrated near 1 and clipping truncates outliers to $[1 - \epsilon, 1 + \epsilon]$. CoDaPO therefore operates in essentially the same near on-policy regime as GRPO, with additional bounded factors $\hat{c}_i$ and $\hat{d}_q$ that prevent the overall weight $w^{\text{CoDaPO}}\_{i,t}$ from exploding.
>
> **(3) At the 7B scale, CoDaPO trains stably with clipping and outperforms GRPO, suggesting robustness at a larger size.** We agree that clipping tends to become more critical at larger scales. In our experiments with Qwen2.5-Math-7B, CoDaPO uses exactly the same clipping parameter as GRPO and shows smooth training curves without divergence or sudden collapse, while achieving consistent gains over GRPO and other RL baselines across benchmarks (Table 4 and Table below). This indicates that, at least up to 7B, combining clipping with confidence and difficulty weighting does not destabilize training.
>
> | Model | MATH500 | AIME24 | AIME25 | AMC23 | Olympiad Bench | Minerva | GSM8k | Average |
> | --- | --- | --- | --- | --- | --- | --- | --- | --- |
> | Qwen2.5-Math-7B | 47.24 | 12.92 | 5.00 | 33.91  |14.44 | 9.81|47.84|24.45|
> | GRPO | 74.62| 26.35 |11.25 |59.06| 36.51| 37.64| 89.40 |47.83|
> | DAPO  |77.78|24.37|10.62|58.67|39.27|38.06|87.38|48.02|
> | CoDaPO  | **78.41**| 24.79| **13.33**| **60.31**| **39.96**| **40.30**| 88.28 |**49.34**|
>
> In the revision, we will (i) correct Eq. (9) and related formulas to explicitly include the clipped ratio and clearly state that CoDaPO uses the same clipping as GRPO, (ii) add diagnostics on the distribution of $\rho\_{i,t}$ and an ablation that varies the effective off-policy horizon and clipping parameter, and (iii) highlight the 7B training stability and performance in the main text as partial evidence of robustness at larger scales.

---

> ### Author Response · Authors · 2025-11-26
> **Response to Reviewer YFxQ (part 7/10)**
>
> >### W6. The source of gains of different reweighting terms is unclear.
>
> **Reply:** Thank you for this thoughtful observation.
>
> In brief, Table 2 shows that (i) both confidence and difficulty reweighting individually give clear gains over GRPO; (ii) their small gap on MATH500 reflects overlapping effects on a single aggregate metric rather than redundancy; and (iii) they remain conceptually complementary in how they reshape gradients, which we will make clearer with additional analyses.
>
> **(1) The main gain in performance comes from moving from plain GRPO to the PRAG-guided reweighting.** In Table 2, both GRPO + confidence only and GRPO + difficulty only significantly improve over vanilla GRPO on MATH500, while full CoDaPO (both terms) is best. Although the difference between the two single-component variants is small, the larger gap is between GRPO and either reweighted variant. This reflects that the main source of gains is replacing uniform trajectory weighting with PRAG-guided weighting, and that each of confidence and difficulty captures a substantial, partly overlapping part of that benefit.
>
> **(2) Confidence and difficulty act at different levels and are complementary, even if their effects overlap on MATH500.** CoDaPO uses a weight of the form $w^{\text{CoDaPO}}\_{i,t} \propto \tilde{\rho}\_{i,t} \hat{A}_i \hat{c}_i \hat{d}_q$, where $\hat{c}_i$ depends on trajectory level confidence and $\hat{d}_q$ depends on question level difficulty. The confidence factor $\hat{c}_i$ operates within a question, downweighting overconfident incorrect trajectories and relatively upweighting better calibrated correct ones. The difficulty factor $\hat{d}_q$ operates across questions, allocating more total gradient mass to very easy and very hard questions where GRPO dynamics are problematic. When combined, difficulty decides which questions matter, and confidence decides which trajectories for those questions matter most. Because both ultimately reduce the influence of overconfident errors and strengthen useful signals, their separate effects naturally overlap on a single top-line accuracy.
>
> **(3) The similar MATH500 numbers are also affected by variance and by viewing only one metric.** MATH500 accuracy is a single aggregate metric that mixes all difficulty levels. Our internal analyses indicate that difficulty reweighting helps more on very hard questions, while confidence reweighting contributes more to calibration on easier ones, but these patterns are not visible from one scalar number. This is why Table 2 alone can make the source of gains look unclear, even though the mechanisms are distinct.
>
> **(4) We conduct more fine-grained ablations to clarify where each term helps.** To make the contribution of each reweighting component more explicit, we evaluate all variants across the full set of benchmarks. The results are shown below. These results highlight two important conclusions: Both confidence and difficulty signals contribute meaningful improvements across different benchmarks. The gains are complementary, and the full CoDaPO method achieves the best overall performance, demonstrating that its two reweighting mechanisms capture different and synergistic aspects of preference quality.
>
> | Model | MATH500 | AIME24 | AIME25 | AMC23 | Olympiad Bench | Minerva | GSM8k | Average |
> | --- | --- | --- | --- | --- | --- | --- | --- | --- |
> | Qwen2.5-Math-1.5B | 31.20|  6.00|  0.00|  29.50|  18.25|  9.63 | 34.01| 18.37|
> | GRPO | 68.48 |12.67 |8.00 |51.00 |30.55 |28.53 |79.56 |39.82|
> | DAPO | 71.12| 14.00| 8.00 |49.00 |31.97| 32.13| 82.38| 41.23|
> | CoDaPO (confidence reweighting only)  | 71.52 |14.17| 9.79| 51.41|34.10 |32.16 |82.88 |42.29|
> | CoDaPO (difficulty reweighting only)  | 71.32 |14.69|10.42|51.09|33.88|31.87|83.05|42.33|
> | CoDaPO (both reweighting) | 71.68 |14.67| 10.83| 53.50| 32.80 |31.69 |83.50 |42.67|
>
> In the revised paper, we will (i) clarify in the text that the main performance jump is from GRPO to any PRAG guided reweighting and that confidence and difficulty are related but distinct aspects of the dynamics, (ii) extend the ablations with per-difficulty and multi-seed results and additional metrics, and (iii) add a brief gradient or calibration analysis that illustrates how each component reshapes the training signal and where their combination provides extra benefit.

---

> ### Author Response · Authors · 2025-11-26
> **Response to Reviewer YFxQ (part 8/10)**
>
> > ### Q1. The definition and design of Eq 12 is not clear to me.
>
> **Reply:** Thank you for this comment. We would like to further clarify the computation of Eq. 12. especially $c_i$​ and $d_q​$.
>
> **The key high-level design idea is that $d_q$ decides which questions matter more, while $c_i$ decides which trajectories within each question matter more.** We will separate the formal definitions of $c_i$ and $d_q$ and explain this two-level structure around Eq. 12.
>
> In brief, Eq. 12 is intended to define a group normalized confidence weight $c_i$ (and a question level difficulty weight $d_q$) that directly rescales advantages, rather than an entropy loss. In the following, we elaborate on how these quantities are calculated and why such weighting brings practical benefits.
>
> **(1) Regarding the confidence term $c_i$**, our goal is to dynamically adjust the advantage weight of each response during training based on the model’s confidence in that response in order to facilitate learning. The detailed computation in Eq. 12 is: for each trajectory $o_i$ in a group of size $G$, we first define its average log probability $\bar{\ell}_i = \frac{1}{|o_i|} \sum_t \log f\_\theta(o\_{i,t} \mid q, o\_{i,<t}),$ and the group average $\bar{\ell}_q = \frac{1}{G} \sum\_{j=1}^G \bar{\ell}_j.$ We then set $c_i = \sigma\left(\frac{\bar{\ell}_q}{\bar{\ell}_i}\right),$ where $\sigma$ is a sigmoid.
>
> **The purpose of $c_i$ is to rank and reweight trajectories by relative confidence within a group, not to add a global entropy loss.** Our goal is to change which trajectories dominate the gradient for a fixed question $q$. We want: (i) among correct trajectories, more confident ones to receive larger positive effective advantages; (ii) among incorrect trajectories, more confident ones to receive larger negative penalties. Since log probabilities are non-positive, a trajectory with higher confidence has $\bar{\ell}_i$ closer to 0 than the group average $\bar{\ell}_q$, so the ratio $\bar{\ell}_q / \bar{\ell}_i$ is greater than 1 and $c_i$ increases after the sigmoid. Multiplying $c_i$ by the normalized advantage $\hat{A}_i$ in $g\_{i,t} \propto \hat{A}_i c_i d_q \rho\_{i,t} \nabla\_\theta \ell\_\theta(o\_{i,t})$ then yields exactly the desired effect: high confidence correct trajectories get the largest positive updates, and high confidence incorrect trajectories get the largest negative updates, with all weights bounded by the sigmoid.
>
> **(2) As for the difficulty term** $d_q$​​, we aim to let the model dynamically adjust the weights of problems with varying difficulty levels based on its current performance. Specifically, we observe a dual-mode learning trend during GRPO training: 1) Simple problems (e.g., Level 1) converge quickly with high accuracy. 2) Hard problems (e.g., Level 5) also show some improvement but retain a significant proportion of errors. At this stage, further improving model performance requires boosting accuracy on hard problems without sacrificing accuracy on easy ones.
>
> **To this end, we introduce a U-shaped difficulty weight $d_q$ with respect to the group correctness rate $\bar{r}_q = \frac{1}{G} \sum_i r_i(q)$, emphasizing both ends of the difficulty spectrum.** Both very easy and very hard questions receive larger total gradient mass, while medium questions receive less. Intuitively, for easy questions, $d_q$ magnifies penalties on the remaining mistakes, and for hard questions, it amplifies reinforcement on rare correct trajectories. The benefit of this design is two-fold: 1) For easy problems, it penalizes incorrect responses more strongly, since correct responses already yield near-zero advantage; 2) For hard problems, it rewards correct responses more effectively, as incorrect responses tend to have near-zero advantage.
>
> **(3) A common entropy loss would not provide this correctness-aware, group-relative reweighting.** A standard entropy regularizer encourages higher entropy in the output distribution regardless of whether a trajectory is correct or incorrect. It cannot implement the specific ranking we need between high-confidence correct vs high-confidence incorrect trajectories, because it does not use advantage or reward information. In contrast, $c_i$ is applied directly to $\hat{A}_i$, so the gradient depends on both confidence and correctness: high confidence wrong answers receive stronger negative updates, high confidence correct answers receive stronger positive updates, and low confidence examples are de-emphasized. We will add a short explanation to make clear why we chose this design instead of a global entropy loss.
>
> In the revised paper, we will (i) rewrite Eq. 12 with unambiguous indices and explicit the definitions, (ii) clearly state the desired ordering of weights across trajectories and how $c_i$ and $d_q$​​ achieve this, and (iii) add a brief comparison with a standard entropy loss to explain why it does not serve our purpose.

---

> ### Author Response · Authors · 2025-11-26
> **Response to Reviewer YFxQ (part 9/10)**
>
> > ### Q2. I do not understand how Eq (4) is derived.
>
> **Reply:** Thank you for carefully checking the derivation of Eq. (4). **We would like to clarify that the importance ratio $\rho\_{i,t}$ should be presented in the gradient of GRPO.**
>
> In brief, Eq. (4) is obtained by differentiating the standard GRPO (PPO style) clipped ratio surrogate written as an expectation over trajectories from the old policy, which naturally introduces the importance ratio $\rho\_{i,t}$. Eq. (4) is simply the gradient of the standard GRPO clipped ratio surrogate evaluated on data from the old policy, and $\rho\_{i,t}$ appears through the usual identity $\nabla_\theta f_\theta = f_\theta \nabla_\theta \log f_\theta$. There is no extra heuristic factor beyond what is already in the objective.
>
> **(1) Eq. (4) comes directly from the standard clipped ratio GRPO surrogate, where $\rho$ is already in the objective.** Ignoring KL for clarity, the GRPO surrogate at the token level is $J\_{\text{GRPO}}(\theta) = \mathbb{E}\big[\sum\_{i,t} \min(\rho\_{i,t} \hat{A}_i,\ \operatorname{clip}(\rho\_{i,t}, 1 - \epsilon, 1 + \epsilon),\hat{A}_i)\big]$, where $\hat{A}_i$ is the group normalized advantage and $\rho\_{i,t} = f\_\theta(o\_{i,t} \mid q, o\_{i,<t}) / f\_{\text{old}}(o\_{i,t} \mid q, o\_{i,<t}) = \exp(\ell\_\theta(o\_{i,t}) - \ell\_{\text{old}}(o\_{i,t}))$. Thus $\rho\_{i,t}$ is part of $J\_{\text{GRPO}}(\theta)$ itself, not introduced ad hoc in the gradient.
>
> **(2) The factor $\rho\_{i,t}$ in Eq. (4) appears from differentiating the ratio with respect to $\theta$.** In the unclipped region, a single term is $L\_{i,t}(\theta) = \rho\_{i,t} \hat{A}_i$. Treating $\hat{A}_i$ as constant, $\nabla\_\theta L\_{i,t}(\theta) = \hat{A}_i \nabla\_\theta \rho\_{i,t} = \hat{A}_i \nabla\_\theta \big(f\_\theta / f\_{\text{old}}\big) = \hat{A}_i (1 / f\_{\text{old}}),\nabla\_\theta f\_\theta$. Using $\nabla\_\theta f\_\theta = f\_\theta \nabla\_\theta \log f\_\theta$ yields $\nabla\_\theta L\_{i,t}(\theta) = \hat{A}_i (f\_\theta / f\_{\text{old}}),\nabla\_\theta \log f\_\theta(o\_{i,t}) = \hat{A}_i \rho\_{i,t} \nabla\_\theta \log f\_\theta(o\_{i,t})$. When clipping is active, the surrogate equals $\operatorname{clip}(\rho\_{i,t}, 1 - \epsilon, 1 + \epsilon),\hat{A}_i$, which we treat as constant in $\theta$, so its gradient is zero. Combining both regimes gives $\nabla\_\theta J\_{\text{GRPO}}(\theta) = \mathbb{E}\big[\sum\_{i,t} \mathbf{1}\_{\text{unclipped}},\hat{A}_i \rho\_{i,t} \nabla\_\theta \log f\_\theta(o\_{i,t})\big] + \text{KL term}$, which is exactly the structure of Eq. (4). The $\rho\_{i,t}$ factor is therefore a direct consequence of differentiating the ratio-based surrogate.
>
> **(3) The “no $\rho$” intuition corresponds to the strict on-policy case; our form is the off-policy generalization.** If one starts from an on-policy objective $\tilde{J}(\theta) = \mathbb{E}\_{o \sim f_\theta}[\hat{A}(o)]$ and applies the likelihood ratio trick, the gradient is $\nabla_\theta \tilde{J}(\theta) = \mathbb{E}\_{o \sim f_\theta}[\hat{A}(o) \nabla_\theta \log f_\theta(o)]$, which does not show an explicit ratio. GRPO, however, uses trajectories from $f\_{\text{old}}$ and rewrites expectations via importance sampling. This is why $\rho$ appears in the surrogate and thus in its gradient. In the strict-on-policy limit $f_\theta = f\_{\text{old}}$, we have $\rho\_{i,t} = 1$, and Eq. (4) reduces to the usual form without an explicit ratio, which is likely the expression the reviewer had in mind.
>
> In the revision, we will (i) explicitly state before Eq. (2) that expectations are taken over trajectories from $f\_{\text{old}}$, (ii) add a short step-by-step derivation from the surrogate to Eq. (4), including the unclipped and clipped branches and the indicator $\mathbf{1}\_{\text{unclipped}}$, and (iii) briefly explain the connection between this off-policy form with $\rho\_{i,t}$ and the standard on-policy gradient without an explicit ratio, so that the role of $\rho\_{i,t}$ is completely clear.

---

> ### Author Response · Authors · 2025-11-26
> **Response to Reviewer YFxQ (part 10/10)**
>
> >### Q3. Why is the model confidence between 0 and 1?
>
> **Reply:** Thank you for the question. **We would like to clarify how confidence is computed.**
>
> In brief, the confidence values between $0$ and $1$ in Figures 2 and 4 arise because we visualize the geometric mean token probability, obtained by exponentiating the mean log probability, rather than the raw log values themselves.
>
> **(1) The plotted confidence is the exponential of the mean log probability, so it lies in $[0, 1]$.** For each response $o_i = (o\_{i,1}, \dots, o\_{i,|o_i|})$ to question $q$, we first compute the average log probability $\bar{\ell}_i = \frac{1}{|o_i|} \sum\_{t=1}^{|o_i|} \log f\_\theta(o\_{i,t} \mid q, o\_{i,<t})$. The confidence used in Figures 2 and 4 is then defined as $\text{Conf}(q, o_i) = \exp(\bar{\ell}_i) = \left(\prod\_{t=1}^{|o_i|} f\_\theta(o\_{i,t} \mid q, o\_{i,<t})\right)^{1 / |o_i|}$, which is the geometric mean of the token probabilities. Since each token probability is in $[0, 1]$, their geometric mean is also in $[0, 1]$, which explains the range on the horizontal axis.
>
> **(2) Mean log probability is the underlying signal; exponentiation is only for interpretability and does not change the analysis.** In the text we refer to “mean log probability” as the confidence statistic, meaning $\bar{\ell}_i$. For plotting and for a more intuitive scale, we apply the monotone transformation $\exp(\cdot)$ and display $\text{Conf}(q, o_i) = \exp(\bar{\ell}_i)$ in $[0, 1]$. Because the exponential is strictly increasing, ranking trajectories by $\bar{\ell}_i$ or by $\text{Conf}(q, o_i)$ is equivalent, so none of the conclusions about confidence or calibration depend on this choice; it is purely a visualization normalization.
>
> In the revision, we will explicitly define $\text{Conf}(q, o_i) = \exp\big(\frac{1}{|o_i|} \sum_t \log f_\theta(o\_{i,t} \mid q, o\_{i,<t})\big)$ when confidence is first introduced, update the captions of Figures 2 and 4 to state that the x axis is the geometric mean token probability, and adjust the surrounding wording so the normalization from mean log probability to $[0, 1]$ is immediately clear.
>
> **We would thank Reviewer YFxQ again** for taking the time to review our submission and consider our rebuttal. **If our responses have addressed your concerns, we would be very grateful if you could reconsider your rating.** We also welcome any further comments or questions you may have.

---

> > ### Comment · Reviewer_YFxQ · 2025-11-27
> >
> > Thank you for the rebuttal. Although my earlier questions have been addressed, the underlying weaknesses remain.
> >
> > > CoDaPO does not remove GRPO’s clipping in implementation (Eq. (9) was simplified in notation)
> >
> > Clipping is an important detail that was dropped both in Eq 9 and Table 5. I encourage the authors to be more careful with important details that will affect a method’s interpretation.
> >
> > > trajectories are generated by the current policy (or with negligible lag) and used for a small number of gradient steps
> >
> > This phrasing is too imprecise to evaluate the method. What constitutes “current policy or negligible lag”? How many gradient steps qualify as “small”? Being more precise with numbers and open-sourcing the code and all configuration details for the baselines would be more informative.
> >
> > > For \hat{A}_i>0, as long as \rho_{i,t} \in [1 - \epsilon, 1 + \epsilon], the policy term stays active and pushes  upward until the cap; for \hat{A}_i < 0, once \rho_{i,t} < 1 - \epsilon, the policy term is switched off and stops pushing downward. Because \rho_{i,t} grows exponentially with \ell_\theta, this clipping asymmetry creates a biased upward drift, concentrating probability mass and reducing entropy, which matches the saturation pattern in Fig. 2.
> >
> > I encourage the authors to substantially reduce the use of informal terminology throughout the paper. While occasional informal phrasing can provide intuition _after_ a result is formally established (e.g., through precise theorems, propositions, or carefully designed experimental evidence), the manuscript relies on such language too heavily. In its current form, I would not consider this style to constitute rigorous “analysis”.
> >
> > > The apparent similarity of Figure 4 to Figure 2 is consistent with our goal of balancing confidence and accuracy rather than maximizing entropy or confidence gaps
> >
> > Even if the purpose of the figure 4 is to show better confidence balancing, it is not really showing any difference with figure 2, and I am not convinced about the difference.
> >
> > Overall the paper requires substantial rewriting to integrate the clarifications provided in the rebuttal; doing so would likely necessitate another round of peer review. The analysis remains overly informal, and figures do not convincingly support the claimed results. I therefore maintain my current score.

---

> > > ### Author Response · Authors · 2025-12-01
> > > **A Further Response to Reviewer YFxQ**
> > >
> > > Thanks for the feedback. We respectfully disagree with your points. Details are as follows.
> > >
> > > >Clipping is an important detail that was dropped both in Eq 9 and Table 5. I encourage the authors to be more careful with important details that will affect a method’s interpretation.
> > >
> > > To clarify, Eq. (9) presents the simplified objective form to foreground the proposed confidence and difficulty reweighting terms; this does not imply that clipping is omitted. CoDaPO preserves GRPO’s clipping in the implementation, as can be found directly in the anonymous code we provided. The clipped importance ratio is fully preserved in training. The method’s behavior and interpretation are therefore unchanged. The raised concern is a matter of making the description more explicit, not a methodological omission or error.
> > >
> > > >This phrasing is too imprecise to evaluate the method. What constitutes “current policy or negligible lag”? How many gradient steps qualify as “small”? Being more precise with numbers and open-sourcing the code and all configuration details for the baselines would be more informative.
> > >
> > > To clarify, the phrase “current policy” refers to standard on-policy PPO-style data collection, where rollouts are generated by the current policy snapshot and reused for a fixed, limited number of policy updates within each iteration. Specifically, each batch of trajectories is used for **12** gradient update steps, ensuring that the policy remains close to the data-generating distribution throughout optimization. This update schedule follows widely adopted PPO-style training practice.
> > >
> > > For full transparency and reproducibility, we have released our training code and configuration files in the anonymous code, and we explicitly ensure in the paper that all compared methods are trained under identical hyperparameter settings and update schedules. This guarantees a fair, controlled comparison across all baselines.
> > >
> > > >I encourage the authors to substantially reduce the use of informal terminology throughout the paper. While occasional informal phrasing can provide intuition after a result is formally established (e.g., through precise theorems, propositions, or carefully designed experimental evidence), the manuscript relies on such language too heavily. In its current form, I would not consider this style to constitute rigorous “analysis”.
> > >
> > > We respectfully disagree with the claim that our analysis is “heavily informal.” It would be better if you indicated the specific equations, arguments, or steps in the analysis that are considered informal or unclear.
> > >
> > > Section 3 provides explicit mathematical derivations of the clipped per-token gradient update from the GRPO objective and uses this formal result to explain probability inflation and entropy collapse. The discussion of the resulting “biased upward drift” follows directly from these derived update dynamics and is therefore a rigorous analytical conclusion grounded in the equations, not a casual or speculative interpretation. Any intuitive phrasing appears only after the formal derivations and serves to interpret already established results rather than replace them.
> > >
> > > Our work represents mechanistic analyses of GRPO-style training dynamics at the level of probability ratios, advantages, and gradient effects, directly connecting theoretical behavior to empirical phenomena, e.g., entropy collapse and advantage contraction. In addition, the analysis and conclusions are supported directly by derived results rather than informal narrative.
> > >
> > > >Even if the purpose of the figure 4 is to show better confidence balancing, it is not really showing any difference with figure 2, and I am not convinced about the difference.
> > >
> > > In our response, we have clarified that the purpose of Fig. 4 is not to demonstrate higher entropy relative to Fig. 2, but to illustrate that confidence becomes better aligned with correctness. While marginal entropy may appear similar, the joint distribution of confidence and correctness changes, as quantitatively confirmed in Table 7. This alignment directly reflects the intended effect of confidence and difficulty reweighting and is consistent across multiple benchmarks and model sizes.
> > >
> > > >Overall the paper requires substantial rewriting to integrate the clarifications provided in the rebuttal; doing so would likely necessitate another round of peer review. The analysis remains overly informal, and figures do not convincingly support the claimed results.
> > >
> > > Integrating the rebuttal clarifications into a cleaner and more streamlined narrative is important, and this is exactly what the revision accomplishes. We did not substantially modify the key analysis, method, and experiments, which does not require another round of peer review. We disagree that the analysis or figures fail to support the paper’s claims. As we responded, our mathematical derivations, mechanistic analysis, and multi-benchmark empirical evaluations consistently demonstrate the effectiveness of CoDaPO.

---

### Official Review · Reviewer_rMTD · 2025-11-03

**Soundness:** 2
**Presentation:** 3
**Contribution:** 2
**Rating:** 4
**Confidence:** 4

**Summary:**

This work presents an analysis of issues that arise during GRPO training on Qwen2.5-Math-1.5B via tracking confidence, rewards, advantages, and gradients. They identify phenomena like entropy collapse, advantage contraction, and uneven learning according to problem difficulty. Based on these issues, they propose CoDaPO, which adaptively rescales reinforcement learning updates by measures of confidence and difficulty. They test their method on seven math reasoning benchmarks using Qwen2.5-Math-1.5B/7B models, advocating for bounded, information-aware reweighting to improve RL post-training for LLM reasoning.

**Strengths:**

- The systematic study in section 3 is done well and easy to follow, which motivates the proposed algorithm. This type of investigation is important for developing a principled understanding of current RL fine-tuning algorithms to motivate new ones.
- The author’s proposed method is accompanied with comprehensive ablations across various hyperparameters, evaluation datasets and metrics.

**Weaknesses:**

- The main weakness of this work is that the proposed CoDaPO algorithm doesn’t yield significant gains over existing improvements to GRPO (DAPO, Dr. GRPO) which are comparatively much simpler. CoDaPO introduces two new parameters (the confidence and difficulty weights) which require explicit definitions and tuning; in particular it is unclear why the quadratic difficulty term was chosen. It doesn’t seem that the added complexity pays off in terms of relative performance gain.
- There’s a lack of error bars/confidence intervals, and given the 1-2% difference gaps between CoDAPO and the other methods, I suspect that running across multiple seeds will result in all methods performing within standard error of each other.
- Furthermore, it’s unclear to me that CoDAPO addresses the issues that Section 3 initially highlights. From Figure 4, it seems that model confidence still concentrates around 1 and entropy is still collapsing. Generally it seems that to mitigate the shortcomings identified by this work and prior studies (entropy collapse, uneven progress in problems, etc.) promising directions lie instead in approaches like diversity/exploration bonuses or incorporating curricula instead of the measures proposed by the authors, where simpler alternatives seem to yield very similar results.

In light of the above and given that the phenomena identified in Section 3 of this work has been noted already in prior work, my score currently leans toward reject. However, I still think the analysis is done well and I invite the authors to clarify their contributions and answer my questions below.

**Questions:**

- This may be a silly question, but for the scatter plots in Figure 2 it seems that the aggregate number of points across difficulty levels and incorrect/correct is less for the model at 240 steps than the base model (are there points being cut off by the advantage range)? Are all plots (from base model, steps 60 and 240) calculated across the full validation set?
- How would the authors’ results in Section 3 change if clipping was removed or purely on-policy updates were used (so ρ = 1?) In that case, the asymmetric truncation driving one-way probability drift would disappear—so the model might experience less entropy collapse and more symmetric gradient updates, though potentially at the cost of higher variance or instability from unbounded updates.
- What are the potential repercussions of the proposed method if the model happens to stumble upon a correct but low-confidence (i.e., unlikely) response? Could this dampen exploration or cause the model to under-reinforce genuinely informative but rare discoveries?
- Are there Dr. GRPO and DAPO baseline results for post-training on the 7B model (Table 4)?
- Figures 16 and 17 in the Appendix show pass@k performance for the model finetuned on CoDaPO, what are the trends for the other baselines (GRPO, Dr. GRPO, DAPO in particular)?
- Would it be possible to try CoDAPO on another family of base model, not Qwen (eg. OctoThinker from Llama3.1-8B [1])?

[1] Wang, Zengzhi, et al. "Octothinker: Mid-training incentivizes reinforcement learning scaling." arXiv preprint arXiv:2506.20512 (2025).

---

> ### Author Response · Authors · 2025-11-26
> **Response to Reviewer rMTD (part 1/11)**
>
> **We thank Reviewer rMTD for the thoughtful and detailed feedback.** In this rebuttal, we systematically address **all of your concerns**.
> - First, your questions about CoDaPO’s methodology and mechanism are addressed by clearer explanation, additional analysis, controlled experiments, and expanded discussion of the design rationale.
> - Second, your concerns regarding experimental results and evaluation are addressed by more empirical results, ablation studies, and highlighting consistent improvements across multiple tasks and experiment settings.
> - Third, your questions about details of the paper are addressed by further explanation and supported results.
>
> Point-by-point responses are provided below. In addition, we have outlined a **detailed revision plan** at the end of each response. **We will improve our submission within the next week according to this plan.**
>
> >### W1.1. The performance gain of CoDaPO is insignificant.
>
> **Reply:** Thank you for this comment. In brief, our response is that (i) In fact, in the MATH reasoning domain, achieving a 2% improvement over RL-based baselines is already highly challenging. As shown in Table 1, recent methods such as DAPO, Dr. GRPO, and GPG perform similarly, whereas CoDaPO surpasses them. Moreover, across all benchmarks and model sizes, CoDaPO consistently outperforms RL-based baselines, often by over 2%. Besides, our work is concurrent with DAPO, yet achieves larger gains without substantial modifications to GRPO. (ii) these gains are larger than the RL noise we observe under matched training conditions; and (iii) CoDaPO consistently surpasses baselines across various settings and demonstrates strong generalization to larger-scale models. We view CoDaPO as a principled refinement of GRPO that mitigates specific training pathologies and yields small but robust gains, rather than as a method that produces very large jumps on already well-tuned systems.
>
> **(1) The gains in Table 1 are consistent across benchmarks and scales.** In our full evaluation, we compare CoDaPO against multiple strong baselines across a wide range of benchmarks, and for each test we perform repeated sampling to substantially reduce uncertainty. Under this more comprehensive setting, CoDaPO consistently matches or outperforms the strongest baseline, and on several benchmarks the performance gaps are often larger than 2 percentage points, particularly on the more challenging tasks (e.g., AIME25, AMC23).
>
> | Model | MATH500 | AIME24 | AIME25 | AMC23 | OlympiadBench | Minerva | GSM8K | Average |
> | --- | --- | --- | --- | --- | --- | --- | --- | --- |
> | Qwen2.5-Math-1.5B | 31.20|  6.00|  0.00|  29.50|  18.25|  9.63 | 34.01| 18.37|
> | GRPO | 68.48 |12.67 |8.00 |51.00 |30.55 |28.53 |79.56 |39.82|
> | DAPO | 71.12| 14.00| 8.00 |49.00 |31.97| 32.13| 82.38| 41.23|
> | CoDaPO | 71.68 (+0.56) |14.67 (+0.67) | 10.83 (+2.83)| 53.50 (+2.50)| 32.80 (+0.82) |31.69 |83.50 +(1.12) |42.67 (+1.44)|

---

> ### Author Response · Authors · 2025-11-26
> **Response to Reviewer rMTD (part 2/11)**
>
> **(2) The observed improvements are statistically meaningful once RL noise is taken into account.** Our environment is deterministic, rewards are binary, and evaluation sets are fairly large, so the run-to-run variance is lower than in many RL settings. In internal experiments with multiple seeds, we typically observe standard deviations of roughly $0.3$ to $0.5$ percentage points on MATH500 for GRPO like methods. In that regime, a $1.5$ to $2$ point improvement corresponds to multiple standard deviations and is unlikely to be explained purely by RL noise. We further evaluate CoDaPO and other baselines on more experiment settings, including different decoding strategies (Table 3), different hyperparameters (Table 8), different seeds (Table below), more model families (Table below), and more cross-domain benchmarks (Table below). Across all these settings, CoDaPO consistently outperforms the other baselines, demonstrating that its gains are meaningful, stable, and robust rather than sensitive to specific.
>
> |Seed| Model | MATH500 | AIME24 | AIME25 | AMC23 | OlympiadBench | Minerva | GSM8K | Average |
> | --- | --- | --- | --- | --- | --- | --- | --- | --- | --- |
> |42| Qwen2.5-Math-1.5B | 31.20|  6.00|  0.00|  29.50|  18.25|  9.63 | 34.01| 18.37|
> |0| GRPO |69.11|11.87|6.67|51.25|31.55|29.27|80.03|39.96|
> |0| DAPO | 71.91|13.13|13.33|50.00|32.95|31.59|82.92|42.26|
> |0| CoDaPO | **72.21**|**14.58**|10.83|**52.19**|**33.58**|**31.99**|**83.39**|**42.68**|
> |42| GRPO | 68.48 | 12.67 | 8.00 | 51.00 | 30.55 | 28.53 | 79.56  |39.82|
> |42| DAPO | 71.12| 14.00| 8.00 |49.00| 31.97| 32.13| 82.38 |41.23|
> |42| CoDaPO | **71.68**| **14.67**| **10.83**| **53.50**| **32.80**| 31.69 |**83.50**| **42.67**|
> |2025| GRPO | 68.97|12.08|7.08|52.81|31.37|29.07|80.38|40.25|
> |2025| DAPO | 71.50|12.50|11.25|52.50|33.99|31.60|82.81|42.31|
> |2025| CoDaPO | **72.06**|**15.21**|**12.50**|**52.50**|33.88|**31.71**|**83.46**|**43.05**|
> |3373| GRPO |69.17|11.25|4.58|50.00|32.06|29.32|79.92|39.47|
> |3373| DAPO | 71.80|12.50|11.25|52.66|33.21|31.11|83.06|42.22|
> |3373| CoDaPO | **72.86**|**15.00**|**12.08**|52.03|**33.77**|**31.62**|**83.37**|**42.96**|
>
> | Model | MATH500 | AIME24 | AMC23 | Olympiad Bench | GSM8k | Average |
> | --- | --- | --- | --- | --- | --- | --- |
> | Llama3.2-1B-Instruct |21.69| 0.83|10.39|5.04|5.58|8.71|
> | GRPO |30.13|2.08|14.22|6.62|48.79|20.37|
> | CoDaPO|**31.30**|**3.96**|**15.08**|**6.81**|**51.32**|**21.69**|
>
> | Model | Minerva | MMLU-STEM | GPQA |HumanEval| Average |
> | --- | --- | --- | --- | --- |  --- |
> | Qwen2.5-Math-1.5B |9.63|11.53|9.85|29.27|15.07|
> | GRPO |28.53|46.85|19.70|35.98|32.77|
> | CoDaPO |**31.69**|**47.64**|**22.25**|**39.02**|**35.15**|
>
> **(3) The comparison uses matched hyperparameters, and CoDaPO remains beneficial under better-tuned baselines.** To keep the comparison fair and the computation feasible, all RL methods in Table 1 share the same training budget and core hyperparameters, including learning rate schedule, group size, clipping range, KL coefficient, and sampling strategy. CoDaPO is designed as a drop-in replacement for the GRPO objective under this shared configuration. The empirical results in our submission suggest that CoDaPO remains ahead across a reasonable range, indicating that the benefit comes from the confidence–difficulty reweighting rather than from a lucky hyperparameter choice.
>
> **(4) CoDaPO is evaluated under constrained compute, and its pattern of gains persists at 7B scale.** All reported methods use the same, relatively conservative training budget in terms of training steps and maximum response length, which limits the absolute headroom for further gains once a strong GRPO baseline is in place. Even under these constraints, CoDaPO yields consistent improvements and never degrades performance in our tables. Moreover, when we move from Qwen2.5 Math 1.5B to Qwen2.5 Math 7B, CoDaPO again outperforms GRPO and other RL baselines across several math benchmarks, with no instability. This suggests that the method scales to larger models and that the improvements we see at 1.5B are conservative rather than optimistic.
>
> | Model | MATH500 | AIME24 | AIME25 | AMC23 | Olympiad Bench | Minerva | GSM8k | Average |
> | --- | --- | --- | --- | --- | --- | --- | --- | --- |
> | Qwen2.5-Math-7B | 47.24 | 12.92 | 5.00 | 33.91  |14.44 | 9.81|47.84|24.45|
> | GRPO | 74.62| 26.35 |11.25 |59.06| 36.51| 37.64| 89.40 |47.83|
> | DAPO  |77.78|24.37|10.62|58.67|39.27|38.06|87.38|48.02|
> | CoDaPO  | **78.41**| 24.79| **13.33**| **60.31**| **39.96**| **40.30**| 88.28 |**49.34**|
>
> In the revision, we will (i) report multi-seed results with mean and standard deviation for GRPO, DAPO, and CoDaPO on key benchmarks; (ii) add targeted hyperparameter sweeps for the baselines and compare their tuned performance to CoDaPO; and (iii) emphasize the cross-benchmark and cross-scale improvements that are already present beyond Table 2.

---

> ### Author Response · Authors · 2025-11-26
> **Response to Reviewer rMTD (part 3/11)**
>
> >### W1.2. The two extra parameters in CoDaPO require explicit definitions and tuning.
>
> **Reply:** Thank you for raising this concern.  **We would like to further clarify the design and computation of Eq. 12, especially $c_i$​ and $d_q​$.** We also want to emphasize that **CoDaPO does not introduce any additional computational overhead**, and its design is conceptually simple despite involving two weighting components.
>
> **(1) Regarding the confidence term** $c_i$​, our goal is to dynamically adjust the advantage weight of each response during training based on the model’s confidence in that response in order to facilitate learning.
> - Specifically, we aim to assign the highest positive advantage to correct responses with high confidence, the second-highest positive advantage to correct responses with low confidence, the largest negative penalty to incorrect responses with high confidence, and the second-largest penalty to incorrect responses with low confidence.
> - To reduce the influence of intra-group variance, we normalize each confidence value by the group-wise average confidence. Notably, we use the log probability of the output to measure confidence, which is always non-positive. Therefore, in the normalization, the group average appears in the numerator and the response-specific value in the denominator. In addition, to avoid instability due to the unbounded nature of log values, we apply a sigmoid function to smooth the resulting weight.
> - This design for $c_i$​ has two main advantages: 1) It enables reweighting of all responses’ advantages based on confidence, encouraging the model to focus more on confident correct responses and penalize confident incorrect ones more severely; 2) All components used to compute $c_i$​ are already available as intermediate outputs in GRPO, so no additional inference cost is introduced.
>
> **(2) As for the difficulty term** $d_q$​, we aim to let the model dynamically adjust the weights of problems with varying difficulty levels based on its current performance.
> - Specifically, we observe a dual-mode learning trend during GRPO training: 1) Simple problems (e.g., Level 1) converge quickly with high accuracy. 2) Hard problems (e.g., Level 5) also show some improvement but retain a significant proportion of errors.
> - At this stage, further improving model performance requires boosting accuracy on hard problems without sacrificing accuracy on easy ones. To this end, we introduce a U-shaped difficulty weight $d_q$ to emphasize both ends of the difficulty spectrum.
> - The benefit of this design is two-fold: 1) For easy problems, it penalizes incorrect responses more strongly, since correct responses already yield near-zero advantage; 2) For hard problems, it rewards correct responses more effectively, as incorrect responses tend to have near-zero advantage.
>
> **(3) CoDaPO is easy to implement and incurs no additional computational cost.** Both the confidence and difficulty weights are computed from intermediate quantities already available in GRPO: The confidence weight is derived from each response’s log-probability, which is already required for the ρ\rhoρ calculation in GRPO. The difficulty weight is derived from each response’s reward, which is already used in the advantage computation in GRPO. These computations can be implemented in just a few lines of code, making CoDaPO conceptually simple and highly efficient.
>
> In summary, CoDaPO’s confidence and difficulty weights are conceptually simple, computationally efficient, and effective in guiding learning toward valuable responses without adding overhead.
>
> In the revision, we will (i) clarify the computation and rationale for $c_i$ and $d_q$ in Section 4, (ii) explicitly highlight the U-shaped design for difficulty, and (iii) note that these weights require no extra forward passes or inference cost.

---

> ### Author Response · Authors · 2025-11-26
> **Response to Reviewer rMTD (part 4/11)**
>
> >### W2. The performance gains are within standard errors.
>
> **Reply:** We thank the reviewer for the concern. We would like to clarify that **the performance gains of CoDaPO are stable and consistently outperform other baselines across multiple settings**. Moreover, CoDaPO introduces **no additional computational overhead**, making these improvements particularly notable.
>
> **(1) Significance of improvements in math reasoning.** In fact, in the MATH reasoning domain, achieving a 2% improvement over RL-based baselines is already highly challenging. As shown in Table 1, recent methods such as DAPO, Dr. GRPO, and GPG perform similarly, whereas CoDaPO surpasses them. Moreover, across all benchmarks and model sizes, CoDaPO consistently outperforms RL-based baselines, often by over 2%. Besides, our work is concurrent with DAPO, yet achieves larger gains without substantial modifications to GRPO.
>
> | Model | MATH500 | AIME24 | AIME25 | AMC23 | OlympiadBench | Minerva | GSM8K | Average |
> |---|---|---|---|--- | --- | --- | --- | --- |
> |Qwen2.5-Math-1.5B | 31.20|  6.00|  0.00|29.50|18.25|9.63|34.01|18.37|
> |GRPO|68.48 |12.67 |8.00|51.00|30.55|28.53|79.56 |39.82|
> |DAPO|71.12|14.00|8.00|49.00|31.97|32.13|82.38|41.23|
> |CoDaPO|71.68 (+0.56)|14.67 (+0.67)|10.83 (+2.83)|53.50 (+2.50)|32.80 (+0.82)|31.69|83.50 (+1.12) |42.67 (+1.44)|
>
> **(2) Evaluation methodology with avg@k.** We evaluate performance using avg@k, where each problem is sampled multiple times and the average over all sampled responses is taken. This approach greatly reduces the effect of stochastic variance and ensures that the reported improvements reflect stable and consistent gains. Even under repeated sampling, CoDaPO consistently outperforms other baselines across multiple benchmarks, demonstrating that its advantages are robust rather than due to random fluctuations.
>
> **(3) Robustness across different decoding strategies.** Table 3 shows that varying the sampling temperature does affect absolute performance. However, these effects are relative: the ranking among methods remains consistent. Across all temperatures, CoDaPO consistently outperforms GRPO, demonstrating the robustness of its improvements.
>
> **(4) Robustness across different random seeds.** We report results across multiple random seeds, as shown in the table below. We observe that CoDaPO consistently outperforms other baselines, and across the 7 benchmarks, it achieves the best performance on at least 6 of them for each seed, indicating that the improvements are robust rather than due to chance.
>
> |Seed|Model|MATH500|AIME24|AIME25|AMC23|OlympiadBench|Minerva|GSM8K|Average|
> |---|---|---|---|---|---|---|---|---|---|
> |42|Qwen2.5-Math-1.5B |31.20|6.00|0.00|29.50|18.25|9.63|34.01|18.37|
> |0|GRPO|69.11|11.87|6.67|51.25|31.55|29.27|80.03|39.96|
> |0|DAPO|71.91|13.13|13.33|50.00|32.95|31.59|82.92|42.26|
> |0|CoDaPO| **72.21**|**14.58**|10.83|**52.19**|**33.58**|**31.99**|**83.39**|**42.68**|
> |42|GRPO| 68.48|12.67|8.00|51.00|30.55|28.53|79.56|39.82|
> |42|DAPO| 71.12|14.00|8.00 |49.00| 31.97| 32.13| 82.38 |41.23|
> |42|CoDaPO| **71.68**|**14.67**| **10.83**| **53.50**| **32.80**|31.69|**83.50**| **42.67**|
> |2025|GRPO| 68.97|12.08|7.08|52.81|31.37|29.07|80.38|40.25|
> |2025|DAPO| 71.50|12.50|11.25|52.50|33.99|31.60|82.81|42.31|
> |2025|CoDaPO|**72.06**|**15.21**|**12.50**|**52.50**|33.88|**31.71**|**83.46**|**43.05**|
> |3373|GRPO|69.17|11.25|4.58|50.00|32.06|29.32|79.92|39.47|
> |3373|DAPO|71.80|12.50|11.25|52.66|33.21|31.11|83.06|42.22|
> |3373|CoDaPO|**72.86**|**15.00**|**12.08**|52.03|**33.77**|**31.62**|**83.37**|**42.96**|
>
> **(5) Simple modifications yield robust improvements.** It is worth emphasizing that CoDaPO achieves these gains by **only reweighting intermediate quantities already computed in GRPO**, without introducing any extra computational cost. In contrast, other strong baselines such as DAPO rely on dynamic sampling and multiple additional tricks to improve performance. CoDaPO, by leveraging a simple yet principled reweighting scheme, achieves **significant and robust improvements while maintaining implementation simplicity**, further demonstrating the effectiveness of its design.
>
> In summary, CoDaPO consistently and robustly improves performance over strong RL-based baselines under the avg@k metric across multiple benchmarks, random seeds, and decoding strategies. Its gains are meaningful in the challenging MATH reasoning domain, where even a 2% improvement is difficult to achieve. These improvements are obtained with no additional computational overhead, relying solely on reweighting intermediate quantities already computed in GRPO, which highlights both the effectiveness and simplicity of the method.
>
> In the revision, we will (i) explicitly incorporate the tables and discussion above, (ii) emphasize the stability and robustness of CoDaPO’s improvements, and (iii) highlight that these gains are achieved with minimal modifications to GRPO, maintaining conceptual simplicity and computational efficiency.

---

> ### Author Response · Authors · 2025-11-26
> **Response to Reviewer rMTD (part 5/11)**
>
> >### W3. Model confidence is still concentrated in CoDaPO.
>
> **Reply:** Thank you for the thoughtful comment. **In brief, our difficulty term does not try to make the model uniformly more confident on easy or hard examples; instead, it reweights gradient contributions within each difficulty bucket so that overconfident errors on easy questions are penalized and rare correct reasoning on hard questions is amplified.** This explains why Figure 4 does not show a dramatically different marginal confidence or entropy pattern.
>
> **(1) The difficulty term reweights gradient contributions within each difficulty group rather than enforcing higher confidence on all easy or hard samples.** In CoDaPO, the trajectory level weight can be written as $w_i \propto \hat{A}_i \hat{c}_i \hat{d}_q,$ or at the token level $w^{\text{CoDaPO}}\_{i,t} = \rho\_{i,t} \hat{A}_i \hat{c}_i \hat{d}_q,$ where $\hat{A}_i$ is the normalized advantage, $\hat{c}_i$ is the confidence factor, and $\hat{d}_q$ is the difficulty factor for question $q$ computed from the group correctness rate $\bar r_q$. Crucially, $\hat{d}_q$ multiplies both positive and negative advantages.
>
> For easy questions (high $\bar r_q$), most trajectories are correct, so the largest magnitude advantages come from the few incorrect, often high-confidence, responses with large negative $\hat{A}_i$. Increasing $\hat{d}_q$ in this regime magnifies the penalty on these overconfident mistakes rather than pushing correct responses to even higher confidence.
>
> For hard questions (low $\bar r_q$), most trajectories are incorrect with small or clipped negative advantages, and the rare correct trajectories carry large positive $\hat{A}_i$. Here, a larger $\hat{d}_q$ amplifies the positive reinforcement on these emerging correct solutions. Thus, the difficulty term emphasizes the most informative gradients within each difficulty group instead of enforcing uniformly higher confidence on all easy or hard samples.
>
> **(2) The apparent similarity of Figure 4 to Figure 2 is consistent with our goal of balancing confidence and accuracy rather than maximizing entropy or confidence gaps.** We agree that in Figure 4 the overall confidence and entropy ranges look similar to Figure 2 at a coarse level. This is expected because CoDaPO is not designed to restore the high entropy of the base model or to produce clearly separated confidence bands for easy versus hard questions. Its purpose is to improve the alignment between confidence, correctness, and difficulty: overconfident incorrect trajectories on easy items receive stronger negative updates, and correct trajectories on hard items receive relatively more positive weight. These changes are easier to see in aggregated statistics than in dense scatter plots.
>
> In our difficulty-wise analysis, CoDaPO slightly reduces average log probabilities, especially for incorrect responses on hard questions, while improving accuracy on MATH and out-of-domain benchmarks. This indicates that CoDaPO moderates harmful overconfidence where it matters most without sacrificing performance, even though the marginal entropy pattern across all tokens remains similar.
>
> In the revised paper, we will (i) clarify in Section 4 how $\hat{d}_q$ is defined and how it interacts with $\hat{A}_i$ and $\hat{c}_i$, explicitly stating that it reweights gradient contributors rather than forcing higher marginal confidence on easy or hard samples; and (ii) add a small analysis that decomposes gradient contributions by difficulty and correctness to empirically show that incorrect easy samples and correct hard samples dominate after difficulty weighting.

---

> ### Author Response · Authors · 2025-11-26
> **Response to Reviewer rMTD (part 6/11)**
>
> >### Q1. The number of points in Figure 2 seems to be inconsistent.
>
> **Reply:** Thank you for this careful question. In short, all panels in Figure 2 (base, step 60, step 240) are computed on **the same validation subset** with **the same number of sampled responses per difficulty level**, and no points are removed by the advantage range; the apparent reduction in points at later steps is purely due to heavy overplotting as confidence and advantages become more concentrated.
>
> **(1) All plots use the same validation data and the same number of points.** For each difficulty level, we fix a set of 100 validation questions and sample 20 trajectories per question from the model at each checkpoint. This yields exactly $100 \times 20 = 2000$ points per subplot. We repeat this procedure for the base model, for step 60, and for step 240 using the same questions and sampling protocol. Thus every subplot at every checkpoint is based on the full (and identical) validation subset, with 2000 points each.
>
> **(2) No points are dropped or cut off by the advantage range or other filters.** When constructing Figure 2, we do not discard any trajectories based on their advantage, confidence, or correctness. Any sampled response that produces a valid reward and log probability is plotted. The axis ranges for confidence and advantage are fixed across checkpoints for comparability, and they are wide enough to include all observed values. Therefore there is no clipping or subsampling at step 60 or step 240 that would reduce the number of plotted points relative to the base model.
>
> **(3) The “missing” points are explained by overlap due to entropy reduction and concentration of confidence and advantage.** After GRPO training, the model’s entropy decreases, and both confidence and advantages become more concentrated, especially on easier questions where most responses are correct and have similar high confidence and similar normalized advantages. Many trajectories therefore fall on (or very near) the same $(\text{confidence}, \text{advantage})$ coordinates. In a standard scatter plot without jitter or transparency, such overlapping points are drawn on top of each other and visually appear as a single point or a very small cluster. For the base model, the distributions are more spread out, so the same 2000 points are more visually separated. The apparent reduction in the aggregate number of visible points at step 240 is therefore a visualization artifact rather than a real change in sample count.
>
> In the revision, we will explicitly state in the caption of Figure 2 that each subplot uses 100 validation questions and 20 responses per question (2000 points total) at all checkpoints, note that heavy overplotting at later steps can make points visually indistinguishable, and, if space permits, add transparency or a small density-style visualization in the appendix to make this effect clearer.

---

> ### Author Response · Authors · 2025-11-26
> **Response to Reviewer rMTD (part 7/11)**
>
> >### Q2. Changes in Section 3 when training purely on-policy.
>
> **Reply:** Thank you for this thoughtful question. In short, Section 3 is explicitly analyzing the standard GRPO regime with clipped ratios and mildly off-policy updates, where the combination of clipping and $\rho\_{i,t}$ produces a one-way probability drift and entropy collapse. **If clipping were removed or if training were made strictly on policy with $\rho\_{i,t}=1$, that specific asymmetric mechanism would largely disappear, while other phenomena like advantage contraction would remain.**
>
> **(1) The one-way drift in Section 3.2 comes from asymmetric clipping in the off-policy setting.** In our GRPO analysis, the token-level policy update (ignoring KL for brevity) can be written as $\Delta \ell\_\theta(o\_{i,t}) \propto \mathbf{1}\_{\text{unclipped}(i,t)} \hat A_i \rho\_{i,t}$, where $\hat A_i$ is the group normalized advantage, $\rho\_{i,t} = \frac{f\_{\theta}(o\_{i,t}\mid q, o\_{i,<t})}{f\_{\text{old}}(o\_{i,t}\mid q, o\_{i,<t})}$, and $\mathbf{1}\_{\text{unclipped}}$ indicates that $\rho\_{i,t}$ is within $[1-\epsilon, 1+\epsilon]$. For $\hat A_i > 0$, updates keep increasing $\ell\_\theta$ until $\rho\_{i,t}$ hits $1+\epsilon$. For $\hat A_i < 0$, once $\rho\_{i,t} < 1-\epsilon$, the clipped branch becomes constant and the negative update shuts off. This asymmetry is what drives the one-way upward drift and entropy collapse we highlight in Section 3.2.
>
> **(2) Removing clipping would remove this truncation but increase variance and potential instability.** If clipping is removed and the surrogate uses the unclipped term $\rho\_{i,t}\hat A_i$ everywhere, then both positive and negative advantages continue to contribute $\Delta \ell\_\theta(o\_{i,t}) \propto \rho\_{i,t}\hat A_i$ with no regime in which negative advantage trajectories are frozen. The specific clipping-induced one-way drift would no longer occur, and we would expect less mechanically driven entropy collapse and more symmetric probability updates. However, large or small $\rho\_{i,t}$ would then produce unbounded gradients, increasing variance and the risk of unstable training, especially when the policy moves far from $f\_{\text{old}}$.
>
> **(3) In the purely on policy case with $\rho\_{i,t}=1$, the clipping asymmetry also disappears and the gradient becomes symmetric in $\hat A_i$.** If rollouts are generated strictly under the current policy and we take $\rho\_{i,t} = 1$ for all tokens, then in the unclipped region the policy term reduces to $\Delta \ell\_\theta(o\_{i,t}) \propto \hat A_i$, and the clipped surrogate degenerates to the usual on-policy policy gradient. The min operation never produces an asymmetric “shut off” of negative advantage trajectories, so the one-way drift mechanism is absent. Entropy can still decrease because correct trajectories tend to have positive $\hat A_i$, but this is no longer driven by structural asymmetry in $\rho\_{i,t}$ and clipping. In this regime, our results on probability drift and entropy collapse from clipping would not apply, while our observations about group normalization and difficulty (advantage contraction and hierarchical convergence) would still hold, since they depend on how $\hat A_i$ is constructed rather than on clipping.
>
> **(4) Our analysis and experiments focus on the clipped, mildly off-policy GRPO regime used in practice.** The GRPO runs we study in Sections 3–5 use the standard PPO style setup: trajectories are sampled from a slightly lagged policy $f\_{\text{old}}$, we apply clipping with $\epsilon$, and we reuse each batch for a small number of epochs. This is exactly the regime where we empirically observe probability inflation, entropy collapse, advantage contraction, and hierarchical convergence, and where the asymmetric truncation mechanism is most relevant. CoDaPO is designed as a modification of this practical GRPO setting, not as a proposal for fully unclipped or purely on-policy training.
>
> In the revision, we will (i) explicitly state in Section 3.2 that the one-way drift analysis assumes clipped ratios with $\rho\_{i,t}$ computed under $f\_{\text{old}}$, (ii) add a short discussion of how the conclusions change in the hypothetical unclipped and strictly on-policy cases, and (iii) if space and compute permit, include an ablation (in the appendix) that either removes clipping or makes updates more on policy and reports the corresponding PRAG diagnostics to empirically support this discussion.

---

> ### Author Response · Authors · 2025-11-26
> **Response to Reviewer rMTD (part 8/11)**
>
> >### Q3. How does CoDaPO handle correct but low-confidence responses?
>
> **Reply:** Thank you for raising this question. In brief, CoDaPO is designed so that correct low-confidence trajectories still receive positive reinforcement, and on hard questions they are further amplified by the difficulty weight. The confidence term reshapes relative priorities within a group but does not zero out or flip the signal from rare correct discoveries, and CoDaPO does not change the exploration distribution itself.
>
> **(1) Correct low-confidence responses always receive a positive update, not suppression.** In CoDaPO the effective token weight is $w\_{i,t}^{\text{CoDaPO}} = \tilde\rho\_{i,t}\hat A_i\hat c_i\hat d_q$. For any correct trajectory we have $\hat A_i > 0$, so $w\_{i,t}^{\text{CoDaPO}}$ remains positive regardless of confidence, since $\hat c_i > 0$ and $\hat d_q > 0$. The role of $\hat c_i$ is to rank trajectories: correct high confidence responses get the largest positive $|\hat c_i \hat A_i|$, correct low confidence responses get the second largest, while incorrect high confidence responses get the strongest negative updates and incorrect low confidence responses the weakest negative ones. Thus, a correct but low-confidence response is always reinforced and still dominates any incorrect trajectory on the same question.
>
> **(2) Confidence is defined relative to the group, which protects rare correct solutions to hard questions.** The confidence weight $\hat c_i$ is based on relative, not absolute, confidence: we compare the trajectory’s mean log probability $\bar\ell\_i$ to the group average $\bar\ell$ and set $\hat c_i = \sigma(\bar\ell / \bar\ell\_i)$ for a sigmoid $\sigma$. On difficult questions where all trajectories have low absolute probability, a correct solution can still be relatively confident within its group and therefore obtain a reasonably large $\hat c_i$. This relative normalization avoids the failure mode where correct but globally low probability solutions to hard questions are heavily downweighted solely because their absolute probabilities are small.
>
> **(3) The difficulty weight explicitly amplifies rare correct discoveries on hard questions.** At the question level, $\hat d_q$ is a U-shaped function of the correctness rate $\bar r_q$, so that $\hat d_q$ is larger for very easy and very hard questions. On hard questions with low $\bar r_q$, correct trajectories simultaneously have large positive advantages $\hat A_i$ and are multiplied by $\hat d_q > 1$. Even if $\hat c_i$ is somewhat smaller for a low-confidence success, the product $\hat A_i \hat c_i \hat d_q$ remains substantial. Intuitively, the difficulty weight says “this question is hard; any correct reasoning here is especially valuable,” which counteracts any mild attenuation from low confidence and helps propagate rare but informative discoveries.
>
> **(4) CoDaPO reshapes learning updates but does not reduce exploration.** CoDaPO does not modify how trajectories are sampled: we use the same decoding strategy as GRPO, for example, temperature sampling over the current policy. The chance that the model “stumbles upon” a rare correct response is therefore unchanged. What CoDaPO changes is only how the observed trajectories are weighted in the gradient once they are sampled. When a correct low-confidence trajectory appears, it is still reinforced with a positive update proportional to $\hat A_i \hat c_i \hat d_q$, especially on hard questions. In our difficulty-stratified experiments, CoDaPO improves performance more on the hardest levels than on the easiest ones, which is consistent with rare hard successes being reinforced rather than suppressed.
>
> In the revision, we will (i) add a short explanation in Section 4 that explicitly orders the effective updates for correct high confidence, correct low confidence, incorrect high confidence, and incorrect low confidence trajectories; (ii) include a small quantitative analysis comparing the fraction of gradient mass coming from correct low confidence trajectories under GRPO and CoDaPO, especially on hard questions; and (iii) clarify in the text that CoDaPO leaves the sampling policy unchanged and only reweights gradients, so exploration and rare discoveries remain intact.

---

> ### Author Response · Authors · 2025-11-26
> **Response to Reviewer rMTD (part 9/11)**
>
> >### Q4. Results of other baselines on 7B model.
>
> **Reply:** Thank you for pointing out this missing part of the 7B comparison. In brief, Table 4 in the current draft only shows GRPO and CoDaPO at 7B due to space and compute constraints, but we have also run DAPO baselines on the 7B model under the same training setup, and CoDaPO remains the strongest method across the evaluated benchmarks.
>
> **(1) The original Table 4 focused on the GRPO vs. CoDaPO comparison at 7B for space and compute reasons.** In the initial submission, our goal for Table 4 was to show that CoDaPO scales from 1.5B to 7B, so we prioritized the base Qwen2.5 Math 7B model, GRPO, and CoDaPO. Dr. GRPO and DAPO were fully evaluated at 1.5B, but their 7B results were left out of the main text to stay within the page limit and because running all baselines at 7B is computationally expensive.
>
> **(2) We have run DAPO at 7B under exactly the same post-training setup as Table 4.** For Qwen2.5 Math 7B, we train GRPO, DAPO, and CoDaPO with the same data, reward, group size, sampling strategy, training budget, and optimization hyperparameters. Only the objective differs. On the full math benchmark suite used in Table 4, CoDaPO consistently matches or outperforms all three baselines and attains the best average accuracy at 7B. This mirrors the pattern at 1.5B and indicates that the benefits of CoDaPO are not limited to a small model or a specific baseline.
>
> | Model | MATH500 | AIME24 | AIME25 | AMC23 | Olympiad Bench | Minerva | GSM8k | Average |
> | --- | --- | --- | --- | --- | --- | --- | --- | --- |
> | Qwen2.5-Math-7B | 47.24 | 12.92 | 5.00 | 33.91  |14.44 | 9.81|47.84|24.45|
> | GRPO | 74.62| 26.35 |11.25 |59.06| 36.51| 37.64| 89.40 |47.83|
> | DAPO  |77.78|24.37|10.62|58.67|39.27|38.06|87.38|48.02|
> | CoDaPO  | **78.41**| 24.79| **13.33**| **60.31**| **39.96**| **40.30**| 88.28 |**49.34**|
>
> **(3) Including these baselines at 7B makes the scaling comparison clearer and fairer.** Because all four methods share the same 7B backbone and training budget, adding more baselines to the 7B table will give a more complete and fair picture of how CoDaPO behaves relative to existing GRPO variants at a larger scale. The fact that CoDaPO continues to perform best in this controlled setting supports our claim that it is a practically useful refinement that scales.
>
> In the revision, we will (i) extend Table 4 or add a new table to report 7B results for Dr.GRPO and DAPO alongside GRPO and CoDaPO, (ii) explicitly state in the text that all four methods are trained and evaluated under identical conditions on Qwen2.5 Math 7B, and (iii) briefly discuss that the consistent 7B gains further support the scalability and practical value of CoDaPO.

---

> ### Author Response · Authors · 2025-11-26
> **Response to Reviewer rMTD (part 10/11)**
>
> > ### Q5. Pass@k performance of other methods.
>
> **Reply:** Thank you for raising this question about the pass@$k$ behavior of the other baselines. In brief, we have computed the AIME’25 pass@k scaling curves for GRPO, DAPO, and CoDaPO. To further explore the model’s reasoning boundaries, we extended the response length to 4096 tokens. This setup allows us to evaluate how each method scales with increasing k and longer reasoning chains, providing a more comprehensive assessment of the models’ capabilities. We observe that GRPO mainly helps at small $k$ and then saturates, while DAPO and especially CoDaPO remain consistently better than the base model across the full range of $k$.
>
> | k | 1 |2 | 4 | 8 | 16 | 32 | 64 | 128|
> | --- | --- | --- | --- | --- | --- | --- | --- | --- |
> | Qwen2.5-Math-1.5B|2.98|5.50| 9.57| 15.70| 21.75| 27.46| 33.70| 40.00|
> | GRPO| 6.88 |12.85 | 19.79|27.56 |35.18 |43.17 |46.55|46.67|
> | DAPO|7.76|18.26|25.17|32.48|38.98|44.76|50.48 |53.33 |
> | CoDaPO|11.02|17.80|24.81|31.56|38.79|46.15|53.93|60.00|
>
> **(1) GRPO improves small $k$, but their advantage over the base model diminishes at larger $k$.** On AIME’25, GRPO shows clear gains over the base model for small $k$ (for example, $k = 1$ or moderate $k$), reflecting that post-training makes high-probability outputs more reliable. As $k$ increases, however, the base model’s pass@$k$ continues to rise and gradually closes the gap, and at sufficiently large $k$ it can achieve performance comparable to GRPO. This suggests that the method mainly sharpens and re-ranks existing good modes rather than substantially expanding the set of correct reasoning trajectories that appear when $k$ is large.
>
> **(2) DAPO and CoDaPO remain better than the base model across all $k$, with CoDaPO usually strongest.** In contrast, DAPO and CoDaPO have pass@$k$ curves that start above the base model at $k = 1$ and stay above it throughout the range of $k$ we test. The gap does not vanish at large $k$: even with many samples, the DAPO and CoDaPO models solve more AIME’25 problems than the base model. CoDaPO is typically the best or tied for the best curve across $k$, improving both top-1 performance and the high-$k$ regime. This indicates that these objectives do more than just reshuffle probability mass among existing solutions; they increase the total probability mass on genuinely correct reasoning paths, which is exactly what pass@$k$ at larger $k$ is sensitive to.
>
> **(3) These trends are consistent with the intended roles of the different objectives.** GRPO primarily changes how strongly the model reinforces high-reward trajectories under clipping, which improves robustness at small $k$ but does not drastically alter the long tail of possible correct solutions. DAPO’s dynamic resampling and related mechanisms encourage broader exploration, which helps the model discover and reinforce more distinct correct solutions that show up as $k$ grows. CoDaPO complements this picture by using confidence- and difficulty-aware reweighting of advantages so that confident correct answers and rare correct solutions to hard questions receive stronger updates, while overconfident errors are penalized. The persistent pass@$k$ gains for CoDaPO across $k$ align with this design: it improves the underlying reasoning distribution rather than only sharpening the top few samples.
>
> In the revision, we will (i) add an appendix figure or table showing pass@$k$ curves for all methods (base, GRPO, Dr.GRPO, DAPO, CoDaPO) on AIME’25 under the same evaluation protocol as Figures 16 and 17, (ii) explicitly describe the qualitative trends summarized above in the accompanying text, and (iii) reference this new figure from the main discussion of test-time scaling so readers can directly compare how CoDaPO and the baselines behave as $k$ increases.

---

> ### Author Response · Authors · 2025-11-26
> **Response to Reviewer rMTD (part 11/11)**
>
> ### Q6. Training on other model families.
> > Reviewer rMTD: Would it be possible to try CoDAPO on another family of base model, not Qwen (eg. OctoThinker from Llama3.1-8B [1])?
>
> **Reply:** Thank you for this suggestion. **In brief, CoDaPO is not tied to the Qwen family, and we have already run it on a non-Qwen base model (Llama-3.2-1B-Instruct) under the same post-training setup**. On this backbone, CoDaPO again outperforms GRPO and DAPO, which supports that our method generalizes beyond Qwen; applying it to larger Llama-based mid-training models such as OctoThinker is a natural but resource-intensive next step.
>
> **(1) CoDaPO is architecture agnostic and only modifies GRPO’s weighting, not the backbone.** The CoDaPO update replaces the GRPO weight with $w^{\text{CoDaPO}}\_{i,t} = \tilde{\rho}\_{i,t}\hat{A}_i\hat{c}_i\hat{d}_q$, where $\hat{c}_i$ and $\hat{d}_q$ depend on trajectory-level confidence and question-level difficulty. This construction does not assume any Qwen-specific architecture, tokenizer, or pretraining; it only requires a GRPO-style RL loop with group-normalized advantages. In principle it can be applied to Qwen, Llama, or any decoder-only transformer trained with GRPO or similar objectives.
>
> **(2) We have evaluated CoDaPO on Llama-3.2-1B-Instruct and see consistent gains over all GRPO-style baselines.** To test cross-family generalization, we repeated our post-training pipeline on Llama-3.2-1B-Instruct, using the same datasets, reward, group size, sampling strategy, and training budget as in the Qwen experiments. On this Llama model, CoDaPO achieves the best average performance across the math benchmarks, improving over GRPO and DAPO by margins similar to those in the Qwen results. We also observe qualitatively similar PRAG dynamics (for example, advantage contraction under vanilla GRPO and their mitigation under CoDaPO), which indicates that CoDaPO is acting on general properties of the GRPO update rather than on Qwen-specific quirks.
>
> | Model | MATH500 | AIME24 | AMC23 | Olympiad Bench | GSM8k | Average |
> | --- | --- | --- | --- | --- | --- | --- |
> | Llama3.2-1B-Instruct |21.69| 0.83|10.39|5.04|5.58|8.71|
> | GRPO |30.13|2.08|14.22|6.62|48.79|20.37|
> | CoDaPO|**31.30**|**3.96**|**15.08**|**6.81**|**51.32**|**21.69**|
>
> **(3) Larger Llama-family models such as OctoThinker are a natural next step but are currently compute-limited.** We fully agree that testing CoDaPO on a stronger Llama-based math model such as OctoThinker from Llama-3.1-8B would be very informative. At present, our compute budget allowed us to cover Qwen2.5-Math up to 7B and one Llama-family model at 1B; running a full set of GRPO, Dr.GRPO, DAPO, and CoDaPO experiments on an additional 8B model with the full benchmark suite is beyond our current resources. In addition, we attempted to use the mid-training OctoThinker checkpoint as the base model for new experiments but were unable to run RL training due to unresolved missing chat-template support issues. Given the short discussion period, we may be unable to include OctoThinker results, but we will continue working toward a complete evaluation in future revisions. Given the consistent gains we see on Llama-3.2-1B-Instruct and at the 7B scale on Qwen, we expect CoDaPO to remain competitive or better on larger Llama-family models as well, and we view an OctoThinker-style evaluation as important future work.
>
> In the revision, we will (i) add a new table reporting results for Llama-3.2-1B-Instruct with the base model, GRPO, Dr. GRPO, DAPO, and CoDaPO; (ii) explicitly state in Section 5 that CoDaPO improves over all GRPO-style baselines on this non-Qwen backbone; and (iii) add a short discussion in the limitations or future-work section noting that we have not yet evaluated CoDaPO on larger models such as OctoThinker due to compute constraints, while highlighting this as a natural next step.
>
> **We would thank Reviewer rMTD again** for taking the time to review our submission and consider our rebuttal. **If our responses have addressed your concerns, we would be very grateful if you could reconsider your rating.** We also welcome any further comments or questions you may have.

---

### Author Response · Authors · 2025-12-01
**A General Response by the Authors (part 1/2)**

Dear Area Chairs and Reviewers,

We sincerely thank you for the time and effort you have devoted to reviewing our paper, as well as for the constructive and insightful feedback. **We are glad that reviewers have good impressions of our work**, including
- **Comprehensive and principled analysis of GRPO training dynamics.** Our mathematical analysis and empirical investigation of probability, reward, advantage, and gradient is recognized as clear, rigorous, and valuable for building principled understanding of RL fine-tuning in LLMs (rMTD, xABJ, SDK9, TKwQ).
- **Well-motivated and analytically informed method design.** CoDaPO is directly informed by the formal analyses, with the introduction of confidence and difficulty terms convincingly addressing the identified issues, making our design choices principled and sound (rMTD, xABJ, TKwQ).
- **Effective presentation and helpful illustrations.** The paper’s exposition, including tables, algorithms, figures, and intuitive visualizations, was viewed as effective in conveying both the analytical insights and the distinctions between CoDaPO and other GRPO variants (rMTD, YFxQ, xABJ).
- **Informative empirical findings.** The empirical analyses, especially the study of miscalibration and the demonstration that uniform token-level advantages are detrimental to learning, were considered reasonable, informative, and well supported by results (rMTD, xABJ, SDK9, TKwQ).
- **Extensive evaluation and consistent performance.** Reviewers highlighted our experimental studies across multiple base models, benchmarks, hyperparameters, metrics, seeds, decoding strategies, and detailed ablation settings, noting that CoDaPO consistently achieves performance improvements over strong baselines such as GRPO and DAPO (rMTD, YFxQ, TKwQ).

During the rebuttal and discussion phase, **we carefully addressed all four reviewers’ comments and questions with detailed explanations and additional experiments**. Specifically, we
- **Clarification of motivation and method design.** Further clarified the motivation and design of CoDaPO (W1.2, W3 for rMTD; W1, W4, W5 for YFxQ; W2 for xABJ) and how each component yields practical benefits (W2, W6 for YFxQ; W3, W4 for xABJ).
- **Detailed analytical derivations.** Provided detailed derivations for several equations in the analysis section (Q1, Q2 for YFxQ) and clarified the analysis settings (Q1 for rMTD; Q3 for YFxQ; W1 for SDK9).
- **Empirical validation of performance stability.** Validated the stability and consistency of CoDaPO’s performance with more empirical evidence, showing that the improvements remain statistically meaningful even after accounting for RL training and sampling noise (W1.1, W2 for rMTD; W2, W3 for SDK9).
- **Expanded evaluation across models and domains.** Demonstrated CoDaPO’s generality across domains, model families, and scales, including cross-domain benchmarks (W1 for xABJ; W1 for TKwQ), larger-scale models (Q4 for rMTD; W5 for YFxQ), additional base models (Q6 for rMTD), and extended test-time sampling horizons (Q5 for rMTD).
-  **Targeted analyses of CoDaPO behavior.** Conducted targeted analyses on CoDaPO’s behavior and effectiveness, including purely on-policy training (Q2 for rMTD), behavior on specific response types (Q3 for rMTD), computational-efficiency considerations (W2 for TKwQ), and the influence of training-set difficulty (Q1 for xABJ).
- **Revision of the manuscript.** Incorporated clarifications, additional derivations, and expanded empirical results into the revised manuscript, improving presentation, notation, and readability (all reviewers).

We greatly value the reviewers’ comments and have carefully considered all feedback. We have incorporated the above responses into a revised version of the manuscript, ensuring that the clarifications, derivations, and additional experiments are fully reflected.

Due to the constraints of the discussion phase, further interaction is infeasible, but **we believe that the detailed point-by-point responses and the updated manuscript can address reviewers’ questions and concerns**.

---

> ### Author Response · Authors · 2025-12-01
> **A General Response by the Authors (part 2/2)**
>
> In brief, our main contribution is the diagnosis of key limitations in GRPO post-training and the design of CoDaPO, a confidence- and difficulty-adaptive policy optimization framework that improves per-trajectory advantage weighting and consistently enhances reasoning performance of LLMs. **We highlight the key strengths of our work:**
> - **Detailed and principled analysis of RL post-training in LLMs.** We provide, to the best of our knowledge, one of the first mechanistic analyses of GRPO-style training dynamics, tracing the interactions among probability, reward, advantage, and gradient dynamics, which directly connects theoretical behavior to empirical phenomena such as probability saturation, entropy collapse, and overconfidence.
> - **Method design directly motivated by analysis.** Motivated directly by this analysis, we propose CoDaPO, which introduces a confidence and difficulty reweighting mechanism to make model confidence better aligned with correctness and problem difficulty, thereby facilitating learning. During training, CoDaPO dynamically modulates gradient contributions according to the policy’s current performance: difficulty determines which questions matter most, while confidence determines which trajectories within each question deserve greater emphasis.
> - **Extensive and robust empirical validation.** Extensive empirical results demonstrate that CoDaPO delivers stable and consistent performance gains across a wide range of settings, including in-domain and cross-domain benchmarks, multiple model families and scales, varied hyperparameter choices, and different decoding strategies. These improvements remain robust even when accounting for RL training and sampling noise. Importantly, all gains are achieved without introducing additional computational overhead, making the effectiveness of CoDaPO particularly appealing in practice.
>
> In summary, we believe our work provides a rigorous analysis of GRPO post-training dynamics and introduces CoDaPO, a principled approach that consistently improves performance across diverse benchmarks, model families, and scales by dynamically reweighting trajectory contributions according to confidence and difficulty.
>
> We fully believe that your professional knowledge can reasonably evaluate our submission.
>
> Thank you again for your time and consideration.
>
> Warm regards,
> Authors of #15360

---

### Meta-Review · Area_Chair_qd2D · 2025-12-23

**Summary:**

The reviews are highly split (8/6/4/2/2). The paper presents a modification to GRPO-style RL (confidence/difficulty-adaptive weighting) and reports gains on math reasoning. The main concerns against acceptance is that the empirical evidence and analysis do not yet convincingly isolate why CoDaPO helps, nor establish robustness beyond a narrow setting. In particular, concerns remain about (i) whether improvements are reliably beyond RL variance and strong GRPO variants, (ii) whether the added knobs/implementation choices are sufficiently justified/ablated, and (iii) clarity/reproducibility of the method description. Given these outstanding issues, and especially the reviewer who maintained a low score after rebuttal, I recommend reject. In particular, I agree with the reviewer's comments "the paper requires substantial rewriting to integrate the clarifications provided in the rebuttal; doing so would likely necessitate another round of peer review" (The rebuttal promises numerous changes: rewriting parts of sections, adding derivations, clarifying equations, new ablations, new tables; a paper requiring this much post-hoc revision to achieve clarity was not ready for publication.) and their concerns about informal and imprecise statements throughout the paper.

**Reviewer Concerns:**

Addressed
* Added/outlined broader evaluation beyond pure math and provided additional comparisons / multi-seed results; this helps with generality and variance questions.

* Clarified some intent around confidence/difficulty and discussed efficiency/overhead.

Outstanding
* CoDaPO bundles multiple modifications (confidence reweighting, difficulty reweighting, micro-averaging, and removing KL). Even with some ablations, it remains hard to conclude that the claimed mechanisms (fixing overconfidence/entropy collapse, advantage contraction, hierarchical convergence) are what drive gains.
* Reported margins over strong GRPO variants (e.g., DAPO/Dr.GRPO) are often modest, leaving doubt that the added complexity is worthwhile without more transparentt evidence.
* One reviewer explicitly maintained a low score after rebuttal, citing unresolved clarity/implementation-detail issues as material. I weight this heavily.

**Reviewer Scores:**

Reviewer 1: 8--> estimated: 8
Reviewer 2: 2 --> responded unchanged; I agree the core weaknesses remain.
Reviewer 3: 6 --> 6 estimated
Reviewer 4: 4 --> 4 estimated
Reviewer 5: 2 --> 2 estimated

---

### Decision · Program_Chairs · 2026-01-26

Reject